# Towards Interpretable Steering for Hallucination Mitigation in Large Vision–Language Models

## ABSTRACT

Large vision-language models (LVLMs) have achieved impressive performance in multimodal understanding and generation, yet they remain prone to *hallucinations*, particularly object hallucinations where entities are described yet do not exist in the input image. Existing mitigation methods often focus on output-level adjustments, while the internal mechanisms driving hallucinations remain poorly understood. In this work, we adopt an internal representation-level perspective by introducing sparse autoencoders (SAEs) to decompose dense visual features into sparse monosemantic neurons for interpreting and steering LVLMs. Building on prior findings that injecting image noise exacerbates hallucinations, we further investigate how noise perturbations reshape internal representations, revealing that noise alters monosemantic neuron activations, disrupts visual semantics, and induces hallucinations. Furthermore, we show that manipulating specific neurons enables controllable influence over LVLM outputs. Based on these insights, we propose **Contrastive Neuron Steering (CNS)**, which selectively amplifies truth neurons while suppressing perturbation-induced activations to mitigate hallucinations, and further enhances understanding of image-specific features through *adaptive neuron constraints* and *always-on neuron suppression*. Extensive experiments and analyses demonstrate that CNS effectively reduces hallucinations. Moreover, our CNS enables interpretable and controllable internal neuron-level interventions, providing both practical mitigation and mechanistic insights into how LVLMs encode and sometimes misrepresent visual information.

## 1 INTRODUCTION

Large vision-language models (LVLMs) (Liu et al., 2023b; Dai et al., 2023; Bai et al., 2023; Zhu et al., 2023) have achieved remarkable progress in multimodal understanding and generation. Despite these advances, LVLMs remain vulnerable to *hallucinations*, particularly object hallucinations where the model describes entities that are not present in the input image (Lee et al., 2018; Leng et al., 2024). Such errors undermine reliability and user trust, while raising critical concerns for safety-sensitive applications such as autonomous systems, medical imaging, and decision support.

To mitigate hallucinations, numerous techniques have been investigated, including visual instruction fine-tuning (Liu et al., 2023b; 2024b; Yu et al., 2024a), integration with external expert models, and contrastive decoding strategies (Leng et al., 2024; Chen et al., 2024; Favero et al., 2024; Wan et al., 2025). Nevertheless, the mechanistic origins of hallucinations remain poorly understood. Existing explanations predominantly attribute hallucinations to language biases, such as the "anchor pattern" (Huang et al., 2023) and "text inertia" (Liu et al., 2024d), which posit that hallucinations emerge from the dominance of linguistic priors over visual features. However, these perspectives largely neglect the internal visual representation space of LVLMs. In this paper, we seek to explore the relationship between internal visual representations and hallucinations, addressing the following fundamental questions: how are visual features organized internally, how they change under perturbations, and which aspects of the representation most directly contribute to hallucinations?

To enable deep and comprehensive analysis, we adopt an internal representation-level perspective to address these questions. The complex, entangled visual features produced by LVLM encoders

are difficult to interpret and control. To make them tractable, we introduce sparse autoencoders (SAEs) (Makhzani & Frey, 2013; Templeton et al., 2024), which have shown strong promise in interpretability research for large language models. By applying SAEs to LVLMs, we decompose dense embeddings into sparse neurons that tend to represent interpretable, concept-specific features (Durmus et al., 2024; Templeton et al., 2024). This enables us to analyze the drivers of hallucinations and design interventions directly within the internal representation space.

Prior works (Leng et al., 2024; Wan et al., 2025) have shown that injecting image noise amplifies visual uncertainty, aggravates hallucinations. We leverage the interpretable latent space to probe how such perturbations manifests in internal visual representations. Through extensive analysis, we find that as noise increases, an increasing number of neurons undergo activation changes, which alters the semantic structure of visual representations and ultimately exacerbates hallucinations and degrades performance (Figs. 5, 8). Our global image-level (Figs. 4, 9) and local patch-level analyses (Figs. 3, 10) further reveal two key patterns: a subset of "always-on" neurons consistently dominates activations while encoding generic global information, whereas most neurons capture concrete, meaningful visual features. Moreover, we demonstrate (Figs. 2, 13, 14, 11) that enhancing or suppressing specific neurons in the sparse space can strengthen or diminish the model's ability to recognize particular concepts. Together, these findings show that noise reshapes the semantic structure of visual features, thereby inducing hallucinations, and importantly, that neuron-level interventions in the sparse space provide a tractable means to steer LVLMs.

Building on these insights, we propose a novel and efficient method, **Contrastive Neuron Steering (CNS)**, for hallucination mitigation from the perspective of internal visual representation space. Specifically, CNS employs noisy images to activate hallucination-related neurons and contrasts them with neurons derived from clean images. To selectively enhance informative neurons while suppressing unstable ones, we design an *adaptive neuron constraint* incorporating both positional and magnitude regularization. Furthermore, to mitigate the influence of *redundant and non-informative* activations and sharpen attention to image-specific features, we introduce *always-on neuron suppression*, which explicitly down-weights neurons persistently active across all images. By directly operating within the visual representation space, CNS offers an effective and complementary solution for hallucination mitigation that remains fully compatible with existing decoding-based approaches.

Extensive experiments across multiple LVLMs and diverse benchmarks demonstrate that CNS substantially reduces hallucination rates. In addition, our detailed analyses and visualizations highlight the interpretability of neuron-level interventions. Together, these findings show that CNS not only improves the reliability of LVLMs in practice but also advances mechanistic understanding of internal visual representations and their role in hallucinations.

In summary, our contributions are as follows:

- We introduce SAEs to interpret and steer the internal visual representations of LVLMs, providing extensive analyses and visualizations that reveal how image noise perturbs neurons, disrupts visual semantics, and ultimately induces hallucinations.

- We find that neuron-level interventions, such as enhancing or suppressing specific neurons in the internal visual representations, can modulate LVLM outputs for targeted concepts, and that coordinating multiple neurons is more effective than manipulating single neurons.

- We propose **CNS**, which amplifies meaningful neurons while suppressing perturbation-induced activations for hallucination mitigation. CNS is compatible with decoding-based mitigation approaches and consistently reduces hallucinations across diverse benchmarks.

## 2 RELATED WORK

**Hallucinations in LVLMs.** LVLMs (OpenAI et al., 2024; Anthropic, 2024; DeepSeek-AI et al., 2025; Comanici et al., 2025; Yang et al., 2025a) have achieved significant progress by combining visual encoders with large language models, enabling multimodal understanding and generation. However, these models remain prone to hallucinations, particularly object hallucinations (Liu et al., 2024a; Lee et al., 2023; Gunjal et al., 2024; Chen et al., 2024; Chuang et al., 2023), where the model generates references to objects not present in the image. The causes include pretraining data biases (Agarwal et al., 2020; Agrawal et al., 2016), over-reliance on parametric knowledge (Leng et al., 2024; Lee et al., 2023; Zhibo et al., 2023), and biased visual feature learning (Zhu et al., 2024; Huang et al., 2023; Yue et al., 2024; Han et al., 2022).

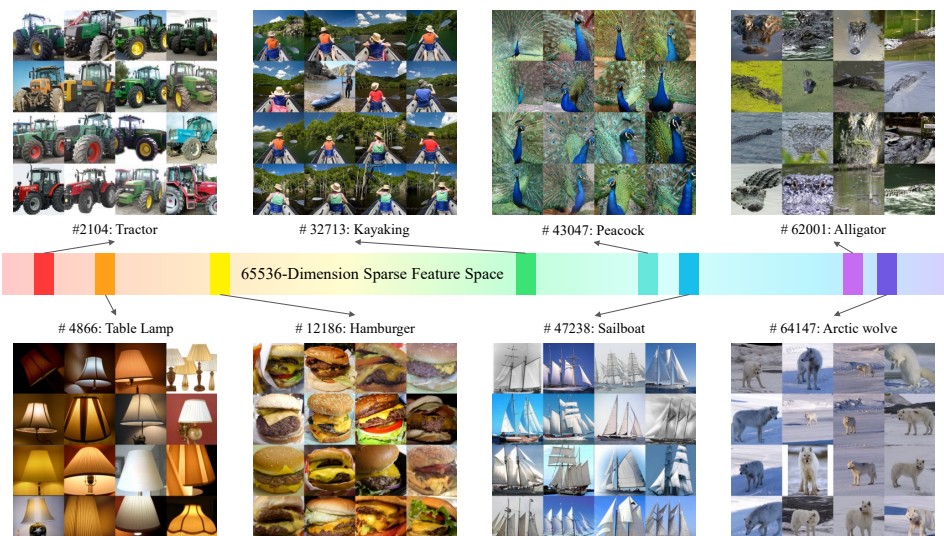

Figure 1: Neuron visualizations from SAE, showing diverse visual patterns and semantic structures.

Existing mitigation strategies fall into two groups: training-driven and training-free. Training-driven approaches fine-tune LVLMs via data augmentation, or reinforcement learning (Liu et al., 2023a; Sun et al., 2023; Zhou et al., 2024a; Liu et al., 2024b; Zhai et al., 2024). Training-free approaches mainly rely on contrastive decoding, which constructs positive/negative pairs to adjust inference-time generation (Yin et al., 2023; Park et al., 2025a; Wang et al., 2024; Li et al., 2023a).

**SAEs for Interpreting and Steering LVLMs.** SAEs (Templeton et al., 2024; Pach et al., 2025; Shu et al., 2025) decompose hidden activations into sparse, monosemantic neurons, providing an interpretable basis for analyzing and steering LVLMs. Recent improvements enhance both sparsity and reconstruction, including BatchTopK (Bussmann et al., 2024a), JumpReLU (Rajamanoharan et al., 2024), and hierarchical Matryoshka variants (Nabeshima, 2024; Bussmann et al., 2024b).

In LLMs, SAEs have been applied to explanation and control (Templeton et al., 2024; Durmus et al., 2024), enabling neuron-level steering to reduce toxicity, sycophancy, or refusal (Gallifant et al., 2025; Nanda et al., 2024), as well as facilitating hallucination detection (Ferrando et al., 2025), in-context learning (Demircan et al., 2025), and improved safety (Wu et al., 2025). Extensions to vision and multimodal domains include Revelio (Kim et al., 2024a), which uncovers interpretable features in diffusion models; Matryoshka SAEs (MSAEs) (Bussmann et al., 2024b), which balance sparsity and reconstruction on CLIP embeddings; and Universal SAEs (USAEs) (Thasarathan et al., 2025), which align concepts across networks. For LVLMs, SAE-V (Lou et al., 2025) enables fine-grained interpretation of cross-modal interactions, while Zhang et al. (Zhang et al., 2024a) show that disentangled features can be directly exploited to steer model behavior.

## 3 PRELIMINARIES: SPARSE AUTOENCODERS (SAEs)

**Background.** The hidden states inside LVLMs are dense and highly entangled, making attribution and control difficult. *SAEs* (Olshausen & Field, 1997; Bricken et al., 2023) address this issue by mapping dense embeddings into a sparse latent space with human-interpretable neurons. Formally, given an input feature $\mathbf{v} \in \mathbb{R}^d$, the SAE encoder produces sparse activations

$$z(\mathbf{v}) = \text{TopK}\big(\text{ReLU}(W_{\text{enc}}\mathbf{v} - \mathbf{b})\big), \tag{1}$$

and the decoder reconstructs the feature as

$$\hat{\mathbf{v}} = W_{\text{dec}}^{\top} z(\mathbf{v}) + \mathbf{b}. \tag{2}$$

This process can be viewed as learning an overcomplete dictionary of concepts, where each latent neuron corresponds to a basis element.

**Inserting SAEs into LVLMs.** SAEs can in principle be applied at different stages of LVLMs, such as intermediate LLM layers or the visual encoder. In this work, we focus on the *visual encoder stage* of LVLMs. This choice is motivated by both scientific and practical considerations: (1) It

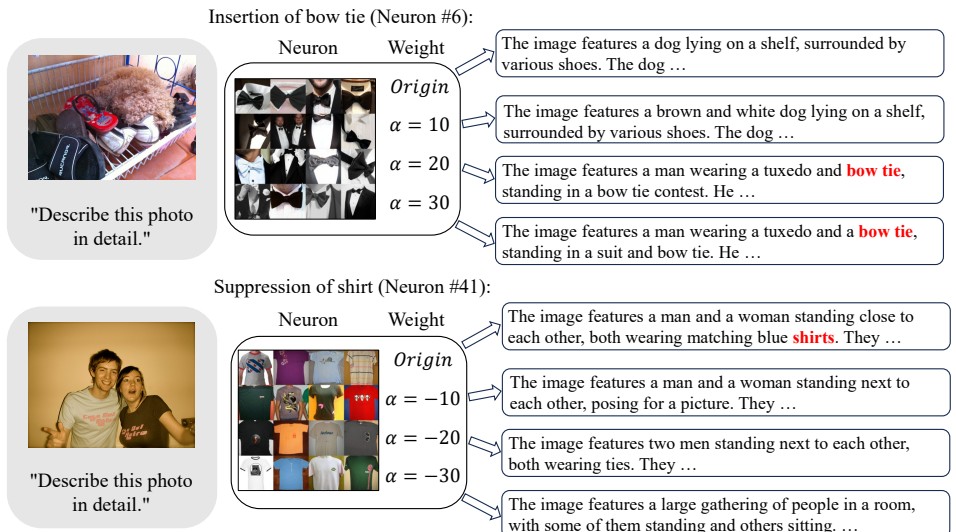

Figure 2: Steering an LVLM: (a) amplifying a "bow tie" neuron emphasizes this concept in generated descriptions, while (b) suppressing a "shirt" neuron prevents it from appearing.

enables deeper mechanistic studies of hallucinations by isolating the specific impact of the visual encoder, distinguishing whether errors arise from corrupted or ambiguous visual encodings, cross-modal misalignment during fusion, or the decoder's reliance on language priors. (2) Some LVLMs adopt same visual backbones, so an SAE trained on one backbone can be reused across LVLMs that share it. (3) Operating on encoder outputs preserves the downstream fusion and decoding pipeline, making SAE-based interventions compatible with existing decoding-level mitigation methods such as contrastive decoding. (4) Inserting SAEs into the visual encoder requires only a single additional encoder forward pass, making it much more efficient than full model re-inference.

**Interpreting and Steering LVLMs.** The sparse latent space of SAEs exhibits two key properties: (1) Sparsity: only a few neurons are active per input, making the representation sparse. Moreover, the magnitude of each active neuron reflects its relative importance, providing an inherent measure of feature relevance; (2) Monosemanticity: each neuron tends to encode a single, consistent concept, in contrast to polysemantic neurons in dense embeddings. To visualize these learned neurons, we identify, for each neuron, the top-16 images with the highest activation values, as shown in Figs. 1, 7. This allows inspection of the concepts captured by each neuron. These properties make SAEs a natural tool for both *interpreting* and *steering* LVLMs. On the analysis side, the neuron dictionary provides a principled way to *interpret* model behavior, enabling us to track how visual features shift under perturbations or correlate with hallucinations. On the control side, SAEs support *neuron-level steering*: by selectively *amplifying* or *suppressing* neurons, we can guide LVLM outputs toward or away from specific concepts. For example, amplifying a "bow tie" neuron emphasizes this concept in generated descriptions, while suppressing a "shirt" neuron prevents it from appearing (Fig.2). In summary, SAEs provide fine-grained interpretability of internal LVLM representations and serve as a foundation for both automated steering nd user-guided feature control.

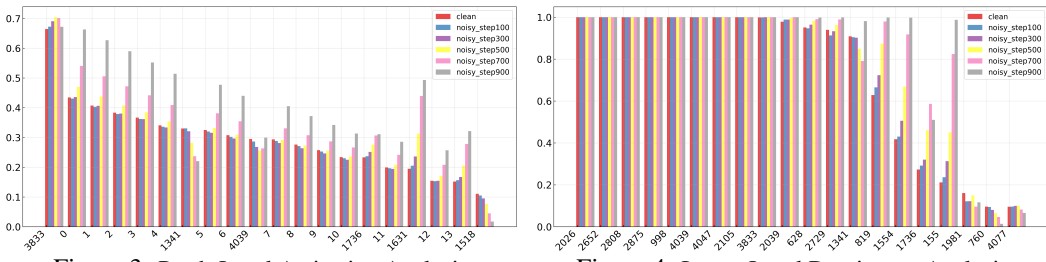

Figure 3: Patch-Level Activation Analysis      Figure 4: Image-Level Dominance Analysis

# 4 ANALYZING HALLUCINATION WITH SAEs

We train Matryoshka SAE (Bussmann et al., 2024b) on LLaVA-1.5's image features extracted from ImageNet, and conduct analyses on the COCO dataset, considering both clean and noisy images.

**Neuron-Level Statistical Analysis.** Beyond offering interpretability and controllability, SAEs also provide a diagnostic lens into how LVLMs encode visual information. We perform two complementary analyses to characterize neuron activations (Figs. 3, 4): (1) **Patch-Level Activation Analysis**: for each spatial patch, we record the Top-$K$ neurons, which highlight detectors of local features such as textures, edges, or object parts. (2) **Image-Level Dominance Analysis**: for each image, we compute the maximum activation of each neuron across patches, and record the Top-$N$ globally dominant neurons. This reveals neurons encoding coarse, image-wide concepts.

A surprising observation emerges from the image-level analysis: a small group of *Always-on neurons* (about 10 out of 65k) consistently appear in the Top-20 across nearly all images 9. Most of these neurons are not particularly strong at the patch level, indicating that they do not correspond to specific local objects, but rather to global statistical regularities such as smooth regions, edge density, or background color distributions. Moreover, these neurons activate across images, yet their activation sets strongly overlap with each other, suggesting that they encode over-generalized "pseudo-global concepts". In contrast, patch-level neurons (fig. 10) show semantic consistency: their high activations concentrate on visually similar images (e.g., cats, grass). Notably, we also identify a unique neuron that ranks highly under both patch-level and image-level statistics. Unlike other neurons, this neuron responds to complex multi-object scenes.

These phenomena resonate with recent studies on the limitations of SAEs. Bussmann et al. (Bussmann et al., 2024b; Nabeshima, 2024) report that standard SAEs often suffer from *feature absorption* and *feature splitting*, where coarse features are either overwritten by more specific ones or fragmented across multiple neurons. More recently, Chanin et al. (Chanin et al., 2025) identify the problem of *feature hedging*, where correlated features become entangled when the dictionary size is mismatched with the true feature complexity, producing latents that activate broadly but lack semantic specificity. Our observed always-on neurons strongly resemble such hedged features: they dominate global activations across diverse images yet encode ambiguous, over-generalized concepts.

**Diagnosing Hallucinations with SAEs.** Prior works (Leng et al., 2024; Wan et al., 2025) have shown that injecting image noise amplifies visual uncertainty and exacerbates hallucinations, resulting in performance degradation. We leverage the interpretable latent space of SAEs to probe how such perturbations reshape LVLM representations and induce hallucinations. To quantify this effect, we evaluate LLaVA-1.5 on the POPE benchmark (COCO, random setup) and report changes in F1 and accuracy under different perturbation conditions. We further examine how model performance is affected by selectively zeroing out different types of SAE neurons. Based on our earlier analysis, we roughly categorize SAE neurons into three groups: top-ranked always-active neurons (top-10), image-specific neurons (primarily within ranks 10–20), and ten randomly selected neurons from ranks beyond 20. This allows us to investigate the distinct characteristics of each neuron group and their respective impacts on model behavior.

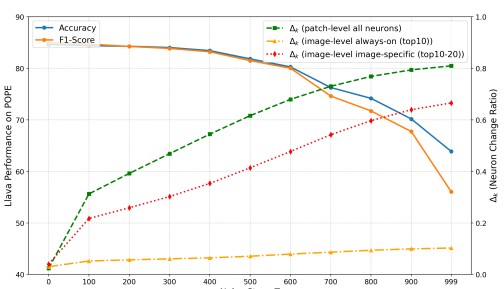

Figure 5: Relationship between noise step, model performance, and neuron change ratio.

Specifically, we measure the *stability of Top-$K$ neuron activations* across clean and perturbed inputs. Given a clean image $v$ and its noisy counterpart $\tilde{v}$, we define the change ratio as

$$\Delta_K(v, \tilde{v}) \;=\; 1 - \frac{|z(v) \cap z(\tilde{v})|}{K}, \tag{3}$$

Higher $\Delta_K$ indicates that more neurons change, reflecting larger disruption of the visual features.

As shown in Fig. 5, increasing noise steps degrades LLaVA performance while raising the neuron-change ratio. At the token level, patch-level neurons change the fastest, indicating that noise perturbs almost every token. At the image level, Always-active neurons remain largely unchanged, whereas image-specific neurons exhibit substantial variation, approaching the magnitude of patch-level changes. These results indicate that noise mainly disrupts image-specific neurons, which in turn triggers hallucinations. This also explains why VCD can use noise to induce hallucinations.

Table 1 illustrates the effects of zeroing different neuron groups on model predictions and hallucination behavior. The results show that suppressing Always-active neurons has minimal impact

Figure 6: Framework overview. A pretrained SAE is integrated into the LVLM visual encoder to decouple internal dense visual features into sparse, monosemantic neurons, enabling neuron-level interpretation and steering. **Contrastive Neuron Steering (CNS)** amplifies neurons activated by clean inputs while suppressing those triggered by perturbations, mitigating hallucinations.

despite their high baseline activations, and randomly selected neurons have negligible effects due to low relevance. In contrast, zeroing image-specific neurons induces substantial activation changes and significantly alters model outputs. These findings indicate that image-specific neurons are the primary drivers of input-dependent behavior and hallucination sensitivity, whereas Always-active neurons contribute little to image-specific reasoning.

Beyond aggregate statistics, qualitative case studies (Fig.8) further illustrate how noisy perturbations induce hallucinations. For instance, when noise is added to an image of a "camera", the activations of camera-related neurons gradually diminish as noise intensity increases, leading LLaVA to generate progressively less accurate captions: from "black Konica Minolta camera with a large lens" to "camera with a large lens," and finally no camera description.

Table 1: Performance impact of zeroing out different types of SAE neurons.

| Neuron Type | Accuracy (%) ↑ | F1-score (%) ↑ |
|---|---|---|
| baseline | 84.63 | 84.99 |
| always-on | 84.68 | 85.08 |
| image-specific | 63.08 | 57.36 |
| random | 84.31 | 84.65 |

Importantly, such fine-grained instability cannot be captured in the original dense feature space. By contrast, SAEs disentangle activations into sparse, interpretable neurons, allowing us to quantify exactly which concept units vanish and which spurious ones emerge.

In summary, image noise primarily perturbs token-level semantic neurons, especially image-specific neurons, reshaping internal visual representations and thereby inducing hallucinations. SAEs thus serve not only as a principled diagnostic tool to quantify these effects, but also as an interpretable lens into the mechanisms linking image-specific neuron disruptions, visual uncertainty, and hallucinations. Based on these findings, we can mitigate hallucinations by identifying image-specific neurons and enhancing their activations to strengthen input-relevant semantic information.

## 5 METHOD

### 5.1 OVERVIEW

Our objective is to improve the reliability of LVLMs by mitigating hallucinations while simultaneously enabling interpretable analysis and controllable steering. As shown in Fig. 6, we integrate a pretrained SAE into the LVLM visual encoder to decouple dense internal visual features into sparse, monosemantic neurons, facilitating neuron-level interpretation and steering. We further propose **Contrastive Neuron Steering (CNS)** to amplify image-specific neurons activated by clean inputs while suppressing those triggered by perturbations, thereby mitigating hallucinations.

## 5.2 Contrastive Neuron Steering (CNS)

Our exploratory analysis reveals three key observations: (i) perturbing images (e.g., by adding noise) significantly alters neuron activations, particularly those of image-specific neurons, and the extent of change correlates with LVLM performance degradation (Fig. 2); (ii) directly scaling individual neuron activations can increase or suppress concept-specific responses, thereby modulating generated outputs (Fig. 5); (iii) suppressing image-specific neurons induces severe hallucinations, whereas suppressing always-on neurons has minimal effect. Together, these findings suggest that targeted neuron-level regulation, especially of image-specific neurons, provides a natural and effective pathway for hallucination mitigation.

Building on these insights, we propose **CNS**. CNS leverages perturbed inputs to contrastively identify neuron types: neurons whose activations decrease under image perturbations are likely *image-specific neurons*, encoding input-dependent semantics, whereas neurons that remain stable or increase are typically *always-on neurons or low-importance, input-agnostic neurons*. CNS then selectively amplifies image-specific neurons while suppressing these non-informative neurons, thereby enhancing image-relevant semantic representations and mitigating hallucinations.

**Contrastive Neuron Regulation (CNR).** Given a clean image $v$ and a perturbed counterpart $v'$ (e.g., Gaussian noise), both are encoded via the SAE to obtain sparse activations:

$$z(v), z(v') \in \mathbb{R}^k. \tag{4}$$

We then compute a regulated activation vector:

$$\tilde{z} = (1 + \lambda)z(v) - \lambda z(v'), \tag{5}$$

where $\lambda$ controls the steering strength. This contrastive formulation promotes stability by reinforcing neurons consistent across perturbations and attenuating those highly sensitive to noise.

**Adaptive Neuron Constraints (ANC).** To ensure reliable and interpretable regulation, we impose two complementary constraints on the updated activations, each acting along a different dimension:

*Position constraint:* Only the neurons corresponding to the top-$K$ most active positions in the clean activations $z(v)$ are eligible for regulation:

$$\tilde{z}_i = \begin{cases} (1 + \lambda)z_i(v) - \lambda z_i(v'), & i \in \text{Top-}K(z(v)), \\ z_i(v), & \text{otherwise.} \end{cases} \tag{6}$$

This ensures that CNS updates only the most salient neuron positions, while leaving inherently inactive or irrelevant positions untouched, thereby avoiding unintended alterations.

*Magnitude constraint:* After contrastive updating, neurons with small activations may still be spurious. We therefore apply an adaptive threshold based on the maximum activation of the clean input:

$$\tilde{z}_i \leftarrow \begin{cases} \tilde{z}_i, & \tilde{z}_i \geq \tau(z(v)), \\ z_i(v), & \text{otherwise.} \end{cases} \tag{7}$$

This preserves only neurons whose updated activations remain sufficiently strong and semantically meaningful, effectively filtering out weak, unstable, or spurious signals.

Together, these two constraints ensure that CNS modifies only the most relevant neurons both in terms of *position* and *magnitude*, enhancing the stability and interpretability of neuron-level steering.

**Always-on Neuron Suppression (ANS).** Empirical analysis (Fig. 3 and Fig. 4) identifies the always-on neurons that consistently rank high across all images and likely encode generic concepts. Table 1 shows that suppressing these neurons has a minimal effect on the overall F1-score and can even slightly improve accuracy. Motivated by this, we propose ANS to reduce their influence and emphasize image-specific semantic information by setting their activations to zero.

$$\tilde{z}_i = \begin{cases} 0, & i \in \mathcal{A}, \\ z_i, & \text{otherwise,} \end{cases} \tag{8}$$

where $\mathcal{A}$ denotes the set of always-active neurons. This encourages the model to focus on image-specific and grounded features, reducing the propagation of generic or misleading signals.

**Reconstruction and Integration into LVLM.** The regulated latent $\tilde{z}$ is decoded into a dense visual embedding that replaces the original encoder output:

$$\hat{\mathbf{v}} = W_{\text{dec}}^{\top}\tilde{z} + \mathbf{b}. \tag{9}$$

## 5.3 Plug-and-play Compatibility

CNS operates at the *representation level*, while contrastive decoding methods like VCD act on the *output distribution*. These approaches are naturally complementary: CNS refines encoder features by amplifying grounded neurons and suppressing noisy ones, producing enhanced embeddings that directly replace the original image features. This allows seamless integration with various decoding-based methods for further hallucination mitigation.

## 6 Experiments

### 6.1 Experimental Settings

**Evaluated LVLMs.** Following prior work, we use three representative open-source LVLMs: LLaVA-1.5 (Liu et al., 2023b), InstructBLIP (Dai et al., 2023), and Qwen-VL (Bai et al., 2023).

**Benchmarks.** (1) **POPE** (Li et al., 2023b), a widely used benchmark for assessing object hallucinations in LVLMs through binary yes/no queries about object existence; (2) **CHAIR** (Rohrbach et al., 2018), which evaluates object hallucinations in image captioning, where LVLMs are prompted to describe 500 randomly selected images from the MSCOCO validation set; (3) **MME-Hallucination** (Fu et al., 2023), a comprehensive benchmark consisting of four subsets: *existence* and *count* for object-level hallucinations, and *position* and *color* for attribute-level hallucinations.

**Baselines.** We compare with existing methods: VCD (Leng et al., 2024), M3ID (Favero et al., 2024), and ONLY (Wan et al., 2025). Unless otherwise specified, we adopt sampling-based decoding as the default, where the next token is sampled directly from the post-softmax probability distribution.

**Implementation Details.** We train SAEs on ImageNet (Deng et al., 2009) using image features extracted from the visual encoder of each LVLM (before the projection layer). We experiment with Matryoshka BatchTopK SAEs (Bussmann et al., 2024b). Unless otherwise stated, the Matryoshka groups are set as $\mathcal{M} = \{0.0625\omega, 0.1875\omega, 0.4375\omega, \omega\}$, approximately doubling the number of active neurons per level. We fix the maximum number of non-zero latent neurons to $K = 20$ and set the expansion factor to $64$. All SAEs are trained for $10^5$ steps with a batch size of 4096, using Adam (Kingma & Ba, 2017) with the learning rate $\frac{16}{125\sqrt{\omega}}$, as suggested by (Gao et al., 2025).

For LVLM experiments, we follow the default query format of each model. Following VCD (Leng et al., 2024), we adopt adaptive plausibility constraints (Li et al., 2023a) with $\beta = 0.1$ and $\alpha = 0.5$. The number of denoising steps is fixed to $500$ unless explicitly stated otherwise. All experiments are conducted on a single NVIDIA RTX 6000 Ada GPU (48GB).

### 6.2 Experimental Results

**Results on POPE.** As shown in Tab. **??**, inserting our method into different baselines consistently improves performance across multiple LVLM backbones and evaluation settings. These results demonstrate that CNR is both robust and broadly generalizable.

**Results on CHAIR.** On the open-ended CHAIR benchmark (Tab. 3), integrating our method into various baselines significantly reduces hallucination rates across different LVLMs. This verifies that CNR enhances open-ended multimodal understanding and yields more faithful generations.

**Results on MME.** In Tab. 4, we report results after inserting our method into existing baselines on the MME benchmark. While evaluated within the same backbone setting, our method consistently improves both object-level (Existence, Count) and attribute-level (Position, Color) scores, confirming that CNR strengthens grounding and fine-grained visual reasoning.

Overall, the CNR-adjusted features can be viewed as enhanced representations of the original visual input, reinforcing grounded evidence while suppressing noise and spurious activations. These results verify that CNR serves as a plug-and-play module that universally boosts the reliability of LVLMs.

Table 2: Results on the POPE. ↑ indicates higher is better. +CNS denotes that the original visual features are replaced with CNS-enhanced. The best and second results are **bolded** and underlined.

| Setup | | Method | LLaVA-1.5 | | | | InstructBLIP | | | | Qwen-VL | | | |
|---|---|---|---|---|---|---|---|---|---|---|---|---|---|---|
| | | | Acc. ↑ | Prec. ↑ | Rec. ↑ | F1 ↑ | Acc. ↑ | Prec. ↑ | Rec. ↑ | F1 ↑ | Acc. ↑ | Prec. ↑ | Rec. ↑ | F1 ↑ |
| MS-COCO | Random | Regular | 84.63 | 83.07 | 87.00 | 84.99 | 83.33 | 82.38 | 84.80 | 83.57 | 85.17 | 97.22 | 72.40 | 83.00 |
| | | Regular + CNS | 85.10 | 83.77 | 87.07 | 85.39 | 84.40 | 83.42 | 85.87 | 84.63 | 86.03 | 97.37 | 74.07 | 84.13 |
| | | VCD | 84.57 | 82.59 | 87.60 | 85.02 | 84.60 | 85.12 | 83.87 | 84.49 | 87.37 | 97.14 | 77.00 | 85.91 |
| | | VCD + CNS | 85.23 | 83.30 | 88.13 | 85.65 | 85.63 | 86.24 | 84.80 | 85.51 | 88.27 | 97.36 | 78.67 | 87.02 |
| | | M3ID | 86.33 | 85.30 | 87.80 | 86.53 | 85.00 | 84.72 | 85.40 | 85.06 | 86.03 | 97.87 | 73.67 | 84.06 |
| | | M3ID + CNS | 86.70 | 85.72 | 88.07 | 86.88 | 85.57 | 85.64 | 85.47 | 85.55 | 87.03 | 97.93 | 75.67 | 85.37 |
| | | ONLY | 89.57 | 90.68 | 88.20 | 89.42 | 86.13 | 86.04 | 86.27 | 86.15 | 89.63 | 95.70 | 83.00 | 88.90 |
| | | ONLY + CNS | 90.27 | 91.20 | 89.13 | 90.16 | 87.07 | 86.92 | 87.27 | 87.09 | 89.90 | 95.86 | 83.40 | 89.20 |
| | Popular | Regular | 81.33 | 78.14 | 87.00 | 82.33 | 76.00 | 72.11 | 84.80 | 77.94 | 84.50 | 94.73 | 73.07 | 82.50 |
| | | Regular + CNS | 82.70 | 79.60 | 87.93 | 83.56 | 77.07 | 73.44 | 84.80 | 78.71 | 84.77 | 94.76 | 73.60 | 82.85 |
| | | VCD | 80.80 | 77.11 | 87.60 | 82.02 | 77.20 | 74.00 | 83.87 | 78.62 | 85.83 | 94.02 | 76.53 | 84.38 |
| | | VCD + CNS | 81.53 | 77.69 | 88.47 | 82.73 | 78.23 | 75.10 | 84.47 | 79.51 | 86.17 | 94.21 | 77.07 | 84.78 |
| | | M3ID | 82.30 | 79.10 | 87.80 | 83.22 | 77.23 | 73.41 | 85.40 | 78.95 | 85.43 | 95.94 | 74.00 | 83.55 |
| | | M3ID + CNS | 83.07 | 80.21 | 87.80 | 83.83 | 78.40 | 74.91 | 85.40 | 79.81 | 86.07 | 96.00 | 75.27 | 84.38 |
| | | ONLY | 86.10 | 84.64 | 88.20 | 86.39 | 77.50 | 73.40 | 86.27 | 79.31 | 87.70 | 91.92 | 82.67 | 87.05 |
| | | ONLY + CNS | 86.90 | 85.50 | 88.87 | 87.15 | 77.73 | 73.64 | 86.40 | 79.51 | 87.87 | 91.95 | 83.00 | 87.25 |
| | Adversarial | Regular | 75.87 | 71.18 | 86.93 | 78.27 | 74.17 | 70.04 | 84.47 | 76.58 | 82.53 | 90.80 | 72.40 | 80.56 |
| | | Regular + CNS | 76.13 | 71.33 | 87.40 | 78.55 | 75.03 | 70.85 | 85.07 | 77.31 | 83.20 | 91.29 | 73.40 | 81.37 |
| | | VCD | 75.23 | 70.23 | 87.60 | 77.96 | 75.80 | 72.29 | 83.67 | 77.56 | 83.10 | 88.46 | 76.13 | 81.83 |
| | | VCD + CNS | 75.83 | 70.67 | 88.33 | 78.52 | 76.80 | 73.16 | 84.67 | 78.49 | 83.70 | 89.16 | 76.73 | 82.48 |
| | | M3ID | 76.63 | 71.79 | 87.73 | 78.97 | 75.40 | 71.26 | 85.13 | 77.58 | 83.03 | 91.12 | 73.20 | 81.18 |
| | | M3ID + CNS | 77.60 | 72.85 | 88.00 | 79.71 | 75.83 | 71.64 | 85.53 | 77.97 | 84.37 | 91.94 | 75.33 | 82.81 |
| | | ONLY | 79.43 | 75.07 | 88.13 | 81.08 | 75.63 | 71.28 | 85.87 | 77.90 | 83.77 | 85.10 | 81.87 | 83.45 |
| | | ONLY + CNS | 79.83 | 75.27 | 88.87 | 81.50 | 75.97 | 71.46 | 86.47 | 78.25 | 84.60 | 85.65 | 83.13 | 84.37 |
| A-OKVQA | Random | Regular | 82.17 | 76.47 | 92.93 | 83.90 | 81.60 | 77.40 | 89.27 | 82.91 | 86.13 | 94.72 | 76.53 | 84.66 |
| | | Regular + CNS | 82.43 | 76.57 | 93.47 | 84.18 | 82.97 | 78.87 | 90.07 | 84.10 | 86.97 | 95.04 | 78.00 | 85.68 |
| | | VCD | 81.70 | 75.55 | 93.73 | 83.67 | 82.53 | 78.91 | 88.80 | 83.56 | 87.80 | 94.09 | 80.67 | 86.86 |
| | | VCD + CNS | 82.83 | 76.90 | 93.87 | 84.54 | 82.90 | 79.05 | 89.53 | 83.96 | 88.40 | 94.31 | 81.73 | 87.57 |
| | | M3ID | 82.90 | 77.13 | 93.53 | 84.54 | 83.17 | 79.04 | 90.27 | 84.28 | 87.47 | 94.89 | 79.20 | 86.34 |
| | | M3ID + CNS | 83.90 | 78.33 | 93.73 | 85.34 | 83.93 | 80.05 | 90.40 | 84.91 | 88.27 | 94.98 | 80.80 | 87.32 |
| | | ONLY | 86.33 | 81.21 | 94.53 | 87.37 | 85.40 | 81.27 | 92.00 | 86.30 | 89.77 | 92.04 | 87.07 | 89.48 |
| | | ONLY + CNS | 86.57 | 81.50 | 94.60 | 87.57 | 86.40 | 82.42 | 92.53 | 87.19 | 90.30 | 92.30 | 87.93 | 90.06 |
| | Popular | Regular | 75.40 | 68.81 | 92.93 | 79.07 | 74.80 | 69.23 | 89.27 | 77.98 | 86.17 | 93.71 | 77.53 | 84.86 |
| | | Regular + CNS | 77.00 | 70.43 | 93.07 | 80.18 | 75.57 | 69.92 | 89.73 | 78.60 | 86.93 | 94.25 | 78.67 | 85.76 |
| | | VCD | 74.83 | 68.02 | 93.73 | 78.83 | 76.33 | 71.08 | 88.80 | 78.96 | 87.03 | 92.24 | 80.87 | 86.18 |
| | | VCD + CNS | 75.60 | 68.66 | 94.20 | 79.43 | 77.33 | 71.93 | 89.67 | 79.82 | 87.43 | 92.70 | 81.27 | 86.61 |
| | | M3ID | 75.93 | 69.18 | 93.53 | 79.54 | 76.27 | 70.52 | 90.27 | 79.18 | 86.83 | 94.31 | 78.40 | 85.62 |
| | | M3ID + CNS | 76.53 | 69.78 | 93.60 | 79.95 | 77.47 | 71.64 | 90.93 | 80.14 | 87.50 | 94.68 | 79.47 | 86.41 |
| | | ONLY | 79.73 | 72.94 | 94.53 | 82.35 | 77.07 | 70.84 | 90.80 | 80.05 | 89.30 | 90.46 | 87.87 | 89.14 |
| | | ONLY + CNS | 80.23 | 73.46 | 94.67 | 82.73 | 77.93 | 71.75 | 92.13 | 80.68 | 90.03 | 91.21 | 88.60 | 89.89 |
| | Adversarial | Regular | 67.07 | 61.33 | 92.40 | 73.72 | 68.30 | 62.83 | 89.60 | 73.87 | 81.17 | 83.18 | 78.13 | 80.58 |
| | | Regular + CNS | 67.93 | 61.98 | 92.80 | 74.32 | 68.97 | 63.40 | 89.73 | 74.30 | 81.77 | 83.67 | 78.93 | 81.23 |
| | | VCD | 67.13 | 61.27 | 93.13 | 73.92 | 70.07 | 64.70 | 88.33 | 74.69 | 81.37 | 81.86 | 80.60 | 81.22 |
| | | VCD + CNS | 69.00 | 62.78 | 93.33 | 75.07 | 71.50 | 65.97 | 88.80 | 75.70 | 82.03 | 82.93 | 80.67 | 81.78 |
| | | M3ID | 67.20 | 61.31 | 93.27 | 73.98 | 68.67 | 63.05 | 90.20 | 74.22 | 81.30 | 83.42 | 78.13 | 80.69 |
| | | M3ID + CNS | 68.97 | 62.70 | 93.67 | 75.11 | 70.70 | 64.75 | 90.87 | 75.62 | 82.37 | 84.07 | 79.87 | 81.91 |
| | | ONLY | 69.10 | 62.67 | 94.47 | 75.35 | 68.17 | 62.67 | 91.33 | 74.15 | 82.17 | 78.87 | 87.87 | 83.13 |
| | | ONLY + CNS | 70.53 | 63.85 | 94.67 | 76.26 | 69.80 | 63.79 | 91.60 | 75.21 | 82.97 | 79.52 | 88.80 | 83.91 |
| GQA | Random | Regular | 82.00 | 76.09 | 93.33 | 83.83 | 80.10 | 76.17 | 87.60 | 81.49 | 84.47 | 89.83 | 77.73 | 83.35 |
| | | Regular + CNS | 82.53 | 76.61 | 93.67 | 84.28 | 80.80 | 76.77 | 88.33 | 82.15 | 85.50 | 90.62 | 79.20 | 84.53 |
| | | VCD | 81.90 | 75.38 | 94.73 | 83.96 | 81.10 | 77.92 | 86.80 | 82.12 | 86.10 | 90.87 | 80.27 | 85.24 |
| | | VCD + CNS | 82.33 | 75.74 | 95.13 | 84.34 | 81.97 | 78.70 | 87.67 | 82.94 | 86.73 | 91.00 | 81.53 | 86.01 |
| | | M3ID | 83.10 | 77.00 | 94.40 | 84.82 | 81.63 | 77.80 | 88.53 | 82.82 | 86.13 | 91.63 | 79.53 | 85.15 |
| | | M3ID + CNS | 83.73 | 77.71 | 94.60 | 85.33 | 82.67 | 79.11 | 88.67 | 83.65 | 87.07 | 92.31 | 80.87 | 86.21 |
| | | ONLY | 86.87 | 81.24 | 95.87 | 87.95 | 83.23 | 79.69 | 89.20 | 84.18 | 88.13 | 90.50 | 86.40 | 87.92 |
| | | ONLY + CNS | 87.77 | 82.35 | 96.13 | 88.71 | 83.50 | 79.89 | 89.53 | 84.44 | 88.53 | 90.19 | 86.47 | 88.29 |
| | Popular | Regular | 71.93 | 65.36 | 93.33 | 76.88 | 72.20 | 66.97 | 87.60 | 75.91 | 80.40 | 82.43 | 77.27 | 79.77 |
| | | Regular + CNS | 72.73 | 66.10 | 93.33 | 77.39 | 72.50 | 67.16 | 88.07 | 76.20 | 81.93 | 83.88 | 79.07 | 81.40 |
| | | VCD | 70.67 | 63.95 | 94.73 | 76.36 | 73.20 | 68.24 | 86.80 | 76.41 | 80.27 | 80.68 | 79.60 | 80.13 |
| | | VCD + CNS | 72.53 | 65.52 | 95.13 | 77.60 | 73.70 | 68.70 | 87.07 | 76.80 | 81.43 | 81.79 | 80.87 | 81.33 |
| | | M3ID | 72.10 | 65.28 | 94.40 | 77.19 | 73.40 | 67.96 | 88.53 | 76.90 | 81.87 | 84.14 | 78.53 | 81.24 |
| | | M3ID + CNS | 73.80 | 66.86 | 94.40 | 78.28 | 75.23 | 69.83 | 88.87 | 78.20 | 82.77 | 84.83 | 79.80 | 82.24 |
| | | ONLY | 74.93 | 67.58 | 95.87 | 79.27 | 74.13 | 68.55 | 89.20 | 77.52 | 81.27 | 78.67 | 85.80 | 82.08 |
| | | ONLY + CNS | 76.03 | 68.59 | 96.07 | 80.03 | 75.67 | 70.16 | 89.33 | 78.59 | 81.50 | 78.97 | 85.87 | 82.27 |
| | Adversarial | Regular | 67.93 | 61.96 | 92.93 | 74.35 | 68.43 | 63.48 | 86.80 | 73.33 | 78.87 | 79.30 | 78.13 | 78.71 |
| | | Regular + CNS | 69.90 | 63.54 | 93.40 | 75.63 | 70.00 | 64.75 | 87.80 | 74.53 | 80.00 | 80.20 | 79.67 | 79.93 |
| | | VCD | 68.23 | 61.90 | 94.87 | 74.91 | 69.27 | 64.38 | 86.27 | 73.73 | 80.53 | 80.57 | 80.47 | 80.52 |
| | | VCD + CNS | 69.37 | 62.83 | 94.87 | 75.59 | 70.47 | 65.49 | 86.53 | 74.55 | 81.97 | 81.99 | 81.93 | 81.96 |
| | | M3ID | 68.60 | 62.23 | 94.67 | 75.09 | 69.23 | 64.09 | 87.47 | 73.98 | 81.00 | 81.63 | 80.00 | 80.81 |
| | | M3ID + CNS | 68.73 | 62.32 | 94.73 | 75.19 | 70.87 | 65.63 | 87.60 | 75.04 | 81.00 | 81.63 | 80.00 | 80.81 |
| | | ONLY | 69.33 | 62.66 | 95.67 | 75.73 | 69.30 | 63.82 | 89.13 | 74.38 | 80.47 | 77.23 | 86.40 | 81.56 |
| | | ONLY + CNS | 70.17 | 63.33 | 95.80 | 76.25 | 71.30 | 65.59 | 89.60 | 75.74 | 81.50 | 78.58 | 86.60 | 82.40 |

## 6.3 ABLATION STUDIES

Here we focus on the core *CNS* module, with additional results in Appendix E.

Table 3: Results on CHAIR. ↓ indicates lower is better.

| Method | LLaVA-1.5 | | | | InstructBLIP | | | | Qwen-VL | | | |
|---|---|---|---|---|---|---|---|---|---|---|---|---|
| | Max Token 64 | | Max Token 128 | | Max Token 64 | | Max Token 128 | | Max Token 64 | | Max Token 128 | |
| | CHAIR$_S$ ↓ | CHAIR$_I$ ↓ | CHAIR$_S$ ↓ | CHAIR$_I$ ↓ | CHAIR$_S$ ↓ | CHAIR$_I$ ↓ | CHAIR$_S$ ↓ | CHAIR$_I$ ↓ | CHAIR$_S$ ↓ | CHAIR$_I$ ↓ | CHAIR$_S$ ↓ | CHAIR$_I$ ↓ |
| Regular | 26.5 | 9.4 | 55.1 | 16.4 | 31.5 | 11.4 | 57.4 | 17.6 | 33.8 | 12.9 | 52.1 | 16.7 |
| Regular + CNS | 25.7 | 8.8 | 54.8 | 16.0 | 30.9 | 11.1 | 57.2 | 16.8 | 33.6 | 12.7 | 51.6 | 16.0 |
| VCD | 24.8 | 8.0 | 54.4 | 16.6 | 30.0 | 10.1 | 60.7 | 18.0 | 33.3 | 13.1 | 50.4 | 17.2 |
| VCD + CNS | 24.3 | 7.6 | 54.3 | 16.3 | 30.0 | 10.1 | 60.6 | 17.7 | 33.1 | 12.3 | 49.8 | 16.6 |
| M3ID | 21.4 | 6.4 | 56.6 | 15.8 | 31.1 | 10.5 | 62.3 | 18.2 | 32.3 | 11.9 | 49.8 | 17.4 |
| M3ID + CNS | 20.7 | 6.2 | 55.9 | 15.3 | 31.1 | 10.2 | 61.9 | 17.9 | 32.2 | 11.7 | 49.2 | 16.6 |
| ONLY | 20.1 | 6.3 | 49.9 | 14.7 | 23.9 | 8.3 | 52.5 | 15.7 | 27.7 | 8.6 | 48.1 | 14.4 |
| ONLY + CNS | **19.7** | **5.7** | **49.6** | **14.6** | **23.4** | **7.7** | **52.4** | **15.3** | **27.0** | **7.8** | **47.4** | **14.1** |

Table 4: Results on MME-Hallucination.

| Method | Object-level | | Attribute-level | | MME Score ↑ |
|---|---|---|---|---|---|
| | Existence ↑ | Count ↑ | Position ↑ | Color ↑ | |
| Regular | 185.00 | 126.67 | 128.33 | 148.33 | 588.33 |
| Regular + CNS | 187.00 | 127.33 | 129.67 | 149.33 | 593.33 |
| VCD | 185.00 | 136.67 | 128.33 | 158.33 | 608.33 |
| VCD + CNS | 186.00 | 138.67 | 131.67 | 159.33 | 615.67 |
| M3ID | 190.00 | 136.67 | 128.33 | 158.33 | 613.33 |
| M3ID + CNS | 191.00 | 138.33 | 129.67 | 159.67 | 618.67 |
| ONLY | 190.00 | 143.33 | 133.33 | 148.33 | 614.99 |
| ONLY + CNS | **191.00** | **144.33** | **134.67** | 149.67 | **619.67** |

Table 5: Ablations studies on CNS components.

| Strategy | POPE ↑ | | | | CHAIR ↓ | |
|---|---|---|---|---|---|---|
| | Acc. | Prec. | Rec. | F1 | CHAIR$_S$ | CHAIR$_I$ |
| Regular | 84.63 | 83.07 | 87.00 | 84.99 | 26.5 | 9.4 |
| +SAE | 84.63 | 83.07 | 87.00 | 84.99 | 26.5 | 9.4 |
| +CNR | 84.87 | 83.45 | 87.01 | 85.19 | 26.2 | 9.2 |
| +ANC | 85.00 | 83.66 | 87.01 | 85.30 | 25.9 | 8.9 |
| +ANS(full CNS) | **85.10** | **83.77** | **87.07** | **85.39** | 25.7 | **8.8** |
| +Zeroing #3833 | 85.03 | 83.73 | 87.02 | 85.34 | 25.9 | 8.9 |

**CNS Components.** We progressively add components to the baseline LVLM: (i) **Baseline**, the unmodified model; (ii) **+SAE**, using only SAE-reconstructed features; (iii) **+ANC**, adding adaptive neuron constraints; (iv) **+ANR**, further applying adaptive neuron regulation; (v) **+ANS**, incorporating always-on neuron suppression, which constitutes the full CNS; (vi) **+ Zeroing #3833**, setting neuron #3833 to zero. As shown in Tab. 5, +SAE preserves baseline performance, confirming faithful reconstruction. Adding ANC, ANR, and ANS yields incremental gains. However, explicitly suppressing neuron #3833 leads to a slight performance drop, indicating that only neurons lacking fine-grained semantic meaning should be removed. The full configuration achieves the strongest hallucination mitigation, demonstrating the effectiveness of our design.

**For more analysis, experiments, and visualizations, please refer to the appendix.**

# 7 CONCLUSION

In conclusion, we present a representation-level approach to understanding and mitigating hallucinations in LVLMs. We show how image noise perturbs internal monosemantic neurons, reshapes visual semantics, and exacerbates hallucinations. Our **CNS** amplifies meaningful neurons while suppressing perturbation-induced activations for hallucination mitigation. Furthermore, it enables interpretable and controllable neuron-level interventions, providing both practical benefits and deeper mechanistic insights into LVLMs' internal visual representations and hallucinations.

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

## A    ADDITIONAL NEURON VISUALIZATIONS

Fig. 7 presents additional examples of neurons discovered by our sparse autoencoder (SAE). Many neurons exhibit strong associations with concrete objects or concepts, such as #14174 for corn, #46469 for oranges, and #61697 for dogs wearing Christmas hats. Beyond object-level semantics, some neurons capture more abstract structural cues, such as #62747, which consistently responds to spiral or fan-shaped patterns. These examples demonstrate the richness and diversity of the learned neuron space, ranging from fine-grained objects to higher-level structural abstractions. Such diversity not only enhances the interpretability of internal visual representations but also provides a strong foundation for precise neuron-level interventions, thereby facilitating both mechanistic understanding and controllable steering of LVLM outputs.

## B    NOISE-INDUCED DISRUPTION OF INTERNAL VISUAL FEATURES LEADING TO HALLUCINATIONS

In Sec. 4, we quantitatively analyzed how noise perturbs internal visual features, causing neuron activations to shift and destabilize. These disruptions reshape the semantics of visual representations, inducing hallucinations and degrading LVLM performance (see Fig. 5).

To illustrate this phenomenon more intuitively, Fig. 8 shows an example image containing a camera. As increasing levels of noise are applied, the activation of the "camera" neuron gradually diminishes. Correspondingly, the LVLM output exhibits a progressive semantic drift: initially describing a "black Konica Minolta camera with a large lens," then simplifying to "camera with a large lens," and eventually omitting the camera entirely. This case demonstrates how noise-induced disruptions at the neuron level directly erode semantic fidelity in visual features, ultimately manifesting as hallucinations in model outputs.

Importantly, this example underscores the value of SAEs: by decomposing dense visual embeddings into sparse, monosemantic neurons, we gain the ability to trace how specific semantic concepts evolve under perturbations. This neuron-level perspective provides interpretability and analytical clarity, enabling us to pinpoint which neurons are destabilized and how this relates to output degradation. Such insights establish a principled foundation for designing targeted interventions to mitigate hallucinations and improve LVLM reliability.

## C    HIGH-FREQUENCY NEURON ANALYSIS AND VISUALIZATION

To gain deeper insights into the functional roles of individual neurons in LVLM visual representations, we perform both image-level and patch-level analyses. These qualitative results complement the quantitative findings in Sec. 4, providing a more intuitive understanding of how neurons encode semantic information.

**Image-Level Analysis and Visualizations.**    Fig. 9 highlights neurons with consistently high activation across different images. We observe that a small subset of "always-on" neurons remain persistently active regardless of image content, often encoding recurring textures or repetitive small objects rather than scene-specific information. The bottom panel further visualizes the top-activated images for each neuron, confirming that these neurons capture similar global patterns across diverse inputs. Within CNS, we reduce their disproportionate influence via Always-on Neuron Suppression (ANS), which decreases redundancy, emphasizes image-specific content, and improves the interpretability of downstream neuron-level interventions.

**Patch-Level Analysis and Visualizations.**    Fig. 10 illustrates neuron activations at the patch level. Unlike always-on neurons, most patch-level neurons respond reliably to localized, semantically meaningful concepts, such as distinctive textures or object parts. This indicates that individual neurons often encode interpretable, fine-grained features, which are particularly well-suited for targeted interventions. By selectively modulating these neurons, we can directly influence which visual concepts are emphasized or suppressed in LVLM outputs, enabling fine-grained and interpretable control.

**Summary.** Together, the image-level and patch-level analyses reveal a dual organization of neuron activations: broadly active global features and selectively tuned local features. This dual perspective underpins our CNS approach, where targeted neuron-level interventions enable controllable and interpretable mitigation of hallucinations, while also deepening mechanistic insights into LVLM visual processing.

## D    MORE COMPLEX AND DIVERSE CASES FOR STEERING LVLMS

We further present diverse and challenging case studies to illustrate how neuron-level steering enables fine-grained and interpretable control in LVLMs. These examples highlight not only the feasibility of manipulating specific concepts but also the varying levels of difficulty imposed by scene complexity and semantic distribution across neurons.

**Multi-Concept Suppression.** Fig. 11 shows a scene containing multiple objects (dog and chair). By suppressing the neurons corresponding to each object, we can selectively remove them from generated captions or descriptions. Interestingly, the difficulty of suppression varies across concepts. Suppressing "chair" neurons is relatively straightforward, with a weight of $\alpha = -30$ sufficient to eliminate chairs from outputs. In contrast, suppressing "dog" neurons requires much stronger intervention, sometimes leaving residual references until $\alpha = -100$ fully removes them. Closer inspection reveals that the SAE encodes a hierarchy of dog-related concepts (e.g., different breeds), making suppression more challenging when the concept is distributed across multiple fine-grained neurons.

**Concept Insertion in Simple and Complex Contexts.** We also examine concept insertion across scenes of different complexity. As shown in Fig. 12, inserting a dog concept into a simple bird-dominated scene requires only a modest weight ($\alpha = 50$) for the concept to appear in outputs. However, inserting the same concept into a more complex scene containing multiple objects demands a much larger weight ($\alpha = 500$) to manifest reliably. This contrast underscores how scene complexity significantly affects the intervention strength required for successful concept insertion.

**Multi-Neuron Steering in Complex Contexts.** In highly complex visual scenes, steering a single dog-related neuron requires very large weights (e.g., $\alpha = 500$) before the concept emerges in outputs (Fig. 13). By contrast, jointly adjusting three dog-related neurons with smaller weights ($\alpha = 20$ each) produces a more natural and robust insertion. A similar pattern is observed for suppression (Fig. 14): targeting a single neuron requires extreme negative weights (e.g., $\alpha = -100$), whereas coordinated modulation of three neurons with smaller magnitudes ($\alpha = -10$ each) removes the concept more effectively and smoothly. These findings highlight the superior stability and efficiency of multi-neuron steering, providing strong support for our CNS design, which automatically selects and adjusts multiple neurons to achieve reliable fine-grained control.

**Summary.** Across multi-concept suppression, concept insertion, and multi-neuron steering, we find that the effectiveness of interventions depends critically on both the semantic distribution of concepts across neurons and the contextual complexity of the scene. These case studies collectively validate neuron-level steering as a powerful and interpretable mechanism for controlling LVLM behavior in diverse scenarios.

## E    ADDITIONAL ABLATION STUDIES

We conduct ablation studies to examine the influence of key hyperparameters and neuron-level components in **Contrastive Neuron Steering (CNS)**. All experiments are performed on the POPE and CHAIR benchmarks using LLaVA-1.5.

### E.1    ABLATION ON CONTRASTIVE REGULARIZATION WEIGHT $\lambda$

To evaluate how the strength of contrastive neuron steering affects hallucination suppression and content preservation, we vary the weight $\lambda$ of Contrastive neuron regulation (CNR) over 0.25, 0.5, 0.75, 1.0, 2.0, keeping other parameters at default. As shown in Tab. 6, a moderate value of $\lambda = 0.5$ achieves the best trade-off, demonstrating the effectiveness of CNS in balancing hallucination reduction and performance retention.

Table 6: Ablation study on the contrastive regularization weight $\lambda$.

| $\lambda$ | POPE ↑ | | | | CHAIR ↓ | |
|---|---|---|---|---|---|---|
| | Acc. | Prec. | Rec. | F1 | CHAIR$_S$ | CHAIR$_I$ |
| 0.25 | 85.17 | 83.78 | 87.19 | 85.45 | 25.8 | 8.9 |
| 0.5 | 85.20 | 83.85 | 87.20 | 85.49 | 25.7 | 8.8 |
| 0.75 | 85.13 | 83.71 | 87.17 | 85.41 | 25.8 | 8.9 |
| 1.0 | 84.87 | 83.38 | 87.25 | 85.27 | 26.1 | 9.1 |
| 2.0 | 84.77 | 83.14 | 87.29 | 85.17 | 26.2 | 9.2 |

### E.2 ABLATION ON PLAUSIBILITY THRESHOLD $\tau$

We vary the adaptive neuron magnitude threshold $\tau$ of Adaptive neuron constraints (ANC) over 0.001, 0.01, 0.1, 0.2, 0.3 to examine its impact on CNS performance. Other parameters remain at default. Tab. 7 shows that $\tau = 0.1$ achieves the best balance, effectively filtering weak neuron signals while retaining salient visual features, confirming the robustness of our method.

Table 7: Ablation study on the plausibility threshold $\tau$.

| $\lambda$ | POPE ↑ | | | | CHAIR ↓ | |
|---|---|---|---|---|---|---|
| | Acc. | Prec. | Rec. | F1 | CHAIR$_S$ | CHAIR$_I$ |
| 0.001 | 85.13 | 83.74 | 87.10 | 85.39 | 25.9 | 9.1 |
| 0.01 | 85.20 | 83.82 | 87.17 | 85.46 | 25.8 | 8.9 |
| 0.1 | 85.20 | 83.85 | 87.20 | 85.49 | 25.7 | 8.8 |
| 0.2 | 85.17 | 83.83 | 87.12 | 85.44 | 25.8 | 8.9 |
| 0.3 | 85.10 | 83.68 | 87.09 | 85.35 | 25.9 | 9.1 |

### E.3 ABLATION ON TOP-$k$ NEURONS FOR ANS

To evaluate the effect of the number of top-$k$ neurons in Always-on Neuron Suppression (ANS), we vary $k$ over 10, 20, 30, 40. Tab. 8 shows that $k = 20$ provides the optimal trade-off, effectively suppressing generic neurons without removing informative, image-specific signals.

Table 8: Ablation study on the number of top-$k$ neurons for ANS.

| Top-$k$ | Acc. | Prec. | Rec. | F1 | CHAIR$_S$ | CHAIR$_I$ |
|---|---|---|---|---|---|---|
| 10.0 | 85.20 | 83.84 | 87.19 | 85.48 | 25.8 | 8.9 |
| 20.0 | 85.20 | 83.85 | 87.20 | 85.49 | 25.7 | 8.8 |
| 30.0 | 85.17 | 83.83 | 87.12 | 85.44 | 25.8 | 8.9 |
| 40.0 | 85.16 | 83.82 | 87.06 | 85.41 | 25.9 | 9.1 |

### E.4 ABLATION ON NOISE STEPS IN ADAPTIVE CONSTRAINTS

We investigate how the number of denoising steps in the adaptive plausibility constraint affects neuron stability and hallucination reduction, testing steps from 0 to 999. Tab. 9 shows that increasing the number of steps improves stability and reduces hallucinations, with 500 steps achieving the best performance while balancing computational cost.

Table 9: Effect of denoising steps in adaptive plausibility constraints.

| Steps | Acc. | Prec. | Rec. | F1 | CHAIR$_S$ | CHAIR$_I$ |
|---|---|---|---|---|---|---|
| 0 | 85.00 | 83.61 | 87.07 | 85.30 | 26.5 | 9.4 |
| 100 | 85.17 | 83.97 | 86.93 | 85.42 | 26.4 | 9.2 |
| 200 | 85.10 | 83.86 | 86.93 | 85.37 | 26.2 | 9.3 |
| 300 | 85.13 | 83.87 | 87.00 | 85.41 | 26.1 | 9.1 |
| 400 | 85.20 | 83.89 | 87.13 | 85.48 | 26.0 | 8.9 |
| 500 | 85.20 | 83.85 | 87.20 | 85.49 | 25.7 | 8.8 |
| 600 | 85.20 | 83.63 | 87.53 | 85.54 | 25.9 | 8.9 |
| 700 | 85.10 | 83.51 | 87.47 | 85.44 | 26.2 | 9.1 |
| 800 | 85.07 | 83.42 | 87.53 | 85.43 | 26.3 | 9.2 |
| 900 | 84.73 | 83.02 | 87.33 | 85.12 | 26.6 | 9.4 |
| 999 | 84.93 | 83.25 | 87.47 | 85.31 | 26.9 | 9.6 |

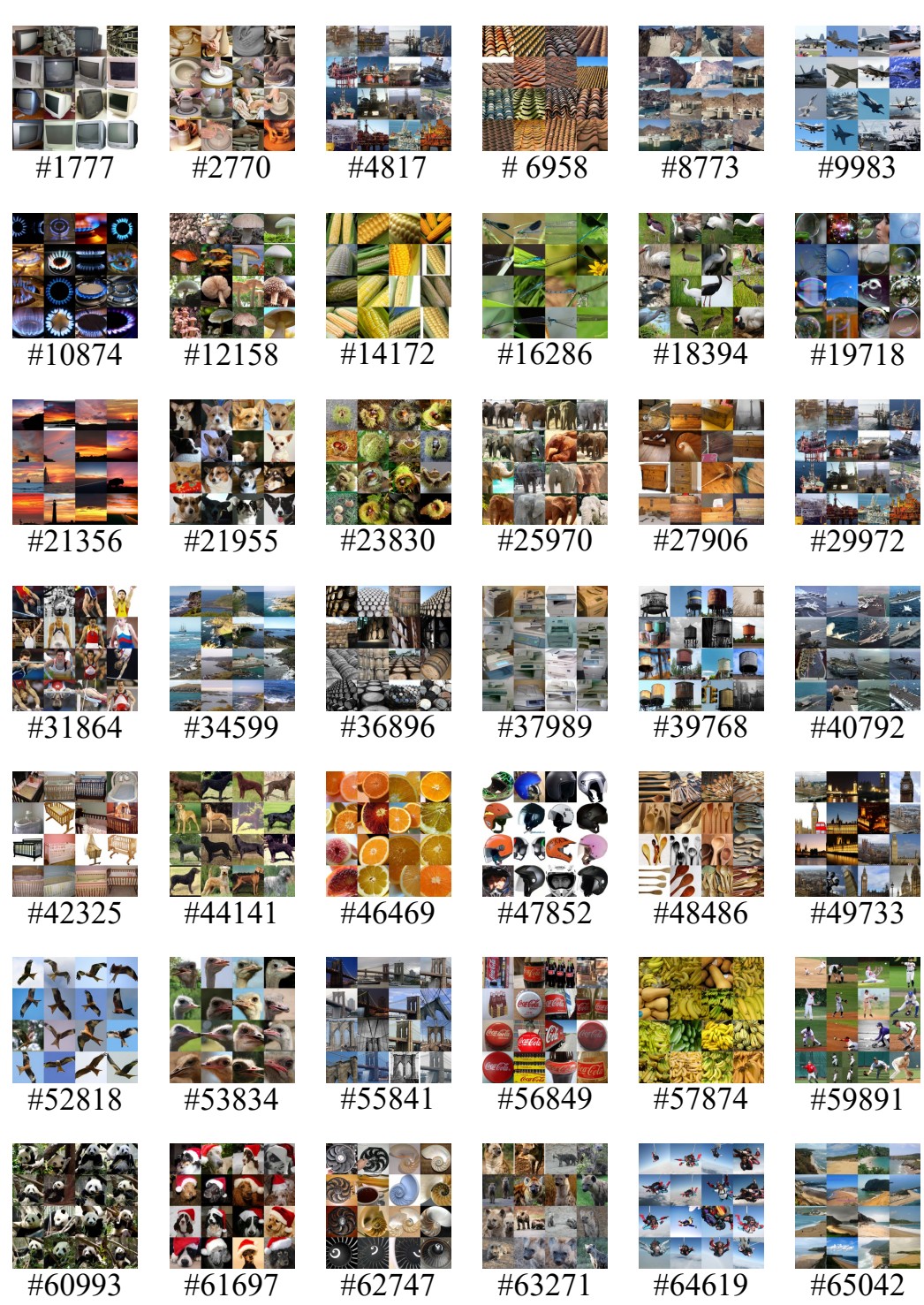

Figure 7: Additional visualizations of monosemantic neurons learned by the SAE. The neurons exhibit diverse semantics, ranging from specific objects (e.g., corn #14174, oranges #46469, Christmas-hat dogs #61697) to abstract structural patterns (e.g., spirals or fan-like shapes #62747). This diversity demonstrates the interpretability of the internal representation space and provides a strong foundation for explaining and steering LVLMs through neuron-level interventions.

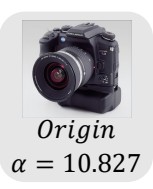 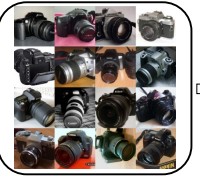

*Origin*
α = 10.827

The image features a **black Konica Minolta camera with a large lens**, sitting on a white background. The camera is positioned in the center of the frame, and the lens is prominently displayed. The camera appears to be a professional model, possibly a DSLR…

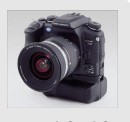

α = 10.607
t = 100

The image features a **black Konica Minolta camera with a large lens**, sitting on a white background. The camera… possibly a Dynax 7 …

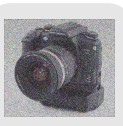

α = 8.531
t = 600

The image features a close-up of a **camera lens**, showcasing the front of the camera. The lens is the main focus of the image, with the camera's body not visible.

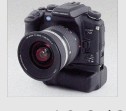

α = 10.352
t = 200

The image features a **black Konica Minolta camera with a large lens**, sitting on a white surface. The camera… possibly a DSLR…

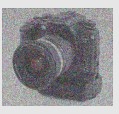

α = 7.217
t = 700

The image features a close-up of a **camera lens**, showcasing the lens and its surroundings. …The camera lens appears to be a part of a larger camera, possibly …

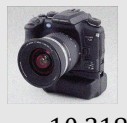

α = 10.318
t = 300

The image features a **black Konica Minolta camera with a large lens**, sitting on a white surface. The camera… possibly a DSLR…

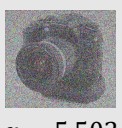

α = 5.502
t = 800

The image features a close-up of a **camera lens**, showcasing the intricate details of the lens. The lens is positioned in the center of the image…

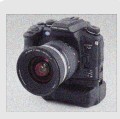

α = 9.968
t = 400

The image features a close-up of a black digital **camera with a large lens**. The camera… The camera appears to be a Nikon model, and it is set to take a picture.

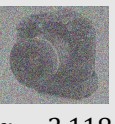

α = 2.118
t = 900

The image features a close-up of a colorful, patterned object, possibly a piece of art or a decorative item. The object is surrounded by a variety of colors and patterns…

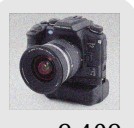

α = 9.409
t = 500

The image features a close-up of a black digital **camera with a large lens**. The camera… The camera appears to be a Nikon model, and it is set to take a picture.

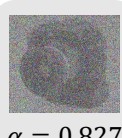

α = 0.827
t = 999

The image features a close-up of a colorful, circular object with a patterned surface. The object appears to be a piece of art or a decorative item…

Figure 8: Example of noise affecting visual feature representations. The image contains a camera. As noise increases, the activation of the "camera" neuron gradually decreases, and the LVLM output progressively loses detail: from "black Konica Minolta camera with a large lens" to "camera with a large lens," and finally no camera description. This demonstrates how noise disrupts internal semantic representations, leading to hallucinations. It also highlights the advantage of SAEs in decoupling dense LVLM features into sparse, monosemantic neurons, allowing us to track and analyze internal visual feature changes at the neuron level.

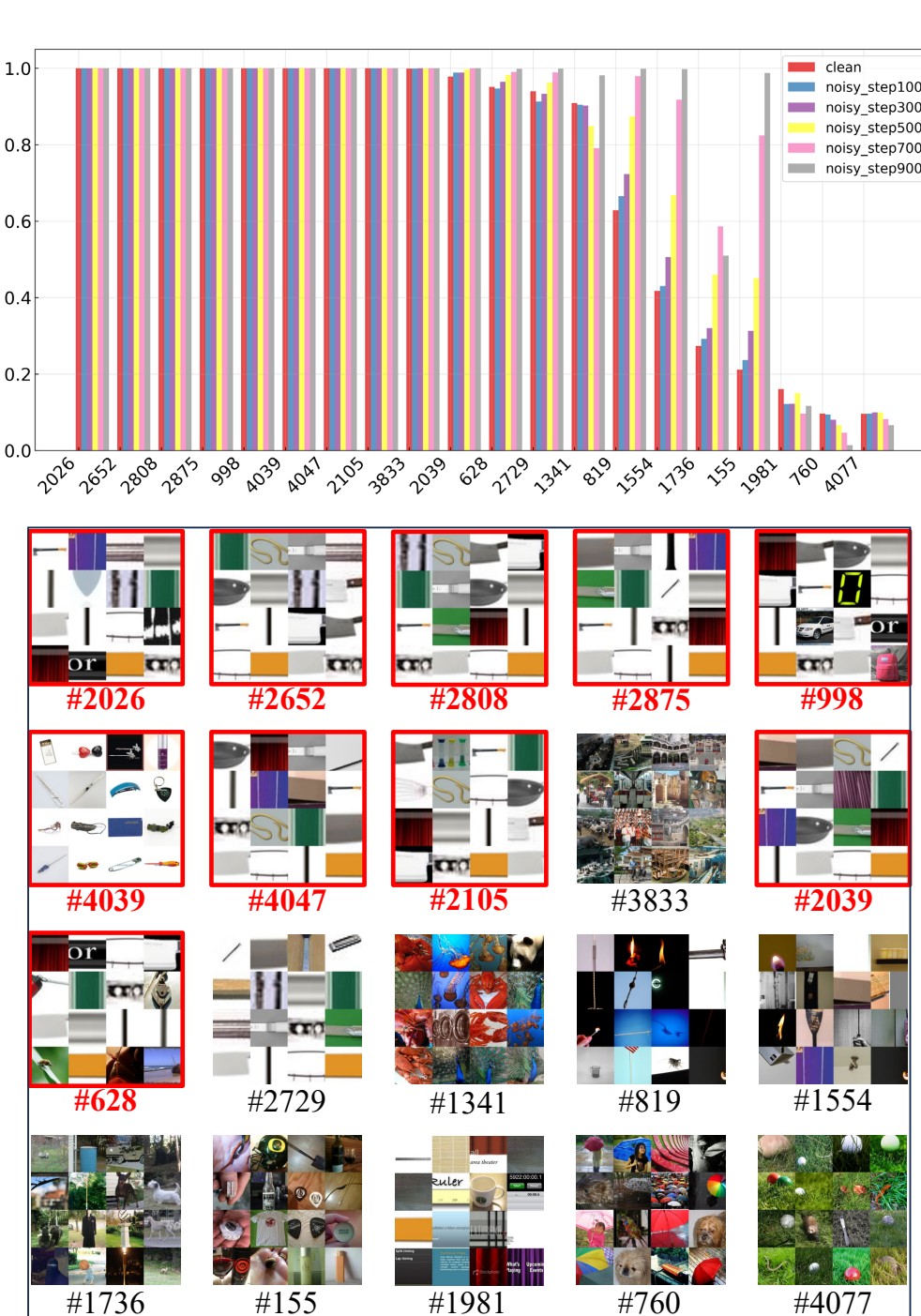

Figure 9: Image-Level Dominance Analysis and Visualization. The top panel shows neurons with consistently high activation rates across different images. The bottom panel visualizes the top-activated images for each neuron. These "always-on" neurons often correspond to recurring textures or small objects and represent similar global information. Red highlights indicate neurons selected for suppression in CNS via ANS. Suppressing these neurons emphasizes image-specific objects, providing an interpretable basis for our method.

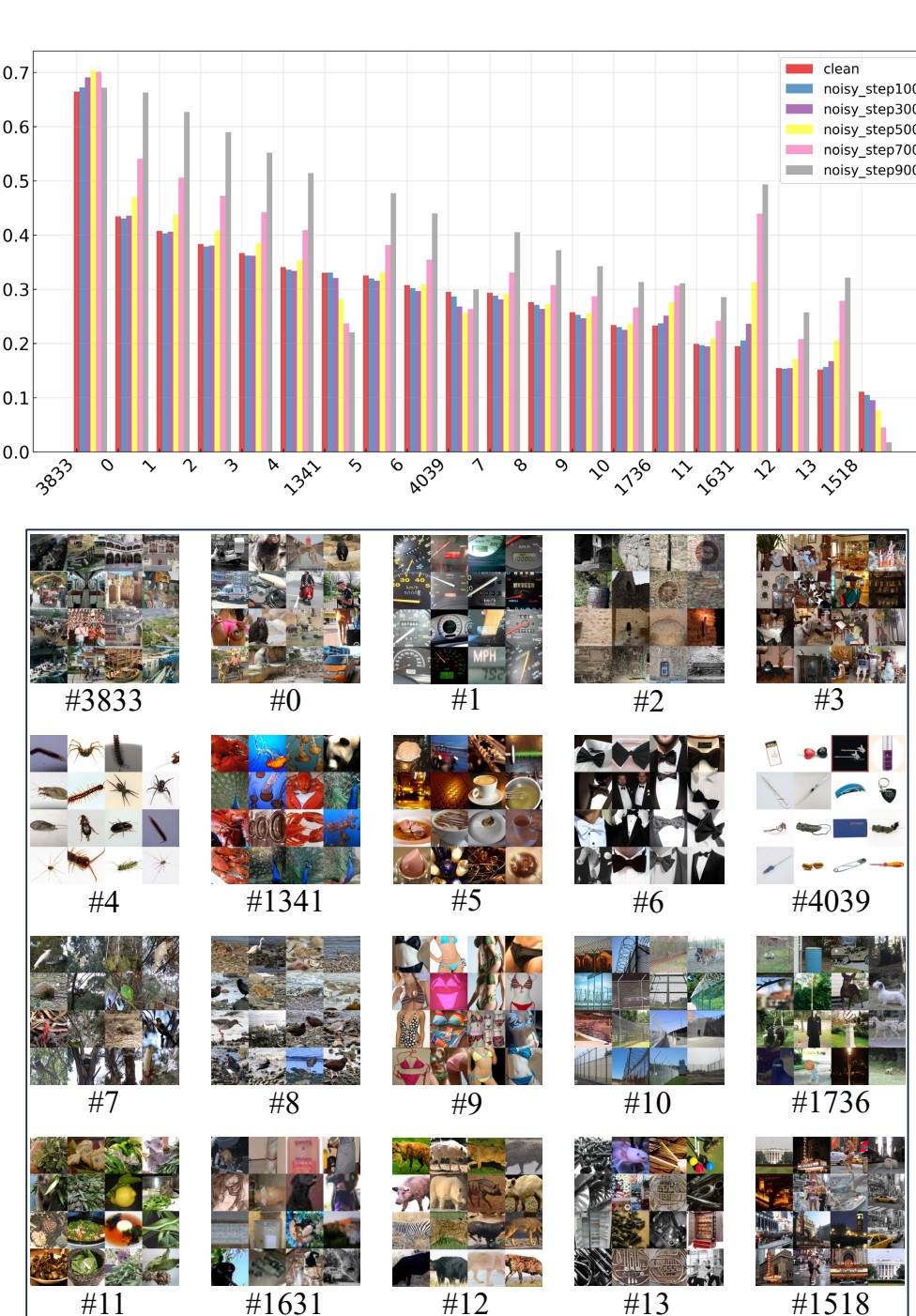

Figure 10: Patch-Level Activation Analysis and Visualization. At the patch level, neurons often capture concrete, localized concepts. Activation patterns show that most neurons reliably represent specific visual features, supporting fine-grained neuron-level interventions.

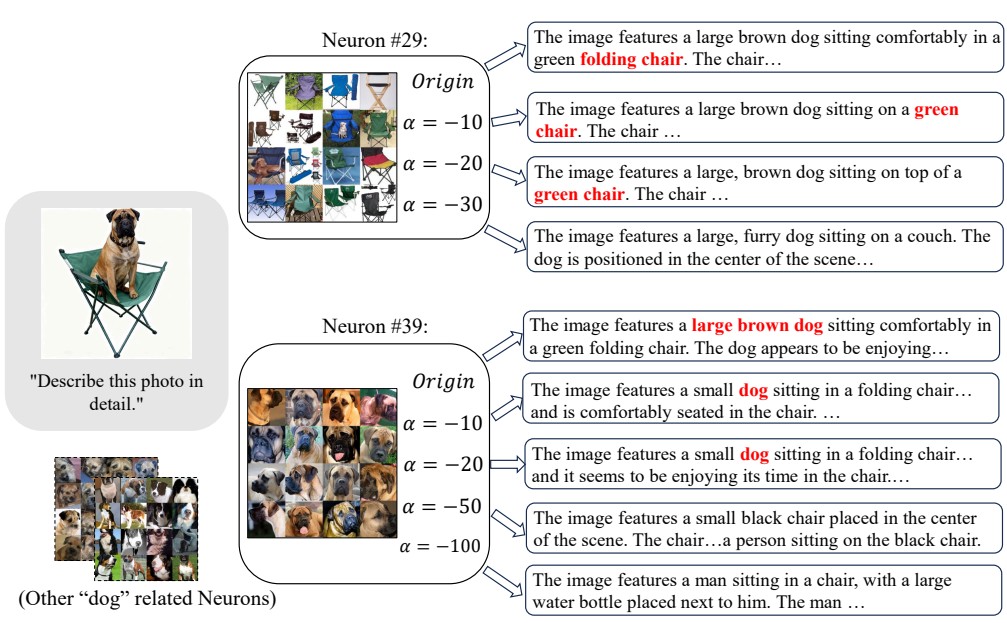

Figure 11: Multi-concept suppression. Suppressing "chair" neurons effectively removes chairs from model outputs. Suppressing "dog" neurons is more challenging, requiring stronger intervention since the SAE has learned a hierarchy of dog-related concepts (e.g., different breeds). This highlights the difficulty of eliminating concepts encoded in multiple fine-grained neurons.

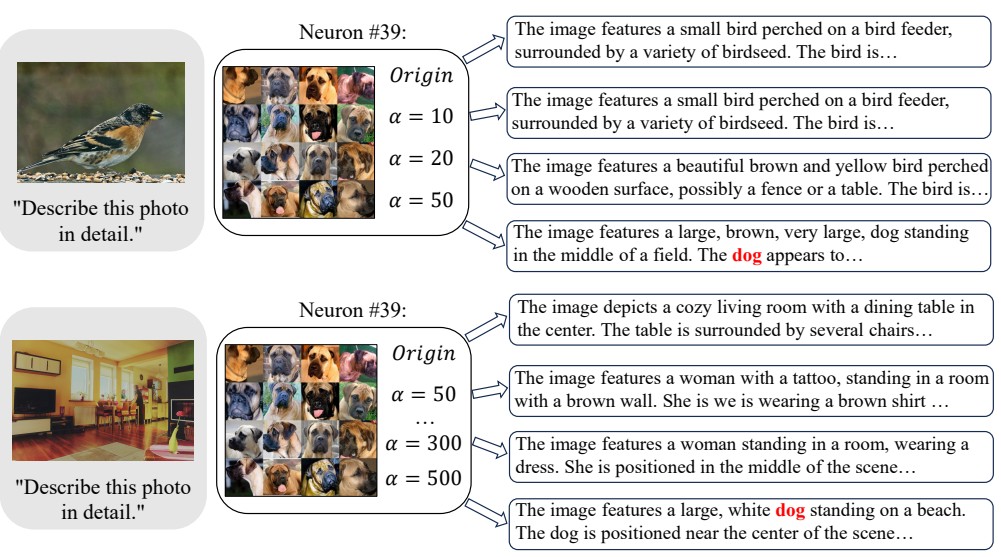

Figure 12: Concept insertion in simple contexts. By slightly amplifying a single dog-related neuron, the model begins to hallucinate the presence of dogs in unrelated scenes. Compared to suppression, concept insertion is easier: small weights suffice to introduce the new concept.

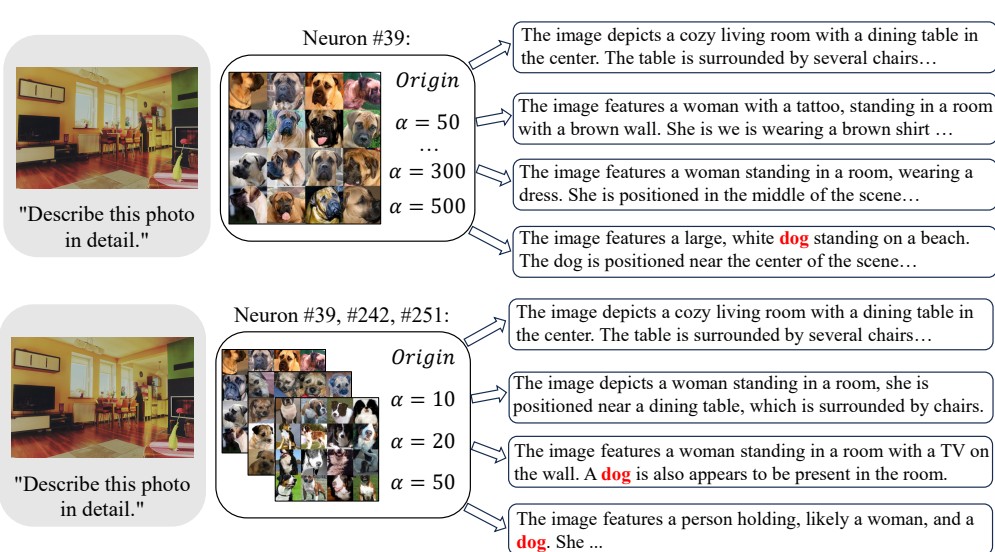

Figure 13: Concept insertion in complex contexts. (a) Steering with a single dog-related neuron requires a very large weight ($\alpha = 500$) to produce visible effects. (b) Coordinated steering of three dog-related neurons with smaller weights ($\alpha = 20$ each) yields natural insertions. This demonstrates the advantage of multi-neuron steering and motivates our CNS approach.

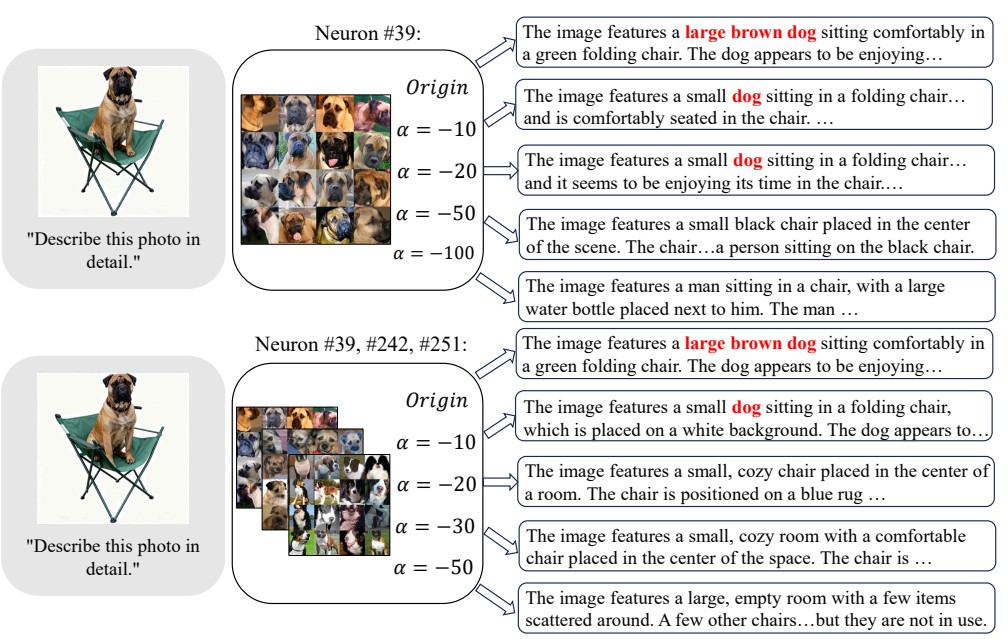

Figure 14: Concept suppression in complex contexts. (a) Suppressing a single dog-related neuron requires a very large negative weight ($\alpha = -100$) before the concept disappears from outputs. (b) Jointly suppressing three dog-related neurons with smaller weights ($\alpha = -10$ each) removes the concept more naturally and reliably, illustrating the effectiveness of multi-neuron steering.

Table 10: Results of the newer and stronger LLaVA-Next and LLaVA-OneVision models on the POPE benchmark, following SECOND's setup.

| Model | LLM | Method | F1 (↑) | | |
|---|---|---|---|---|---|
| | | | MSCOCO | OKVQA | GQA |
| LLaVA-Next (CLIP-336) | | baseline | 86.5 | 88.8 | 86.3 |
| | | baseline + CNS (ours) | 87.0 | 89.3 | 86.8 |
| | Vicuna-7B | VCD | 87.3 | 88.6 | 84.9 |
| | | SECOND | 87.5 | 89.1 | 86.3 |
| | | SECOND + CNS (ours) | **88.2** | **90.2** | **87.1** |
| LLaVA-OneVision (SigLIP-384) | | baseline | 87.4 | 88.7 | 86.3 |
| | | baseline + CNS (ours) | 87.9 | 89.2 | 86.8 |
| | Qwen2-0.5B | VCD | 86.4 | 88.9 | 86.4 |
| | | SECOND | 86.3 | 88.1 | 86.7 |
| | | SECOND + CNS (ours) | **87.1** | **89.6** | **87.6** |

Table 11: Results of the more recent and stronger Qwen2.5-VL on the POPE benchmark, following MFCD's setup.

| Model | Decoding | F1 Score (↑) | | |
|---|---|---|---|---|
| | | Random | Popular | Adversarial |
| | Sample (base) | 80.03 | 78.93 | 80.03 |
| | Sample + CNS (ours) | 80.61 | 79.52 | 80.58 |
| | Dola | 77.46 | 77.43 | 77.47 |
| Qwen2.5-VL | VCD | 81.39 | 79.91 | 79.98 |
| | SID | 79.95 | 79.38 | 78.82 |
| | MFCD (ours) | 82.91 | 82.01 | 81.75 |
| | MFCD + CNS (ours) | **83.45** | **82.54** | **82.29** |

## F    REBUTTAL

### F.1    EVALUATION ON MORE METHODS AND MODELS

To further verify the generality and robustness of VDC, we extend our evaluation to a broader set of **hallucination mitigation methods**, including DoLA (Chuang et al., 2023), OPERA (Huang et al., 2023), VCD (Leng et al., 2024), Woodpecker (Yin et al., 2023), LURE (Zhou et al., 2023), HALC (Chen et al., 2024), CODE (Kim et al., 2024b), EAH (Zhang et al., 2024b), VHR (He et al., 2025), AD-HH (Yang et al., 2025b), SID (Huo et al., 2024), SECOND (Park et al., 2025b), and MFCD (Liu et al., 2025), as well as diverse **LVLM architectures**, including MiniGPT-4 (Zhu et al., 2023), mPLUG-Owl2 (Ye et al., 2024), and the more recent and stronger LLaVA-Next (Liu et al., 2024c), LLaVA-OneVision, (Li et al., 2024) and Qwen2.5-VL (Bai et al., 2025). We evaluate these models on both the **POPE** and **CHAIR** benchmarks.

As shown in Tabs. 10, 11, 12, and 14, our proposed **CNS** consistently improves performance across different decoding strategies and hallucination mitigation methods. Specifically, CNS reduces hallucination rates (lower $CHAIR_S$ and $CHAIR_I$ scores) while maintaining or improving captioning quality (higher F1 and BLEU scores), demonstrating its effectiveness across a wide range of LVLM architectures and mitigation strategies. These results confirm that CNS is a generally applicable and robust module for enhancing the reliability of open-ended visual question answering and image captioning tasks.

### F.2    EVALUATION ON MORE CHALLENGING BENCHMARKS

We further evaluate our CNS method on several recently proposed, more challenging benchmarks to assess its robustness and generality.

Table 12: Results of the newer and stronger LLaVA-Next with a maximum token length of 128 on CHAIR Benchmark, following the VHR's setup.

| Method | CHAIRs ↓ | CHAIRi ↓ |
|---|---|---|
| Greedy | 29.08±2.09 | 8.08±0.74 |
| Greedy + CNS (ours) | 28.22±1.82 | 7.26±0.56 |
| DoLa | 28.76±2.58 | 8.12±0.78 |
| VCD | 30.80±2.48 | 8.72±0.94 |
| CODE | 27.84±2.73 | 7.98±0.92 |
| EAH | 28.13±1.13 | 6.62±0.49 |
| VHR | 24.96±2.09 | 6.80±0.59 |
| VHR + CNS (ours) | 24.42±1.68 | 6.28±0.36 |

Table 13: Following the evaluation protocol of AD-HH, we report results of $CHAIR_S$ and $CHAIR_I$ on COCO and Nocaps (out-of-domain) image captioning tasks, where lower scores indicate better performance. Our method yields strong improvements over existing approaches and can be seamlessly combined with AD-HH, achieving further reductions in hallucination rates.

| Methods | COCO | | | | Nocaps (Out-of-Domain) | | | |
|---|---|---|---|---|---|---|---|---|
| | LLaVA-7B | | MiniGPT-4 | | LLaVA-7B | | MiniGPT-4 | |
| | $CHAIR_S$ ↓ | $CHAIR_I$ ↓ | $CHAIR_S$ ↓ | $CHAIR_I$ ↓ | $CHAIR_S$ ↓ | $CHAIR_I$ ↓ | $CHAIR_S$ ↓ | $CHAIR_I$ ↓ |
| Greedy | 51.8 | 13.3 | 40.6 | 13.7 | 43.2 | 14.3 | 57.4 | 20.0 |
| Greedy + CNS (ours) | 49.6 | 12.2 | 40.4 | 13.2 | 39.6 | 13.4 | 53.8 | 18.2 |
| DoLA | 53.8 | 13.9 | 41.0 | 13.8 | 42.0 | 13.7 | 57.2 | 20.4 |
| OPERA | 50.2 | 14.5 | 35.2 | 12.8 | 44.2 | 14.4 | 46.2 | 16.2 |
| VCD | 55.4 | 15.7 | 38.8 | 14.8 | 43.6 | 14.4 | 48.2 | 17.5 |
| LURE | 51.2 | 13.4 | 46.4 | 14.2 | 41.8 | 14.4 | 55.8 | 19.6 |
| HALC | 50.2 | 12.4 | 36.4 | 11.8 | 40.2 | 12.2 | 53.0 | 18.0 |
| AD-HH | 29.6 | 8.0 | 35.2 | 11.7 | 35.6 | 9.4 | 46.8 | 16.2 |
| AD-HH + CNS (ours) | 28.9 | 7.8 | 34.6 | 11.3 | 35.1 | 9.1 | 46.2 | 15.6 |

**AMBER (Wang et al., 2023).** We adopt the generative subset of the AMBER, where models generate captions in response to the prompt "Describe the image." We measure hallucinations and caption quality using four metrics: CHAIR detects objects mentioned in the caption that are absent from the annotated descriptions, Cover measures the completeness of object coverage, Hal quantifies the hallucination rate, and Cog evaluates whether hallucinations resemble human-like patterns. To reduce computational costs for adversarial evaluation, we sample 50 images from this subset.

**MMHal-Bench (Sun et al., 2023).** This benchmark evaluates LVLMs from multiple perspectives, including attributes, relations, and counting. It assesses both hallucination rates and overall informativeness of the generated responses. An automatic GPT-4 evaluator compares model outputs to human-written references and ground truth object labels, ensuring a comprehensive assessment of visual understanding.

**HallusionBench (Guan et al., 2024).** Designed to test image-context reasoning, HallusionBench comprises 346 images and 1129 carefully crafted questions. It challenges LVLMs on nuanced visual reasoning, including GPT-4V(ision), Gemini Pro Vision, Claude 3, and LLaVA-1.5.

Across all three benchmarks (Tables 15,16 and 17), applying CNS consistently improves performance over baseline and other mitigation methods. These results demonstrate that CNS effectively reduces hallucinations and enhances reasoning and coverage in even more complex and diverse evaluation settings.

### F.3 MORE CASES ILLUSTRATING THE LINK BETWEEN HALLUCINATIONS AND NEURON ACTIVATIONS

Fig. 15 and Fig. 16 further demonstrate how hallucinations in LVLMs arise from abnormal or competing neuron activations under different captioning or questioning conditions, and how targeted

Table 14: Results of CHAIR Benchmark for various LVLMs using different decoding models and methods, following HALC's setup. Lower $CHAIR_S$ and $CHAIR_I$ scores indicate fewer hallucinations, while higher BLEU scores generally correspond to better captioning quality.

| Method | MiniGPT-4 | | | LLaVA-1.5 | | | mPLUG-Owl2 | | |
|---|---|---|---|---|---|---|---|---|---|
| | $CHAIR_S \downarrow$ | $CHAIR_I \downarrow$ | BLEU$\uparrow$ | $CHAIR_S \downarrow$ | $CHAIR_I \downarrow$ | BLEU$\uparrow$ | $CHAIR_S \downarrow$ | $CHAIR_I \downarrow$ | BLEU$\uparrow$ |
| Greedy | $30.87_{\pm5.45}$ | $12.33_{\pm2.07}$ | $14.33_{\pm0.00}$ | $20.80_{\pm0.08}$ | $6.77_{\pm0.07}$ | $15.93_{\pm0.00}$ | $23.20_{\pm0.35}$ | $8.33_{\pm0.28}$ | $15.37_{\pm0.00}$ |
| Greedy + CNS(ours) | $30.52_{\pm3.82}$ | $12.05_{\pm1.65}$ | $14.58_{\pm0.00}$ | $20.42_{\pm0.07}$ | $6.55_{\pm0.06}$ | $16.28_{\pm0.00}$ | $22.85_{\pm0.28}$ | $8.08_{\pm0.21}$ | $15.69_{\pm0.00}$ |
| DoLA | $30.87_{\pm2.52}$ | $11.70_{\pm0.13}$ | $14.93_{\pm0.00}$ | $21.00_{\pm0.67}$ | $6.70_{\pm0.38}$ | $15.93_{\pm0.00}$ | $24.60_{\pm0.24}$ | $8.73_{\pm0.30}$ | $15.40_{\pm0.00}$ |
| OPERA | $30.00_{\pm0.43}$ | $11.67_{\pm0.22}$ | $14.87_{\pm0.00}$ | $21.13_{\pm0.12}$ | $6.73_{\pm0.18}$ | $16.27_{\pm0.01}$ | $22.13_{\pm0.86}$ | $7.57_{\pm0.16}$ | $15.53_{\pm0.00}$ |
| Woodpecker | $28.87_{\pm2.20}$ | $10.20_{\pm0.85}$ | $15.30_{\pm0.01}$ | $23.85_{\pm4.62}$ | $7.50_{\pm0.01}$ | $17.05_{\pm0.00}$ | $26.33_{\pm1.98}$ | $8.43_{\pm0.80}$ | $16.43_{\pm0.00}$ |
| LURE | $27.88_{\pm2.25}$ | $10.20_{\pm0.85}$ | $15.03_{\pm0.11}$ | $19.48_{\pm2.35}$ | $6.5_{\pm0.38}$ | $15.97_{\pm0.01}$ | $21.27_{\pm0.06}$ | $7.67_{\pm0.16}$ | $15.65_{\pm0.05}$ |
| VCD | $30.27_{\pm0.44}$ | $12.60_{\pm0.45}$ | $14.33_{\pm0.00}$ | $23.33_{\pm5.66}$ | $7.90_{\pm0.53}$ | $14.67_{\pm0.01}$ | $27.27_{\pm7.32}$ | $9.73_{\pm1.22}$ | $14.40_{\pm0.00}$ |
| HALC | $17.80_{\pm0.03}$ | $8.10_{\pm0.14}$ | $14.91_{\pm0.00}$ | $13.80_{\pm0.08}$ | $5.50_{\pm0.14}$ | $16.10_{\pm0.01}$ | $17.33_{\pm4.30}$ | $7.43_{\pm0.11}$ | $16.27_{\pm0.00}$ |
| HALC + CNS(ours) | $17.35_{\pm0.02}$ | $7.70_{\pm0.12}$ | $15.25_{\pm0.00}$ | $13.30_{\pm0.07}$ | $5.15_{\pm0.12}$ | $16.45_{\pm0.01}$ | $16.70_{\pm3.80}$ | $7.05_{\pm0.09}$ | $16.65_{\pm0.00}$ |

Table 15: MMHal-Bench evaluates LVLMs on multiple aspects of visual understanding, including object attributes, relations, and counting. Metrics include Average Score (overall informativeness, higher is better) and Hallucination Rate (percentage of incorrect or hallucinated content, lower is better).

| Method | MMHal-Bench | |
|---|---|---|
| | Average Score $\uparrow$ | Hallucination Rate $\downarrow$ |
| baseline | 1.86 | 63.5 |
| baseline + CNS (ours) | 2.09 | 54.8 |
| VCD | 2.12 | 54.2 |
| VCD + CNS (ours) | 2.28 | 53.6 |
| OPERA | 2.33 | 50.0 |
| Less-is-more | 2.15 | 54.2 |
| VACoDe | 2.13 | 54.4 |

neuron modulation provides a principled way to suppress spurious signals and improve factual reliability.

In the black-apple case (Fig. 15), the model is asked a factual attribute question: "What is the color of the apple?". Although the image clearly contains a black apple, the model incorrectly answers "red." This error originates from the activation patterns inside the model. Neuron 2836, which is associated with red strawberries, shows unusually strong activation that overwhelms the evidence coming from apple-related neurons. This dominance causes the model to prioritize an irrelevant concept, leading to the hallucinated prediction. By suppressing neuron 2836 or enhancing apple-specific neurons 3085 and 1941, the activation distribution shifts toward the correct concept, enabling the model to output the accurate color "black." This case illustrates that hallucinations can emerge when irrelevant semantic units become overly activated and that controlling specific neurons helps restore proper grounding.

In the sheep and dog case (Fig. 16), the model is first asked to describe the image and successfully identifies the sheep as the primary object because sheep-related neurons exhibit strong and focused activation. However, because the sheep is partially occluded, neurons associated with other animals, such as the dog, are also activated. These extra activations do not affect general captioning, where the model only needs to describe the main scene. However, when the prompt shifts to a concept-specific question such as "Is there a dog in the image?", the residual activation of the dog-related concept becomes influential enough to mislead the model into answering "yes." After suppressing the dog-related neuron 2480 by setting its weight to a strongly negative value, the model correctly responds "no." This case shows that even mild unintended activation of unrelated concepts can produce hallucinations under targeted queries and that neuron-level control is effective in suppressing such spurious signals.

Together, these examples reveal a consistent pattern. Hallucinations often arise when semantically irrelevant neurons receive excessive activation or when competing concepts are inadvertently triggered by visual ambiguity or occlusion. By adjusting the activations of specific neurons, either by suppressing misleading semantic units or by amplifying the correct ones, the model's internal representation becomes more aligned with ground-truth visual evidence, resulting in more reliable and factual outputs.

Table 16: AMBER benchmark focuses on generative captioning hallucinations. Key metrics are CHAIR (detects objects mentioned in captions that do not exist in the image, lower is better), Cover (measures completeness of ground truth object coverage, higher is better), Hal (hallucination rate, lower is better), and Cog (evaluates human-like hallucination patterns, lower is better).

| Method | CHAIR ($\downarrow$) | Cover ($\uparrow$) | Hall ($\downarrow$) | Cog ($\downarrow$) |
|---|---|---|---|---|
| Regular | 7.8 | 51.0 | 36.4 | 4.2 |
| **Regular + CNS (ours)** | 7.2 | 51.8 | 34.1 | 3.8 |
| VCD | 7.5 | 50.8 | 36.2 | 4.1 |
| **VCD + CNS (ours)** | 7.1 | 51.6 | 33.2 | 3.4 |
| OPERA | 7.3 | 49.6 | 32.0 | 3.5 |
| DoLA | 7.6 | 51.6 | 36.0 | 4.0 |
| Woodpecker | 6.9 | 48.9 | 30.4 | 3.6 |
| M3ID | 7.4 | 49.9 | 33.2 | 3.7 |

Table 17: HallusionBench measures LVLM performance on complex image-context reasoning tasks. Metrics include Question Pair Accuracy (consistency between related questions), Figure Accuracy (reasoning on figures), Easy Question Accuracy, Hard Question Accuracy, and Overall Question Accuracy. Higher values indicate better reasoning performance.

| Model | Q. Pair Acc | Figure Acc | Easy Q. Acc | Hard Q. Acc | Question Acc |
|---|---|---|---|---|---|
| LLaVA-1.5 (GPT Eval) | 10.55 | 24.86 | 49.67 | 29.77 | 46.94 |
| + VCD | 10.92 | 25.13 | 49.88 | 30.05 | 47.12 |
| + CNS (ours) | 11.10 | 25.30 | 50.02 | 30.21 | 47.25 |
| + VCD + CNS (ours) | **11.35** | **25.58** | **50.19** | **30.44** | **47.38** |

## F.4 ABLATION ON SAE SCALE

Since the SAE architecture consists of only two linear layers, one for encoding and one for decoding, scaling is achieved by increasing the dimensionality of these layers. Beyond our default setting of 64, we tested expansion factors of 128, 192, and 256. As shown in Table 18, increasing the expansion factor generally improves performance and enhances hallucination mitigation. However, the gains diminish as the expansion factor grows larger because training

Table 18: Ablation study on the effect of scaling the SAE.

| Expand factor | Acc. | F1 | CHAIR$_S$ | CHAIR$_I$ |
|---|---|---|---|---|
| 64 | 85.20 | 85.49 | 25.7 | 8.8 |
| 128 | 87.43 | 87.56 | 22.4 | 7.2 |
| 192 | 88.28 | 88.34 | 20.6 | 6.4 |
| 256 | 89.09 | 89.18 | 19.4 | 5.6 |

SAEs with very high dimensionalities is more challenging. In particular, dead neurons, which remain inactive during training, become increasingly prevalent at higher dimensions and limit practical improvements. Fortunately, SAE architectures and training strategies are still evolving. We use the current state-of-the-art Matryoshka SAE and expect that future advances in SAE design or training methods may further improve feature disentanglement and reduce hallucinations.

## F.5 DISCUSSION AND COMPARISON WITH PREVIOUS APPROACHES THAT EDIT INTERNAL REPRESENTATIONS OF VLMS

We provide explicit discussion and comparison here. Jiang et al. (Jiang et al., 2025c) leverage a logits lens technique to project intermediate VLM features into the vocabulary space, enabling interpretation and editing of internal representations. Their intervention performs global orthogonalization in the latent space during decoding to suppress hallucination-related components. Kaduri et al. (Kaduri et al., 2025) present a detailed analysis of attention flow in VLMs, examining how visual information is encoded in query tokens and how cross-modal signals propagate across layers. Their study is inherently attention-centric and focuses on token-level interventions during decoding, including knockout experiments on attention modules.

In contrast, our work provides a complementary perspective. By employing Sparse Autoencoders, we decompose visual features into fine-grained, interpretable neuron-level components, which allows direct analysis and intervention on the semantic factors underlying hallucinations. Furthermore, our method operates entirely during the prefill stage, avoiding the need to modify representations at each decoding step. This results in a efficient, and mechanistically grounded approach that differs from prior decoding-side editing strategies and can naturally complement them in future studies. We sincerely thank the reviewers again for highlighting these works, which helped strengthen the positioning of our contribution.

## F.6 DISCUSSION AND COMPARISON WITH STANDARD FINE-TUNING-BASED ROBUSTNESS TECHNIQUES

Based on common observations, targeted fine-tuning may achieve stronger performance than a training-free approach. Fine-tuning large LVLMs for hallucination mitigation is generally computationally expensive and time-consuming, which is why training-free mitigation strategies remain more practical and widely adopted in current literature. In addition, prior work has lacked intuitive tools for directly comparing and interpreting internal feature activations, making it difficult to systematically analyze how perturbations affect the model's internal representations. This partially explains why fine-tuning-based robustness studies are rare in LVLM hallucination research.

Our SAE-based framework can be combined with fine-tuning-based robustness techniques, offering a complementary and highly interpretable perspective. By examining neuron-level activations for clean versus noisy inputs, we can directly observe how perturbations reshape semantic representations, an insight that was previously inaccessible. Beyond forcibly aligning entire representations, SAEs further allow alignment of the top-$k$ core semantic components between clean and noisy inputs, ensuring that the model consistently focuses on essential concepts rather than noise-induced spurious ones.

While combining fine-tuning with our SAE-based analysis would be an excellent and promising direction for future work, limited computational resources, along with the absence of established benchmarks and baselines for such LVLM-scale robustness fine-tuning, currently prevent us from pursuing this direction. Nevertheless, we believe our internal neuron-level approach provides a unique and valuable angle for understanding and mitigating hallucinations in LVLMs. This is indeed a valuable direction for future exploration, and we sincerely appreciate the reviewer's insightful suggestion.

## F.7 DISCUSSION AND COMPARISON WITH REGISTER NEURONS

We are actively exploring potential connections between our *always-on neurons* and other phenomena such as *register neurons* (Darcet et al., 2024; Jiang et al., 2025b), *massive activations* (Sun et al., 2024), and *attention sinks* (Xiao et al., 2023; Kang et al., 2025). Intuitively, these phenomena may be related, as all involve high-norm activations and exhibit input-invariant behavior. In our observations, always-on neurons typically have activation magnitudes between 10–80, whereas most of the top-40 neurons are in the 5–15 range. They consistently appear across inputs and primarily correspond to non-core, global features.

**Shared characteristics: input-invariant, globally stable activations**

- **Activation pattern:** persistently active across diverse inputs.
- **Input-independence:** independent of specific local visual content.
- **Global role:** encode global computations or statistical factors within the model.

However, there are key differences between always-on neurons and register neurons:

- **Origin:** Register neurons arise from the outputs of MLPs within a layer, whereas always-on neurons are extracted via SAE decomposition from the entire output of the layer.
- **Distribution and consistency:** Register neurons vary in number across images, with abnormal activations appearing on a different number of tokens for each input. Always-on neurons, in contrast, are consistently present in the same set of neurons across nearly all images, hence the name "always-on".

Table 19: Hallucination intervention results following the setup of Jiang et al. (Jiang et al., 2025c) on InstructBLIP and LLaVA-1.5.

| Model | Method | CHAIR$_i$ ↓ | CHAIR$_s$ ↓ |
|-------|--------|-----------|-----------|
| LLaVA-1.5 | Greedy | 49.2 | 14.2 |
| | Greedy + CNS (ours) | 47.6 | 13.4 |
| | Nucleus | 55.8 | 17.1 |
| | Nucleus + CNS (ours) | 54.6 | 16.3 |
| | Beam Search | 52.4 | 15.0 |
| | Beam Search + CNS (ours) | 51.8 | 14.6 |
| | OPERA | 44.8 | 12.8 |
| | OPERA + CNS (ours) | 44.2 | 12.1 |
| | Jiang et al. | 42.0 | 12.2 |
| | Jiang et al. + CNS (ours) | 41.4 | 11.8 |

Table 20: Comparison against both training-free and fine-tuning–based hallucination mitigation methods on the AMBER benchmark evaluated with LLaVA-1.5-7B. Results are reported in terms of CHAIR, Hallucination, and Cognitive Hallucination (Cog.), with lower scores indicating improved hallucination reduction.

| Method | AMBER | | |
|--------|-------|-----|------|
| | CHAIR ↓ | Hal. ↓ | Cog. ↓ |
| baseline | 8.4 | 35.5 | 4.0 |
| baseline + CNS (ours) | 7.6 | 34.2 | 3.4 |
| VCD (Leng et al., 2024) | 9.1 | 39.8 | 4.2 |
| VCD + CNS (ours) | 8.4 | 38.2 | 3.8 |
| OPERA (Huang et al., 2023) | 6.5 | 28.5 | 3.1 |
| OPERA + CNS (ours) | 5.8 | 27.2 | 2.8 |
| DoLa (Chuang et al., 2023) | 6.2 | 27.7 | 2.9 |
| DoLa + CNS (ours) | 5.6 | 26.4 | 2.3 |
| HA-DPO (Zhao et al., 2023) | 6.7 | 30.9 | 3.3 |
| EFUF (Xing et al., 2024) | 5.8 | 28.2 | 3.1 |
| POVID (Zhou et al., 2024b) | 5.3 | 28.7 | 3.0 |
| CLIP-DPO (Ouali et al., 2024) | 3.7 | 16.6 | 1.3 |
| RLAIF-V (Yu et al., 2024b) | 2.8 | 15.7 | 0.9 |
| TPO (He et al., 2024) | 3.6 | 20.5 | 1.6 |
| SENTINEL (Peng et al., 2025) | 2.9 | 14.6 | 1.2 |

- **Analysis and interpretability:** Always-on neurons can be directly visualized through SAE, providing intuitive insights into model behavior. Register neurons are primarily analyzed numerically and interpreted indirectly via their effect on model outputs.

In summary, while both exhibit stable, input-invariant activations, our always-on neurons represent global latent factors in the feature space, enabling direct and interpretable analysis of internal representations. In contrast, register neurons are structural components tied to MLP parameters.

**Further exploration:** Investigating deeper connections between always-on neurons and phenomena such as register neurons, massive activations, and attention sinks may require carefully designed experiments. Analyzing SAE factors from a norm-based perspective could also provide valuable insights, complementing traditional magnitude-based analyses. Importantly, SAEs offer a mechanism to interpret the underlying structure of these persistent or abnormal activations, potentially revealing why certain neurons consistently exhibit high activation across inputs. This interpretability enables a more intuitive understanding of these phenomena and represents a promising direction for future research.

## F.8 COMPARISON AGAINST BOTH TRAINING-FREE AND FINE-TUNING–BASED HALLUCINATION MITIGATION METHODS

We briefly review several representative hallucination-mitigation fine-tuning techniques. These approaches vary in supervision format, optimization strategy, and data construction pipeline, offering a broad landscape of current practice.

- **EFUF** (Xing et al., 2024). EFUF mitigates hallucinations without requiring paired data by combining gradient ascent with three specialized loss functions. The method performs gradient descent on real objects and gradient ascent on hallucinated ones, refining the model through contrastive adjustment of generation behaviors.

- **HA-DPO** (Zhao et al., 2023). HA-DPO formulates hallucination mitigation as a preference optimization task. Given two responses for the same image, the model is trained to prefer the non-hallucinated response using a DPO-style loss, augmented with a causal LM objective for stability. All samples are rewritten with GPT-4 to ensure stylistic consistency.

- **POVID** (Zhou et al., 2024b). POVID strengthens inferior responses by generating augmented hallucinated samples via GPT-4V and image perturbations. Using 17k preference pairs, the method fine-tunes LLaVA-1.5-7B to distinguish and avoid hallucinated outputs.

- **CLIP-DPO** (Ouali et al., 2024). CLIP-DPO replaces human or large-model scoring with CLIP-based preference signals. It uses CLIP as a reward evaluator to judge which response aligns better with image content, enabling scalable preference optimization without costly annotation or GPT judging.

- **RLAIF-V** (Yu et al., 2024b). RLAIF-V employs "feedback from peer models," decomposing a response into sub-responses and aggregating feedback from smaller models to reduce reliance on GPT-4. The final model is aligned through four iterative rounds of DPO training.

- **TPO** (He et al., 2024). TPO focuses on topic-level hallucinations through self-correction. It generates best/worst alternatives for each semantic topic using the model itself and constructs strong preference pairs via a deconfounded topic replacement process.

- **SENTINEL** (Peng et al., 2025). SENTINEL performs sentence-level early intervention to stop hallucinations before they propagate. It detects hallucinated objects using open-vocabulary detectors, labels faithful vs.hallucinated captions without human annotation, and applies preference training so the model favors hallucination-free descriptions.

Across these representative fine-tuning approaches, we observe that lightweight preference- or loss-based methods such as EFUF (Xing et al., 2024), HA-DPO (Zhao et al., 2023), and POVID (Zhou et al., 2024b) achieve moderate improvements while relying on modest training data and limited optimization. Their performance is comparable to training-free strategies, indicating that early-stage fine-tuning alone provides limited hallucination suppression.

In contrast, CLIP-DPO (Ouali et al., 2024), RLAIF-V (Yu et al., 2024b), TPO (He et al., 2024), and SENTINEL (Peng et al., 2025) introduce newly constructed preference datasets, external scoring modules, or multi-stage reinforcement-style optimization. Starting from CLIP-DPO, these methods achieve substantial gains in hallucination reduction, but at the cost of large-scale data generation, full-model fine-tuning, and multi-stage training.

Importantly, our SAE-based analysis may further improve these pipelines. By decomposing model representations into sparse, interpretable features, our method reveals which neurons are responsible for specific hallucination behaviors and why certain failure modes emerge. These insights can support these fine-tuning approaches in several ways. SAE-identified hallucination-related neurons highlight characteristic failure patterns, guiding the construction of more targeted and informative preference datasets. Examining neuron activations before and after each optimization stage can expose which hallucination behaviors remain unaddressed, providing diagnostics for multi-stage RL/DPO pipelines. Similarly, applying our neuron-level analyzes or techniques such as the logits lens analyzes proposed by Jiang et al. (Jiang et al., 2025c) to training data can help analyze underperforming data regions or training stages that require further refinement.

Some recent works have begun exploring SAE-driven data analysis for both LLMs (Jiang et al., 2025a; Yona et al., 2025) and LVLMs (Lou et al., 2025), demonstrating the promise of

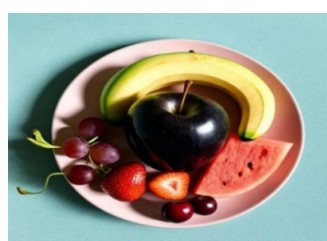

Q: What is the color of the apple?
A: The color of the apple is red.

Suppress the 2836 neuron associated with red strawberries.
Q: What is the color of the apple?
A: The color of the apple is black.

Enhance the 3085, 1941 neurons associated with apple.
Q: What is the color of the apple?
A: The color of the apple is black.

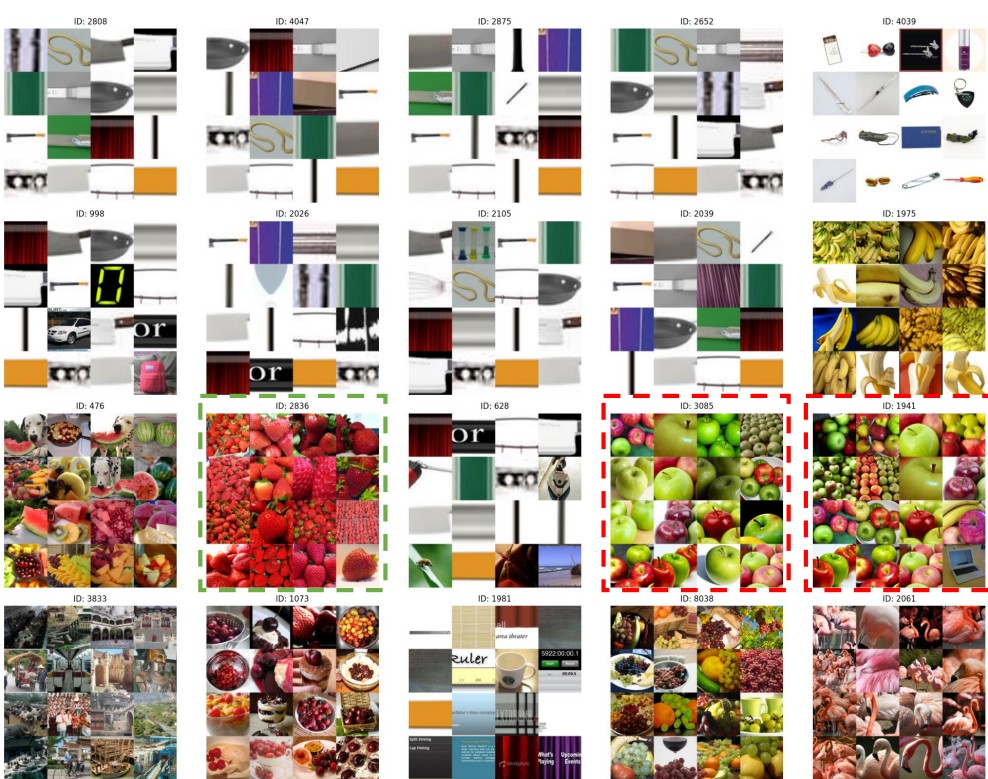

Figure 15: The image shows a fruit bowl containing multiple fruits, including a black apple. When asked "What is the color of the apple?" the model initially answers "red," reflecting a hallucinated prediction. This mistake arises because neuron 2836, which is associated with red strawberries (green boxes), exhibits unusually strong activation that overwhelms the evidence coming from apple-related neurons (red boxes). After suppressing neuron 2836 or enhancing apple-specific neurons 3085 and 1941, the model correctly outputs "black." This case illustrates how abnormal activation of irrelevant semantic units can lead to hallucinations and how targeted modulation of specific neurons can restore proper visual grounding and factual accuracy, thereby mitigating hallucinations.

representation-level tools in guiding data-centric improvements. Building on these advances, our framework enables richer analyses from multiple perspectives, including the behavior of different neuron types and their roles in hallucination emergence, thereby offering new interpretative angles for understanding and improving model training.

Leveraging such representation-level insights for data management and dataset design may ultimately yield more effective fine-tuning and stronger hallucination mitigation.

Figure 16: The activation maps indicate that the model primarily focuses on the sheep, with sheep-related neurons strongly activated (red boxes), allowing it to describe the image correctly. However, because the sheep is partially occluded, neurons linked to other animal concepts, such as the dog (green boxes), are also activated. While these additional activations do not affect general image description, they become problematic when the model is asked a concept-specific question such as "Is there a dog in the image?" leading it to incorrectly answer "yes." After suppressing the dog-related neuron 2480 by setting its weight to –10, the model correctly responds "no." This example highlights that extra activation of irrelevant concepts can mislead the model under targeted queries and that neuron-level modulation provides an effective means to suppress such spurious signals and mitigate hallucinations.

