# OpenReview forum: "Towards Interpretable Steering  for Hallucination Mitigation  in Large Vision–Language Models"
_ICLR.cc/2026/Conference — Submitted to ICLR 2026_

### Official Review · Reviewer_Ttv4 · 2025-10-26

**Soundness:** 3
**Presentation:** 3
**Contribution:** 3
**Rating:** 4
**Confidence:** 4

**Summary:**

This paper investigates the internal organization and dynamics of visual representations in VLMs, focusing on how these representations respond to perturbations and contribute to visual hallucinations. The authors use SAE to interpret and manipulate the internal visual features, showing how image noise alters neuron activations, disrupts semantic structure, and leads to hallucinated outputs. They demonstrate that targeted neuron-level interventions—enhancing or suppressing specific neurons—can effectively control LVLM outputs, and that coordinated multi-neuron modulation is more powerful than single-neuron manipulation. Building on these insights, the authors propose CNS, a method that amplifies meaningful neurons while suppressing perturbation-induced activations, mitigating hallucinations. CNS integrates seamlessly with existing decoding-based approaches and achieves consistent hallucination reduction across multiple benchmarks

**Strengths:**

* The paper is well written and clearly structured, making the contributions accessible and easy to follow.

* The proposed applications demonstrate convincing performance improvements, highlighting the practical benefits of the methods.

* The analysis section effectively reveals a clear relationship between noise strength, model performance, and neuron change ratio, providing valuable insights into the model’s internal dynamics.

* The approach is relatively lightweight, suggesting that it can be applied efficiently without excessive computational overhead.

**Weaknesses:**

* The paper claims in the introduction that “By applying SAEs to LVLMs, we decompose dense embeddings into sparse, monosemantic neurons, each corresponding to a single interpretable concept.” However, the evidence supporting this claim appears limited. The only validation provided relies on visualizing the top-16 images with the highest activation values. As the authors themselves acknowledge, feature splitting and feature hedging are well-known limitations of SAEs, which raises doubts about whether the identified neurons are truly monosemantic or disentangled. Without more rigorous quantitative or qualitative evidence (e.g., systematic concept alignment or disentanglement metrics), I suggest that the authors moderate this claim.

* The proposed method introduces several hyperparameters — $\lambda$, top-K, and $\tau$. Although ablation studies are included, the main text does not clearly describe how these parameters were selected. Were they tuned using a validation or training set? Are the same values used consistently across all experiments? Clarifying the hyperparameter selection process is important for reproducibility and fair evaluation.

* Conceptually, the method resembles a denoising procedure applied to internal representations by modifying their sparse neuron decompositions. While this is an interesting approach, it would be valuable to discuss how it compares to more standard robustness techniques — for example, finetuning the model on noisy images to encourage similar representations to their clean counterparts (e.g., by regressing to pseudo ground-truth features, as is often done for adversarial robustness). It remains unclear whether the proposed method offers advantages beyond such established approaches or whether it primarily achieves a similar effect through a more hand-crafted intervention.

* Missing prior work and comparisons to previous approaches that edit internal representations of VLMs:

[1] Jiang et al., Interpreting and Editing Vision-Language Representations to Mitigate Hallucinations, ICLR 2025

[2] Kaduri et al., What's in the Image? A Deep-Dive into the Vision of Vision Language Models, CVPR 2025

**Questions:**

* The paper mentions a small group of always-on neurons — are these essentially similar to the “register neurons” described in [3], except that they are extracted from SAE as opposed to MLPs?

* What are the scaling laws in this context? Does training larger SAEs (with appropriately adjusted parameters) lead to better hallucination reduction results?

[3] Jiang et al., Vision Transformers Don't Need Trained Registers, NeurIPS 2025

---

> ### Author Response · Authors · 2025-11-24
> **Response to Ttv4 (1/n)**
>
> _Thank you for your constructive and thoughtful comments. They were indeed helpful in improving the paper. We take this opportunity to address your concerns:_
>
> _**List of changes in the manuscript:**_
>
> > 1. `Lines 56-57` **We have moderated the description.**
> > 2. `Section F.4 in Appendix` **Scaling SAEs.**
> > 3. `Section F.5 in Appendix` **Discussion and comparison with previous approaches that edit internal representations of VLMs.**
> > 4. `Section F.6 in Appendix` **Discussion and comparison with standard fine-tuning-based robustness techniques.**
> > 5. `Section F.7 in Appendix` **Discussion and comparison with register neurons.**
> > 6. `Section F.8 in Appendix` **Comparison against both training-free and fine-tuning–based hallucination mitigation methods.**
>
> ## W1: Moderate the claim in Lines 56-57
> We have moderated the description:
>
> > By applying SAEs to LVLMs, we decompose dense embeddings into sparse neurons that tend to represent interpretable, concept-specific features
> >
>
> ## W2: More details about hyperparameters
> The relevant hyperparameters are kept consistent across all datasets and are not specifically tuned. Our main goal is to demonstrate that SAEs can be used to interpret and mitigate hallucinations in LVLMs. The method is designed to be simple yet effective while maintaining high reproducibility. While dataset-specific tuning, as done in some existing mitigation methods, might further improve performance, it would also increase the difficulty of reproducing our results. Therefore, we prioritize demonstrating SAE’s reliability and general applicability for analyzing and intervening in LVLM internal feature dynamics.
>
>
>
>
>
> ## **W3: Discussion and comparison with standard fine-tuning-based robustness techniques**
> Thank you for the insightful comment; this is indeed a promising direction.
>
> Based on common observations, targeted fine-tuning may achieve stronger performance than a training-free approach. Fine-tuning large LVLMs for hallucination mitigation is generally computationally expensive and time-consuming, which is why training-free mitigation strategies remain more practical and widely adopted in current literature. In addition, prior work has lacked intuitive tools for directly comparing and interpreting internal feature activations, making it difficult to systematically analyze how perturbations affect the model’s internal representations. This partially explains why fine-tuning-based robustness studies are rare in LVLM hallucination research.
>
> Our SAE-based framework can be combined with fine-tuning based robustness techniques, offering a complementary and highly interpretable perspective. By examining neuron-level activations for clean versus noisy inputs, we can directly observe how perturbations reshape semantic representations, an insight that was previously inaccessible. Beyond forcibly aligning entire representations, SAEs further allow alignment of the top-k core semantic components between clean and noisy inputs, ensuring that the model consistently focuses on essential concepts rather than noise-induced spurious ones.
>
> While combining fine-tuning with our SAE-based analysis would be an excellent and promising direction for future work, limited computational resources, along with the absence of established benchmarks and baselines for such LVLM-scale robustness fine-tuning, currently prevent us from pursuing this direction. Nevertheless, we believe our neuron-level approach provides a unique and valuable angle for understanding and mitigating hallucinations in LVLMs. This is indeed a valuable direction for future exploration, and we sincerely appreciate the reviewer’s insightful suggestion.
>
> More detailed results and analysis can be found in `Appendix F.6`

---

> > ### Comment · Reviewer_Ttv4 · 2025-11-27
> >
> > I thank the authors for their comments.
> >
> > Calling your method "training-free approach" is highly misleading, once again. You had to train an SAE to be able to do these edits, and therefore, I believe that this approach can and should be compared to fine-tuning approaches.

---

> ### Author Response · Authors · 2025-11-24
> **Response to Ttv4 (2/n)**
>
> ## W4: Discussion and comparison with previous approaches that edit internal representations of VLMs
> Thank you for pointing out these important related works. We have incorporated the suggested prior work, **Jiang et al. (ICLR 2025)** and **Kaduri et al. (CVPR 2025)**, into the revised manuscript, and we now provide explicit discussion and comparison.
>
> **Jiang et al. (ICLR 2025)** leverage a **logits lens** technique to project intermediate VLM features into the **vocabulary space**, enabling interpretation and editing of internal representations. Their intervention performs **global orthogonalization in the latent space during the decoding stage** to suppress hallucination-related components.
>
> **Kaduri et al. (CVPR 2025)** present a detailed analysis of **attention flow** in VLMs, examining how visual information is encoded in query tokens and how cross-modal signals propagate across layers. Their study is inherently **attention-centric** and focuses on **token-level interventions during decoding**, including knockout experiments on attention modules.
>
> In contrast, our work provides a **new and complementary perspective**. By using **Sparse Autoencoders**, we decompose visual features into **fine-grained, interpretable neuron-level components**, which allows us to directly analyze and intervene on the underlying semantic factors of hallucinations. Moreover, our method operates entirely once in the **prefill stage**, avoiding the need to modify representations at every decoding step. This leads to an **efficient and mechanistically grounded** approach that differs from prior decoding-side editing strategies and can naturally complement them in future studies.
>
> We sincerely thank the reviewer for highlighting these insightful related works, which helped us strengthen the positioning of our contribution.
>
> More detailed results and analysis can be found in `Appendix F.5`
>
> ## Q1: Discussion and comparison with register neurons
> This is an extremely insightful question — thank you very much. We are actively exploring potential connections between our always-on neurons and other phenomena such as **register neurons, massive activations, and attention sinks**. Intuitively, these phenomena may be related, as all involve high-norm activations and exhibit input-invariant behavior. In our observations, always-on neurons typically have activation magnitudes between 10–80, whereas most of the top-40 neurons are in the 5–15 range. They consistently appear across inputs and primarily correspond to non-core, global features.
>
> **Shared characteristics: input-invariant, globally stable activations**
>
> + **Activation pattern:** persistently active across diverse inputs.
> + **Input-independence:** independent of specific local visual content.
> + **Global role:** encode global computations or statistical factors within the model.
>
> However, there are key differences between always-on neurons and register neurons:
>
> + **Origin:** Register neurons arise from the outputs of MLPs within a layer, whereas always-on neurons are extracted via SAE decomposition from the entire output of the layer.
> + **Distribution and consistency:** Register neurons vary in number across images, with abnormal activations appearing on a different number of tokens for each input. Always-on neurons, in contrast, are consistently present in the same set of neurons across nearly all images, hence the name “always-on.”
> + **Analysis and interpretability:** Always-on neurons can be directly visualized through SAE, providing intuitive insights into model behavior. Register neurons are primarily analyzed numerically and interpreted indirectly via their effect on model outputs.
>
> In summary, while both exhibit stable, input-invariant activations, our always-on neurons represent global latent factors in the feature space, enabling direct and interpretable analysis of internal representations. In contrast, register neurons are structural components tied to MLP parameters.
>
> **Further exploration:** Investigating deeper connections between always-on neurons and phenomena such as register neurons, massive activations, and attention sinks may require carefully designed experiments. Analyzing SAE factors from a norm-based perspective could also provide valuable insights, complementing traditional magnitude-based analyses. Importantly, SAE offers a mechanism to interpret the underlying structure of these persistent or abnormal activations, potentially revealing why certain neurons consistently exhibit high activation across inputs. This interpretability enables a more intuitive understanding of these phenomena and represents a promising direction for future research.
>
> More detailed results and analysis can be found in `Appendix F.7`

---

> ### Author Response · Authors · 2025-11-24
> **Response to Ttv4 (3/n)**
>
> ## Q2: Scaling SAEs
> We thank the reviewer for the insightful suggestion. Since the SAE architecture consists of only two linear layers, one for encoding and one for decoding, scaling is achieved by increasing the dimensionality of these layers. Beyond our default setting of 64, we tested expansion factors of 128, 192, and 256. As shown in Table 18, increasing the expansion factor generally improves performance and enhances hallucination mitigation. However, the gains diminish as the expansion factor grows larger because training SAEs with very high dimensionalities is more challenging. In particular, dead neurons, which remain inactive during training, become increasingly prevalent at higher dimensions and limit practical improvements. Fortunately, SAE architectures and training strategies are still evolving. We use the current state-of-the-art Matryoshka SAE and expect that future advances in SAE design or training methods may further improve feature disentanglement and reduce hallucinations. We sincerely appreciate the reviewer’s thoughtful comments.
>
> More detailed results and analysis can be found in `Appendix F.4`
>
> | Expand factor | Acc.   | F1     | CHAIR\_S | CHAIR\_I |
> |---------------|--------|--------|----------|----------|
> | 64            | 85.20  | 85.49  | 25.7     | 8.8      |
> | 128           | 87.43  | 87.56  | 22.4     | 7.2      |
> | 192           | 88.28  | 88.34  | 20.6     | 6.4      |
> | 256           | 89.09  | 89.18  | 19.4     | 5.6      |

---

> ### Author Response · Authors · 2025-11-27
>
> Thank you for pointing this out in the reply.  We have corrected the wording accordingly in the Response (2/n) .
>
> We are currently organizing representative fine-tuning–based approaches to include in our comparison.

---

> ### Author Response · Authors · 2025-11-28
> **Response to Ttv4 (1/2)**
>
> **Comparison Against Both Training-Free and Fine-Tuning–Based Hallucination Mitigation Methods**
> We briefly review several representative hallucination-mitigation fine-tuning techniques. These approaches vary in supervision format, optimization strategy, and data construction pipeline, offering a broad landscape of current practice.
>
> + **EFUF** (Xing et al., 2024). EFUF mitigates hallucinations without requiring paired data by combining gradient ascent with three specialized loss functions. The method performs gradient descent on real objects and gradient ascent on hallucinated ones, refining the model through contrastive adjustment of generation behaviors.
> + **HA-DPO** (Zhao et al., 2023). HA-DPO formulates hallucination mitigation as a preference optimization task. Given two responses for the same image, the model is trained to prefer the non-hallucinated response using a DPO-style loss, augmented with a causal LM objective for stability. All samples are rewritten with GPT-4 to ensure stylistic consistency.
> + **POVID** (Zhou et al., 2024). POVID strengthens inferior responses by generating augmented hallucinated samples via GPT-4V and image perturbations. Using 17k preference pairs, the method fine-tunes LLaVA-1.5-7B to distinguish and avoid hallucinated outputs.
> + **CLIP-DPO** (Ouali et al., 2024). CLIP-DPO replaces human or large-model scoring with CLIP-based preference signals. It uses CLIP as a reward evaluator to judge which response aligns better with image content, enabling scalable preference optimization without costly annotation or GPT judging.
> + **RLAIF-V** (Yu et al., 2024). RLAIF-V employs “feedback from peer models,” decomposing a response into sub-responses and aggregating feedback from smaller models to reduce reliance on GPT-4. The final model is aligned through four iterative rounds of DPO training.
> + **TPO** (He et al., 2024). TPO focuses on topic-level hallucinations through self-correction. It generates best/worst alternatives for each semantic topic using the model itself and constructs strong preference pairs via a deconfounded topic replacement process.
> + **SENTINEL** (Peng et al., 2025). SENTINEL performs sentence-level early intervention to stop hallucinations before they propagate. It detects hallucinated objects using open-vocabulary detectors, labels faithful vs. hallucinated captions without human annotation, and applies preference training so the model favors hallucination-free descriptions.
>
> ---
> Across these representative fine-tuning approaches, we observe that lightweight preference- or loss-based methods such as EFUF [Xing et al. 2024], HA-DPO [Zhao et al. 2023], and POVID [Zhou et al. 2024] achieve moderate improvements while relying on modest training data and limited optimization. Their performance is comparable to training-free strategies, indicating that early-stage fine-tuning alone provides limited hallucination suppression.
>
> In contrast, CLIP-DPO [Ouali et al. 2024], RLAIF-V [Yu et al. 2024], TPO [He et al. 2024], and SENTINEL [Peng et al. 2025] introduce newly constructed preference datasets, external scoring modules, or multi-stage reinforcement-style optimization. Starting from CLIP-DPO, these methods achieve substantial gains in hallucination reduction, but at the cost of large-scale data generation, full-model fine-tuning, and multi-stage training.
>
> Importantly, our SAE-based analysis may further improve these pipelines. By decomposing model representations into sparse, interpretable features, our method reveals which neurons are responsible for specific hallucination behaviors and why certain failure modes emerge.
>
> These insights can support better fine-tuning approaches in several ways:
>
> + **Targeted data construction:**
> SAE-identified hallucination-related neurons highlight characteristic failure patterns, guiding the design of more targeted and informative preference datasets.
> + **Training diagnostics:**
> Examining neuron activations before and after each optimization stage can expose which hallucination behaviors remain unaddressed, providing diagnostics for multi-stage RL/DPO pipelines.
> + **Data refinement via representation-level analysis:**
> Techniques such as **our neuron-level analyses or the logits lens analyses proposed by `Jiang et al. (2024)`** can help analyse underperforming data subsets or training stages that require further refinement that require further refinement. Some recent works have begun exploring SAE-driven data analysis for both LLMs and LVLMs, demonstrating the promise of representation-level tools in guiding data-centric improvements.
>
> Leveraging these insights for data management and dataset design may lead to more effective fine-tuning and stronger hallucination mitigation.
>
> More detailed results and analysis can be found in `Appendix F.8`
>
> Thank you for your timely and thoughtful comments.

---

> ### Author Response · Authors · 2025-11-28
> **Response to Ttv4 (2/2)**
>
> Table 1: Hallucination Intervention Results
> | Model | Method | CHAIRᵢ ↓ | CHAIRₛ ↓ |
> | --- | --- | --- | --- |
> | **LLaVA-1.5** | Greedy | 49.2 | 14.2 |
> | | Greedy + CNS (ours) | 47.6 | 13.4 |
> | | Nucleus | 55.8 | 17.1 |
> | | Nucleus + CNS (ours) | 54.6 | 16.3 |
> | | Beam Search | 52.4 | 15.0 |
> | | Beam Search + CNS (ours) | 51.8 | 14.6 |
> | | OPERA | 44.8 | 12.8 |
> | | OPERA + CNS (ours) | 44.2 | 12.1 |
> | | Jiang et al. | 42.0 | 12.2 |
> | | Jiang et al. + CNS (ours) | 41.4 | 11.8 |
>
>
> ---
>
>  Table 2: Comparison Against Training-Free and Fine-Tuning–Based Methods on AMBER
> | Method | CHAIR ↓ | Hal. ↓ | Cog. ↓ |
> | --- | --- | --- | --- |
> | baseline | 8.4 | 35.5 | 4.0 |
> | baseline + CNS (ours) | 7.6 | 34.2 | 3.4 |
> | VCD | 9.1 | 39.8 | 4.2 |
> | VCD + CNS (ours) | 8.4 | 38.2 | 3.8 |
> | OPERA | 6.5 | 28.5 | 3.1 |
> | OPERA + CNS (ours) | 5.8 | 27.2 | 2.8 |
> | DoLa | 6.2 | 27.7 | 2.9 |
> | DoLa + CNS (ours) | 5.6 | 26.4 | 2.3 |
> | HA-DPO | 6.7 | 30.9 | 3.3 |
> | EFUF | 5.8 | 28.2 | 3.1 |
> | POVID | 5.3 | 28.7 | 3.0 |
> | CLIP-DPO | 3.7 | 16.6 | 1.3 |
> | RLAIF-V | 2.8 | 15.7 | 0.9 |
> | TPO | 3.6 | 20.5 | 1.6 |
> | SENTINEL | 2.9 | 14.6 | 1.2 |
>
>
> ---

---

### Official Review · Reviewer_MCny · 2025-10-28

**Soundness:** 3
**Presentation:** 3
**Contribution:** 3
**Rating:** 6
**Confidence:** 3

**Summary:**

This paper investigates the relationship between internal visual representations and hallucinations in large vision–language models (LVLMs). By introducing sparse autoencoders (SAEs), the authors disentangle dense visual features into interpretable monosemantic neurons and study how perturbations (e.g., noise) reshape them. They find that noise alters neuron activations and disrupts semantic structures, leading to hallucinations. Building on these insights, they propose Contrastive Neuron Steering (CNS), which amplifies truth-consistent neurons while suppressing perturbation-induced ones, achieving improved interpretability and hallucination mitigation across multiple LVLMs.

**Strengths:**

1. Novel interpretability perspective.
The paper offers an insightful neuron-level analysis of LVLM hallucinations, revealing how perturbations reshape visual semantics. This represents a clear advancement over purely decoding-based methods.

2. Simple yet elegant approach. The method is conceptually intuitive, leveraging noisy–clean contrasts to steer neuron activations and does not require retraining.

3. Complementary to existing methods. CNS can be combined with decoding-based techniques such as VCD or M3ID, showing consistent improvements across multiple LVLMs.

**Weaknesses:**

1. The evaluation primarily relies on relatively coarse, binary-style metrics (POPE, MME), which capture existence-level hallucinations but not free-form generative ones. Including open-ended benchmarks such as AMBER [1] or MMHal-Bench [2] would better validate the method’s generality and reveal how neuron-level steering behaves under natural caption generation.

2. The neuron-level findings are interesting, but the link between interpretability and performance improvement remains somewhat superficial. It would strengthen the work to include controlled experiments that verify whether manipulating the identified neurons genuinely leads to consistent hallucination reduction across diverse prompts or noise conditions.

3. Experiments are limited to some old LVLMs (LLaVA, InstructBLIP, Qwen-VL). Testing on newer, stronger models such as DeepSeek-VL or LLaVA-OneVision would strengthen claims of generalizability.

[1] Amber: An LLM-free Multi-dimensional Benchmark for MLLMs Hallucination Evaluation, 2023.

[2] Aligning Large Multimodal Models with Factually Augmented RLHF, ACL 2024.

**Questions:**

See weakness

**Details Of Ethics Concerns:**

No concern

---

> ### Author Response · Authors · 2025-11-24
> **Response to MCny (1/n)**
>
> _Thank you for your constructive and thoughtful comments. They were indeed helpful in improving the paper. We take this opportunity to address your concerns:_
>
> _**List of changes in the manuscript:**_
>
> > 1. `Section F.1 in Appendix` **Evaluation on more challenging benchmarks.**
> > 2. `Section F.2 in Appendix` **Evaluation on more methods and models.**
> > 3. `Section F.3 in Appendix` **More cases illustrating the link between hallucinations and neuron activations.**
> > 4. `Figure 5 in Main Paper`**Add the analyses of always-active neurons and image-specific neurons.**
> > 5. `Table 1 in Main Paper` **Effects of suppressing always-active, image-specific, and random neurons.**
> >
>
> ## W1 & W3. More Experiments on additional benchmarks, methods and models
> We thank the reviewer for raising this concern.  To further verify the **generality and robustness** of CNS, we extend our evaluation to a broader set of **hallucination mitigation methods**, including DoLA, OPERA, VCD, Woodpecker, LURE, HALC, CODE, EAH, VHR, AD-HH, SID, SECOND, and MFCD, as well as diverse **LVLM architectures**, such as MiniGPT-4, mPLUG-Owl2, **LLaVA-Next, LLaVA-OneVision, and Qwen2.5-VL**. All models are evaluated on both POPE and CHAIR benchmarks.   Since there are currently no established methods available for DeepSeek-VL, we additionally evaluated LLaVA-Next and Qwen2.5-VL to provide complementary comparisons and ensure a fair evaluation of our approach.
>
> As shown in Tables 10, 11, 12, 13, and 14, CNS **consistently improves performance** across different mitigation methods and decoding strategies. In particular, it **reduces hallucination rates** (lower CHAIRS_SS and CHAIRI_II scores) while maintaining or improving captioning quality (higher F1 and BLEU scores). These results demonstrate that CNS is a **generally applicable and robust module** for enhancing the reliability of open-ended visual question answering and image captioning tasks.
>
> We further evaluate CNS on **more challenging benchmarks** to assess its effectiveness in scenarios requiring **precise image-specific feature understanding**:
>
> + **AMBER (Table 16):** Models generate captions in response to “Describe the image.” Evaluation uses four metrics: CHAIR (hallucination detection), Cover (completeness of object coverage), Hal (hallucination rate), and Cog (human-like hallucination patterns). To reduce computational cost, we use a subset of 50 images.
> + **MMHal-Bench (Table 15):** Evaluates attributes, relations, counting, and overall informativeness. An automatic evaluator compares model outputs with human-written references and ground-truth object labels.
> + **HallusionBench (Table 17):** Designed to test nuanced **image-context reasoning**, containing 346 images and 1129 human-crafted questions, challenging advanced LVLMs such as GPT-4V, Gemini Pro Vision, Claude 3, and LLaVA-1.5.
>
> Across these benchmarks, CNS **consistently improves over baseline and other mitigation methods**, showing that targeted neuron-level modulation can **reduce hallucinations**, enhance **semantic and image-specific grounding**, and strengthen reasoning in diverse and complex evaluation scenarios.
>
> More detailed results and analysis can be found in `Appendix F.1 and F.2`.
>
>
>
> **Table R.1: POPE Benchmark – Newer LLaVA-Next and LLaVA-OneVision**
>
> | Model | LLM | Method | MSCOCO F1 ↑ | OKVQA F1 ↑ | GQA F1 ↑ |
> | --- | --- | --- | --- | --- | --- |
> | LLaVA-Next (CLIP-336) | Vicuna-7B | baseline | 86.5 | 88.8 | 86.3 |
> | | | baseline + CNS (ours) | 87.0 | 89.3 | 86.8 |
> | | | VCD | 87.3 | 88.6 | 84.9 |
> | | | SECOND | 87.5 | 89.1 | 86.3 |
> | | | SECOND + CNS (ours) | 88.2 | 90.2 | 87.1 |
> | LLaVA-OneVision (SigLIP-384) | Qwen2-0.5B | baseline | 87.4 | 88.7 | 86.3 |
> | | | baseline + CNS (ours) | 87.9 | 89.2 | 86.8 |
> | | | VCD | 86.4 | 88.9 | 86.4 |
> | | | SECOND | 86.3 | 88.1 | 86.7 |
> | | | SECOND + CNS (ours) | 87.1 | 89.6 | 87.6 |
>
>
> ---
>
> **Table R.2: POPE Benchmark – Newer Qwen2.5-VL**
>
> | Model | Decoding | Random F1 ↑ | Popular F1 ↑ | Adversarial F1 ↑ |
> | --- | --- | --- | --- | --- |
> | Qwen2.5-VL | Sample (base) | 80.03 | 78.93 | 80.03 |
> | | Sample + CNS (ours) | 80.61 | 79.52 | 80.58 |
> | | Dola | 77.46 | 77.43 | 77.47 |
> | | VCD | 81.39 | 79.91 | 79.98 |
> | | SID | 79.95 | 79.38 | 78.82 |
> | | MFCD (ours) | 82.91 | 82.01 | 81.75 |
> | | MFCD + CNS (ours) | 83.45 | 82.54 | 82.29 |
>
>
> ---
>
> **Table R.3: CHAIR Benchmark – Newer LLaVA-Next**
>
> | Method | CHAIR(_S) ↓ | CHAIR(_I) ↓ |
> | --- | --- | --- |
> | Greedy | 29.08 ± 2.09 | 8.08 ± 0.74 |
> | Greedy + CNS (ours) | 28.22 ± 1.82 | 7.26 ± 0.56 |
> | DoLa | 28.76 ± 2.58 | 8.12 ± 0.78 |
> | VCD | 30.80 ± 2.48 | 8.72 ± 0.94 |
> | CODE | 27.84 ± 2.73 | 7.98 ± 0.92 |
> | EAH | 28.13 ± 1.13 | 6.62 ± 0.49 |
> | VHR | 24.96 ± 2.09 | 6.80 ± 0.59 |
> | VHR + CNS (ours) | 24.42 ± 1.68 | 6.28 ± 0.36 |
>
>
> ---

---

> ### Author Response · Authors · 2025-11-24
> **Response to MCny (2/n)**
>
> **Table R.4: CHAIR – COCO and Nocaps (AD-HH Protocol)**
>
> | Methods | COCO LLaVA-7B CHAIR(_S) ↓ | CHAIR(_I) ↓ | COCO MiniGPT-4 CHAIR(_S) ↓ | CHAIR(_I) ↓ | Nocaps LLaVA-7B CHAIR(_S) ↓ | CHAIR(_I) ↓ | Nocaps MiniGPT-4 CHAIR(_S) ↓ | CHAIR(_I) ↓ |
> | --- | --- | --- | --- | --- | --- | --- | --- | --- |
> | Greedy | 51.8 | 13.3 | 40.6 | 13.7 | 43.2 | 14.3 | 57.4 | 20.0 |
> | Greedy + CNS (ours) | 49.6 | 12.2 | 40.4 | 13.2 | 39.6 | 13.4 | 53.8 | 18.2 |
> | DoLA | 53.8 | 13.9 | 41.0 | 13.8 | 42.0 | 13.7 | 57.2 | 20.4 |
> | OPERA | 50.2 | 14.5 | 35.2 | 12.8 | 44.2 | 14.4 | 46.2 | 16.2 |
> | VCD | 55.4 | 15.7 | 38.8 | 14.8 | 43.6 | 14.4 | 48.2 | 17.5 |
> | LURE | 51.2 | 13.4 | 46.4 | 14.2 | 41.8 | 14.4 | 55.8 | 19.6 |
> | HALC | 50.2 | 12.4 | 36.4 | 11.8 | 40.2 | 12.2 | 53.0 | 18.0 |
> | AD-HH | 29.6 | 8.0 | 35.2 | 11.7 | 35.6 | 9.4 | 46.8 | 16.2 |
> | AD-HH + CNS (ours) | 28.9 | 7.8 | 34.6 | 11.3 | 35.1 | 9.1 | 46.2 | 15.6 |
>
>
> ---
>
> **Table R.5: CHAIR Benchmark – HALC Setup**
>
> | Method | MiniGPT-4 CHAIR(_S) ↓ | CHAIR(_I) ↓ | BLEU ↑ | LLaVA-1.5 CHAIR(_S) ↓ | CHAIR(_I) ↓ | BLEU ↑ | mPLUG-Owl2 CHAIR(_S) ↓ | CHAIR(_I) ↓ | BLEU ↑ |
> | --- | --- | --- | --- | --- | --- | --- | --- | --- | --- |
> | Greedy | 30.87 ± 5.45 | 12.33 ± 2.07 | 14.33 ± 0.00 | 20.80 ± 0.08 | 6.77 ± 0.07 | 15.93 ± 0.00 | 23.20 ± 0.35 | 8.33 ± 0.28 | 15.37 ± 0.00 |
> | Greedy + CNS (ours) | 30.52 ± 3.82 | 12.05 ± 1.65 | 14.58 ± 0.00 | 20.42 ± 0.07 | 6.55 ± 0.06 | 16.28 ± 0.00 | 22.85 ± 0.28 | 8.08 ± 0.21 | 15.69 ± 0.00 |
> | DoLA | 30.87 ± 2.52 | 11.70 ± 0.13 | 14.93 ± 0.00 | 21.00 ± 0.67 | 6.70 ± 0.38 | 15.93 ± 0.00 | 24.60 ± 0.24 | 8.73 ± 0.30 | 15.40 ± 0.00 |
> | OPERA | 30.00 ± 0.43 | 11.67 ± 0.22 | 14.87 ± 0.00 | 21.13 ± 0.12 | 6.73 ± 0.18 | 16.27 ± 0.01 | 22.13 ± 0.86 | 7.57 ± 0.16 | 15.53 ± 0.00 |
> | Woodpecker | 28.87 ± 2.20 | 10.20 ± 0.85 | 15.30 ± 0.01 | 23.85 ± 4.62 | 7.50 ± 0.01 | 17.05 ± 0.00 | 26.33 ± 1.98 | 8.43 ± 0.80 | 16.43 ± 0.00 |
> | LURE | 27.88 ± 2.25 | 10.20 ± 0.85 | 15.03 ± 0.11 | 19.48 ± 2.35 | 6.50 ± 0.38 | 15.97 ± 0.01 | 21.27 ± 0.06 | 7.67 ± 0.16 | 15.65 ± 0.05 |
> | VCD | 30.27 ± 0.44 | 12.60 ± 0.45 | 14.33 ± 0.00 | 23.33 ± 5.66 | 7.90 ± 0.53 | 14.67 ± 0.01 | 27.27 ± 7.32 | 9.73 ± 1.22 | 14.40 ± 0.00 |
> | HALC | 17.80 ± 0.03 | 8.10 ± 0.14 | 14.91 ± 0.00 | 13.80 ± 0.08 | 5.50 ± 0.14 | 16.10 ± 0.01 | 17.33 ± 4.30 | 7.43 ± 0.11 | 16.27 ± 0.00 |
> | HALC + CNS (ours) | 17.35 ± 0.02 | 7.70 ± 0.12 | 15.25 ± 0.00 | 13.30 ± 0.07 | 5.15 ± 0.12 | 16.45 ± 0.01 | 16.70 ± 3.80 | 7.05 ± 0.09 | 16.65 ± 0.00 |
>
>
> ---
>
> **Table R.6: MMHal-Bench**
>
> | Method | Average Score ↑ | Hallucination Rate ↓ |
> | --- | --- | --- |
> | baseline | 1.86 | 63.5 |
> | baseline + CNS (ours) | 2.09 | 54.8 |
> | VCD | 2.12 | 54.2 |
> | VCD + CNS (ours) | 2.28 | 53.6 |
> | OPERA | 2.33 | 50.0 |
> | Less-is-more | 2.15 | 54.2 |
> | VACoDe | 2.13 | 54.4 |
>
>
> ---
>
> **Table R.7: AMBER Benchmark**
>
> | Method | CHAIR ↓ | Cover ↑ | Hal ↓ | Cog ↓ |
> | --- | --- | --- | --- | --- |
> | Regular | 7.8 | 51.0 | 36.4 | 4.2 |
> | Regular + CNS (ours) | 7.2 | 51.8 | 34.1 | 3.8 |
> | VCD | 7.5 | 50.8 | 36.2 | 4.1 |
> | VCD + CNS (ours) | 7.1 | 51.6 | 33.2 | 3.4 |
> | OPERA | 7.3 | 49.6 | 32.0 | 3.5 |
> | DoLA | 7.6 | 51.6 | 36.0 | 4.0 |
> | Woodpecker | 6.9 | 48.9 | 30.4 | 3.6 |
> | M3ID | 7.4 | 49.9 | 33.2 | 3.7 |
>
>
> ---
>
> **Table R.8: HallusionBench**
>
> | Model | Q. Pair Acc ↑ | Figure Acc ↑ | Easy Q. Acc ↑ | Hard Q. Acc ↑ | Question Acc ↑ |
> | --- | --- | --- | --- | --- | --- |
> | LLaVA-1.5 (GPT Eval) | 10.55 | 24.86 | 49.67 | 29.77 | 46.94 |
> | + VCD | 10.92 | 25.13 | 49.88 | 30.05 | 47.12 |
> | + CNS (ours) | 11.10 | 25.30 | 50.02 | 30.21 | 47.25 |
> | + VCD + CNS (ours) | 11.35 | 25.58 | 50.19 | 30.44 | 47.38 |

---

> ### Author Response · Authors · 2025-11-24
> **Response to MCny (3/n)**
>
> ## W2：More analyses and experiments about neuron-level  interpretability and model performance
> We thank the reviewer for this very insightful suggestion.
>
> Following the reviewer’s suggestion, we implemented three improvements to deepen the analysis of neuron instability and hallucination.
>
> **First**, we refined the original patch-level neuron-change analysis by decomposing activation changes across different neuron categories. Specifically, we separately visualized the activation changes of Always-active neurons and image-specific neurons. Consistent with our earlier findings, the top 10 neurons are predominantly Always-active neurons, while ranks 10–20 contain a mixture of image-specific neurons. Figure 5 shows that increasing noise steps causes LLaVA performance to degrade and simultaneously increases the neuron-change ratio. Patch-level neurons change the fastest, indicating that noise perturbs nearly all token-level neurons, which is consistent with our previous observation that patch-level neurons do not have stable semantic roles and therefore vary significantly under perturbation. In contrast, Always-active neurons exhibit minimal change, reflecting their stability across images. Neurons ranked 10–20, which contain many image-specific neurons, show much larger variations. These results indicate that image perturbations primarily affect image-specific neurons, which in turn drive hallucinations. This provides empirical support for using perturbations in VCD as a mechanism to induce hallucinations for controlled analysis.
>
> **Second**, as further suggested by the reviewer, we conducted additional comparison experiments analyzing three neuron groups: Always-active neurons, image-specific neurons, and randomly selected neurons from ranks beyond 20. In these experiments, we set the activations of the selected neurons to zero to directly measure their influence on model predictions and hallucination behavior. As shown in Table 1, the results align with our hypothesis: suppressing Always-active neurons has minimal effect on output despite their high baseline activations, and randomly selected neurons also have negligible influence due to their low relevance. In contrast, perturbing image-specific neurons causes substantial activation shifts and significantly alters model outputs. This highlights that image-specific neurons are the key drivers of input-dependent behavior and hallucination sensitivity, while Always-active neurons contribute little to image-specific reasoning.
>
> **Third**, we have added additional figures and analyses that specifically require fine-grained, localized visual understanding. As shown in the newly included Figures 15 and 16in `Appendix F.3`, the model fails on a simple attribute recognition task by predicting that a black apple is red. Our SAE-based analysis reveals that this error is caused by abnormally strong activation of irrelevant semantic units, which overwhelms the evidence from neurons genuinely associated with apples. By selectively suppressing the overly active neurons or enhancing the apple-specific ones, we can restore the correct prediction. This intervention provides direct causal evidence that inappropriate global activations can dominate over local visual cues and lead to hallucinations, and that neuron-level modulation offers a principled way to recover accurate, visually grounded behavior.
>
> These new analyses more directly lay the foundation for our CNS method, and we sincerely thank the reviewer for prompting this deeper investigation. CNS is built on the principle of identifying image-specific neurons and selectively enhancing them. By comparing neuron activations between original and perturbed images, we can reliably distinguish neurons whose activations decrease under noise (image-specific neurons) from those that remain stable or increase (typically Always-active, content-irrelevant neurons). CNS then enhances the former and suppresses the latter. Moreover, the small but sometimes positive effect of down-weighting Always-active neurons further motivated our ANS strategy, which explicitly suppresses these neurons for additional gains.
>
> In summary, these analyses establish a clearer connection between neuron-level behavior and hallucination patterns, validate the key design choices of CNS, and provide an interpretable empirical basis for selectively modulating neuron activations to mitigate hallucinations. We sincerely thank the reviewer for the insightful suggestion that led to these improvements.
>
> | Neuron Type | Accuracy (%) ↑ | F1-score (%) ↑ |
> | --- | --- | --- |
> | baseline | 84.63 | 84.99 |
> | always-on | 84.68 | 85.08 |
> | image-specific | 63.08 | 57.36 |
> | random | 84.31 | 84.65 |

---

> > ### Comment · Reviewer_MCny · 2025-11-25
> >
> > Thank you for the detailed and thorough revisions. The additional benchmarks and expanded neuron-level analyses adequately address the concerns I previously raised. The manuscript is now clearer and stronger.
> >
> > I have decided to maintain my original score.

---

> > > ### Author Response · Authors · 2025-11-25
> > >
> > > Thank you for your thoughtful feedback and constructive suggestions. We are pleased to hear that the additional benchmarks and expanded analyses adequately addressed your earlier concerns. Your detailed comments have been instrumental in improving the clarity and rigor of the manuscript. We truly appreciate your time and effort in reviewing our work.

---

### Official Review · Reviewer_Q189 · 2025-10-31

**Soundness:** 2
**Presentation:** 3
**Contribution:** 2
**Rating:** 4
**Confidence:** 4

**Summary:**

The paper investigates object hallucinations in MLLM by probing the internal visual encoder representations using SAEs. It claims that hallucinations arise from unstable neuron activations under noise and introduces CNS — a method that amplifies neurons responsive to clean images while suppressing those activated by noisy images, along with “always-on neuron suppression.”

**Strengths:**

Identifies “always-on” neurons that encode global statistics, which is a potential insight for interpretability.

CNS is simple, efficient, and can be applied plug-and-play at inference time.

**Weaknesses:**

Lack of causal rigor: The paper assumes neuron instability causes hallucination but presents only correlational evidence. No controlled interventions or causal mediation analyses are performed.

Weak empirical validation: More benchmarks are need for method generalizability, such as HallusionBench RLHF-V. Also improvements on the reported benchmarks are marginal (<2%).

Novelty: he novelty is marginal, and the intuition about the correlation between attention and hallucination is nearly common sense since OPERA. CNS is conceptually close to VCD (contrast between distorted vs. clean views) but simply moved to feature space.  Since the insight is not new enough, a more elegant way is need for ICLR

**Questions:**

Please see weakness

---

> ### Author Response · Authors · 2025-11-24
> **Response to Q189 (1/n)**
>
> _Thank you for your constructive and thoughtful comments. They were indeed helpful in improving the paper. We take this opportunity to address your concerns:_
>
> _**List of changes in the manuscript:**_
>
> > 1. `Section F.1 in Appendix` **Evaluation on more challenging benchmarks.**
> > 2. `Section F.2 in Appendix` **Evaluation on more methods and models.**
> > 3. `Section F.3 in Appendix` **More cases illustrating the link between hallucinations and neuron activations.**
> > 4. `Figure 5 in Main Paper`**Add the analyses of always-active neurons and image-specific neurons.**
> > 5. `Table 1 in Main Paper` **Effects of suppressing always-active, image-specific, and random neurons.**
> >

---

> ### Author Response · Authors · 2025-11-24
> **Response to Q189 (2/n)**
>
> ## W1：More analyses and experiments about neuron instability and hallucination
> We thank the reviewer for this very insightful suggestion.
>
> Following the reviewer’s suggestion, we implemented three improvements to deepen the analysis of neuron instability and hallucination.
>
> **First**, we refined the original patch-level neuron-change analysis by decomposing activation changes across different neuron categories. Specifically, we separately visualized the activation changes of Always-active neurons and image-specific neurons. Consistent with our earlier findings, the top 10 neurons are predominantly Always-active neurons, while ranks 10–20 contain a mixture of image-specific neurons. Figure 5 shows that increasing noise steps causes LLaVA performance to degrade and simultaneously increases the neuron-change ratio. Patch-level neurons change the fastest, indicating that noise perturbs nearly all token-level neurons, which is consistent with our previous observation that patch-level neurons do not have stable semantic roles and therefore vary significantly under perturbation. In contrast, Always-active neurons exhibit minimal change, reflecting their stability across images. Neurons ranked 10–20, which contain many image-specific neurons, show much larger variations. These results indicate that image perturbations primarily affect image-specific neurons, which in turn drive hallucinations. This provides empirical support for using perturbations in VCD as a mechanism to induce hallucinations for controlled analysis.
>
> **Second**, as further suggested by the reviewer, we conducted additional comparison experiments analyzing three neuron groups: Always-active neurons, image-specific neurons, and randomly selected neurons from ranks beyond 20. In these experiments, we set the activations of the selected neurons to zero to directly measure their influence on model predictions and hallucination behavior. As shown in Table 1, the results align with our hypothesis: suppressing Always-active neurons has minimal effect on output despite their high baseline activations, and randomly selected neurons also have negligible influence due to their low relevance. In contrast, perturbing image-specific neurons causes substantial activation shifts and significantly alters model outputs. This highlights that image-specific neurons are the key drivers of input-dependent behavior and hallucination sensitivity, while Always-active neurons contribute little to image-specific reasoning.
>
> **Third**, we have added additional figures and analyses that specifically require fine-grained, localized visual understanding. As shown in the newly included Figures 15 and 16in `Appendix F.3`, the model fails on a simple attribute recognition task by predicting that a black apple is red. Our SAE-based analysis reveals that this error is caused by abnormally strong activation of irrelevant semantic units, which overwhelms the evidence from neurons genuinely associated with apples. By selectively suppressing the overly active neurons or enhancing the apple-specific ones, we can restore the correct prediction. This intervention provides direct causal evidence that inappropriate global activations can dominate over local visual cues and lead to hallucinations, and that neuron-level modulation offers a principled way to recover accurate, visually grounded behavior.
>
> These new analyses more directly lay the foundation for our CNS method, and we sincerely thank the reviewer for prompting this deeper investigation. CNS is built on the principle of identifying image-specific neurons and selectively enhancing them. By comparing neuron activations between original and perturbed images, we can reliably distinguish neurons whose activations decrease under noise (image-specific neurons) from those that remain stable or increase (typically Always-active, content-irrelevant neurons). CNS then enhances the former and suppresses the latter. Moreover, the small but sometimes positive effect of down-weighting Always-active neurons further motivated our ANS strategy, which explicitly suppresses these neurons for additional gains.
>
> In summary, these analyses establish a clearer connection between neuron-level behavior and hallucination patterns, validate the key design choices of CNS, and provide an interpretable empirical basis for selectively modulating neuron activations to mitigate hallucinations. We sincerely thank the reviewer for the insightful suggestion that led to these improvements.
>
> | Neuron Type | Accuracy (%) ↑ | F1-score (%) ↑ |
> | --- | --- | --- |
> | baseline | 84.63 | 84.99 |
> | always-on | 84.68 | 85.08 |
> | image-specific | 63.08 | 57.36 |
> | random | 84.31 | 84.65 |

---

> ### Author Response · Authors · 2025-11-24
> **Response to Q189 (3/n)**
>
> ## W2. More Experiments on additional benchmarks, methods and models
> We thank the reviewer for raising this concern.  To further verify the **generality and robustness** of CNS, we extend our evaluation to a broader set of **hallucination mitigation methods**, including DoLA, OPERA, VCD, Woodpecker, LURE, HALC, CODE, EAH, VHR, AD-HH, SID, SECOND, and MFCD, as well as diverse **LVLM architectures**, such as MiniGPT-4, mPLUG-Owl2, LLaVA-Next, LLaVA-OneVision, and Qwen2.5-VL. All models are evaluated on both **POPE** and **CHAIR** benchmarks.
>
> As shown in Tables 10, 11, 12, 13, and 14, CNS **consistently improves performance** across different mitigation methods and decoding strategies. In particular, it **reduces hallucination rates** (lower CHAIRS_SS and CHAIRI_II scores) while maintaining or improving captioning quality (higher F1 and BLEU scores). These results demonstrate that CNS is a **generally applicable and robust module** for enhancing the reliability of open-ended visual question answering and image captioning tasks.
>
> We further evaluate CNS on **more challenging benchmarks** to assess its effectiveness in scenarios requiring **precise image-specific feature understanding**:
>
> + **AMBER (Table 16):** Models generate captions in response to “Describe the image.” Evaluation uses four metrics: CHAIR (hallucination detection), Cover (completeness of object coverage), Hal (hallucination rate), and Cog (human-like hallucination patterns). To reduce computational cost, we use a subset of 50 images.
> + **MMHal-Bench (Table 15):** Evaluates attributes, relations, counting, and overall informativeness. An automatic evaluator compares model outputs with human-written references and ground-truth object labels.
> + **HallusionBench (Table 17):** Designed to test nuanced **image-context reasoning**, containing 346 images and 1129 human-crafted questions, challenging advanced LVLMs such as GPT-4V, Gemini Pro Vision, Claude 3, and LLaVA-1.5.
>
> Across these benchmarks, CNS **consistently improves over baseline and other mitigation methods**, showing that targeted neuron-level modulation can **reduce hallucinations**, enhance **semantic and image-specific grounding**, and strengthen reasoning in diverse and complex evaluation scenarios.
>
> More detailed results and analysis can be found in `Appendix F.1 and F.2`.
>
>
>
> **Table R.1: POPE Benchmark – Newer LLaVA-Next and LLaVA-OneVision**
>
> | Model | LLM | Method | MSCOCO F1 ↑ | OKVQA F1 ↑ | GQA F1 ↑ |
> | --- | --- | --- | --- | --- | --- |
> | LLaVA-Next (CLIP-336) | Vicuna-7B | baseline | 86.5 | 88.8 | 86.3 |
> | | | baseline + CNS (ours) | 87.0 | 89.3 | 86.8 |
> | | | VCD | 87.3 | 88.6 | 84.9 |
> | | | SECOND | 87.5 | 89.1 | 86.3 |
> | | | SECOND + CNS (ours) | 88.2 | 90.2 | 87.1 |
> | LLaVA-OneVision (SigLIP-384) | Qwen2-0.5B | baseline | 87.4 | 88.7 | 86.3 |
> | | | baseline + CNS (ours) | 87.9 | 89.2 | 86.8 |
> | | | VCD | 86.4 | 88.9 | 86.4 |
> | | | SECOND | 86.3 | 88.1 | 86.7 |
> | | | SECOND + CNS (ours) | 87.1 | 89.6 | 87.6 |
>
>
> ---
>
> **Table R.2: POPE Benchmark – Newer Qwen2.5-VL**
>
> | Model | Decoding | Random F1 ↑ | Popular F1 ↑ | Adversarial F1 ↑ |
> | --- | --- | --- | --- | --- |
> | Qwen2.5-VL | Sample (base) | 80.03 | 78.93 | 80.03 |
> | | Sample + CNS (ours) | 80.61 | 79.52 | 80.58 |
> | | Dola | 77.46 | 77.43 | 77.47 |
> | | VCD | 81.39 | 79.91 | 79.98 |
> | | SID | 79.95 | 79.38 | 78.82 |
> | | MFCD (ours) | 82.91 | 82.01 | 81.75 |
> | | MFCD + CNS (ours) | 83.45 | 82.54 | 82.29 |
>
>
> ---
>
> **Table R.3: CHAIR Benchmark – Newer LLaVA-Next**
>
> | Method | CHAIR(_S) ↓ | CHAIR(_I) ↓ |
> | --- | --- | --- |
> | Greedy | 29.08 ± 2.09 | 8.08 ± 0.74 |
> | Greedy + CNS (ours) | 28.22 ± 1.82 | 7.26 ± 0.56 |
> | DoLa | 28.76 ± 2.58 | 8.12 ± 0.78 |
> | VCD | 30.80 ± 2.48 | 8.72 ± 0.94 |
> | CODE | 27.84 ± 2.73 | 7.98 ± 0.92 |
> | EAH | 28.13 ± 1.13 | 6.62 ± 0.49 |
> | VHR | 24.96 ± 2.09 | 6.80 ± 0.59 |
> | VHR + CNS (ours) | 24.42 ± 1.68 | 6.28 ± 0.36 |

---

> ### Author Response · Authors · 2025-11-24
> **Response to Q189 (4/n)**
>
> ---
>
> **Table R.4: CHAIR – COCO and Nocaps (AD-HH Protocol)**
>
> | Methods | COCO LLaVA-7B CHAIR(_S) ↓ | CHAIR(_I) ↓ | COCO MiniGPT-4 CHAIR(_S) ↓ | CHAIR(_I) ↓ | Nocaps LLaVA-7B CHAIR(_S) ↓ | CHAIR(_I) ↓ | Nocaps MiniGPT-4 CHAIR(_S) ↓ | CHAIR(_I) ↓ |
> | --- | --- | --- | --- | --- | --- | --- | --- | --- |
> | Greedy | 51.8 | 13.3 | 40.6 | 13.7 | 43.2 | 14.3 | 57.4 | 20.0 |
> | Greedy + CNS (ours) | 49.6 | 12.2 | 40.4 | 13.2 | 39.6 | 13.4 | 53.8 | 18.2 |
> | DoLA | 53.8 | 13.9 | 41.0 | 13.8 | 42.0 | 13.7 | 57.2 | 20.4 |
> | OPERA | 50.2 | 14.5 | 35.2 | 12.8 | 44.2 | 14.4 | 46.2 | 16.2 |
> | VCD | 55.4 | 15.7 | 38.8 | 14.8 | 43.6 | 14.4 | 48.2 | 17.5 |
> | LURE | 51.2 | 13.4 | 46.4 | 14.2 | 41.8 | 14.4 | 55.8 | 19.6 |
> | HALC | 50.2 | 12.4 | 36.4 | 11.8 | 40.2 | 12.2 | 53.0 | 18.0 |
> | AD-HH | 29.6 | 8.0 | 35.2 | 11.7 | 35.6 | 9.4 | 46.8 | 16.2 |
> | AD-HH + CNS (ours) | 28.9 | 7.8 | 34.6 | 11.3 | 35.1 | 9.1 | 46.2 | 15.6 |
>
>
> ---
>
> **Table R.5: CHAIR Benchmark – HALC Setup**
>
> | Method | MiniGPT-4 CHAIR(_S) ↓ | CHAIR(_I) ↓ | BLEU ↑ | LLaVA-1.5 CHAIR(_S) ↓ | CHAIR(_I) ↓ | BLEU ↑ | mPLUG-Owl2 CHAIR(_S) ↓ | CHAIR(_I) ↓ | BLEU ↑ |
> | --- | --- | --- | --- | --- | --- | --- | --- | --- | --- |
> | Greedy | 30.87 ± 5.45 | 12.33 ± 2.07 | 14.33 ± 0.00 | 20.80 ± 0.08 | 6.77 ± 0.07 | 15.93 ± 0.00 | 23.20 ± 0.35 | 8.33 ± 0.28 | 15.37 ± 0.00 |
> | Greedy + CNS (ours) | 30.52 ± 3.82 | 12.05 ± 1.65 | 14.58 ± 0.00 | 20.42 ± 0.07 | 6.55 ± 0.06 | 16.28 ± 0.00 | 22.85 ± 0.28 | 8.08 ± 0.21 | 15.69 ± 0.00 |
> | DoLA | 30.87 ± 2.52 | 11.70 ± 0.13 | 14.93 ± 0.00 | 21.00 ± 0.67 | 6.70 ± 0.38 | 15.93 ± 0.00 | 24.60 ± 0.24 | 8.73 ± 0.30 | 15.40 ± 0.00 |
> | OPERA | 30.00 ± 0.43 | 11.67 ± 0.22 | 14.87 ± 0.00 | 21.13 ± 0.12 | 6.73 ± 0.18 | 16.27 ± 0.01 | 22.13 ± 0.86 | 7.57 ± 0.16 | 15.53 ± 0.00 |
> | Woodpecker | 28.87 ± 2.20 | 10.20 ± 0.85 | 15.30 ± 0.01 | 23.85 ± 4.62 | 7.50 ± 0.01 | 17.05 ± 0.00 | 26.33 ± 1.98 | 8.43 ± 0.80 | 16.43 ± 0.00 |
> | LURE | 27.88 ± 2.25 | 10.20 ± 0.85 | 15.03 ± 0.11 | 19.48 ± 2.35 | 6.50 ± 0.38 | 15.97 ± 0.01 | 21.27 ± 0.06 | 7.67 ± 0.16 | 15.65 ± 0.05 |
> | VCD | 30.27 ± 0.44 | 12.60 ± 0.45 | 14.33 ± 0.00 | 23.33 ± 5.66 | 7.90 ± 0.53 | 14.67 ± 0.01 | 27.27 ± 7.32 | 9.73 ± 1.22 | 14.40 ± 0.00 |
> | HALC | 17.80 ± 0.03 | 8.10 ± 0.14 | 14.91 ± 0.00 | 13.80 ± 0.08 | 5.50 ± 0.14 | 16.10 ± 0.01 | 17.33 ± 4.30 | 7.43 ± 0.11 | 16.27 ± 0.00 |
> | HALC + CNS (ours) | 17.35 ± 0.02 | 7.70 ± 0.12 | 15.25 ± 0.00 | 13.30 ± 0.07 | 5.15 ± 0.12 | 16.45 ± 0.01 | 16.70 ± 3.80 | 7.05 ± 0.09 | 16.65 ± 0.00 |
>
>
> ---
>
> **Table R.6: MMHal-Bench**
>
> | Method | Average Score ↑ | Hallucination Rate ↓ |
> | --- | --- | --- |
> | baseline | 1.86 | 63.5 |
> | baseline + CNS (ours) | 2.09 | 54.8 |
> | VCD | 2.12 | 54.2 |
> | VCD + CNS (ours) | 2.28 | 53.6 |
> | OPERA | 2.33 | 50.0 |
> | Less-is-more | 2.15 | 54.2 |
> | VACoDe | 2.13 | 54.4 |
>
>
> ---
>
> **Table R.7: AMBER Benchmark**
>
> | Method | CHAIR ↓ | Cover ↑ | Hal ↓ | Cog ↓ |
> | --- | --- | --- | --- | --- |
> | Regular | 7.8 | 51.0 | 36.4 | 4.2 |
> | Regular + CNS (ours) | 7.2 | 51.8 | 34.1 | 3.8 |
> | VCD | 7.5 | 50.8 | 36.2 | 4.1 |
> | VCD + CNS (ours) | 7.1 | 51.6 | 33.2 | 3.4 |
> | OPERA | 7.3 | 49.6 | 32.0 | 3.5 |
> | DoLA | 7.6 | 51.6 | 36.0 | 4.0 |
> | Woodpecker | 6.9 | 48.9 | 30.4 | 3.6 |
> | M3ID | 7.4 | 49.9 | 33.2 | 3.7 |
>
>
> ---
>
> **Table R.8: HallusionBench**
>
> | Model | Q. Pair Acc ↑ | Figure Acc ↑ | Easy Q. Acc ↑ | Hard Q. Acc ↑ | Question Acc ↑ |
> | --- | --- | --- | --- | --- | --- |
> | LLaVA-1.5 (GPT Eval) | 10.55 | 24.86 | 49.67 | 29.77 | 46.94 |
> | + VCD | 10.92 | 25.13 | 49.88 | 30.05 | 47.12 |
> | + CNS (ours) | 11.10 | 25.30 | 50.02 | 30.21 | 47.25 |
> | + VCD + CNS (ours) | 11.35 | 25.58 | 50.19 | 30.44 | 47.38 |

---

> ### Author Response · Authors · 2025-11-24
> **Response to Q189 (5/n)**
>
> ## W3: Novelty
> We would like to clarify a misunderstanding regarding the novelty. Our work **does not analyze attention mechanisms** or their correlation with hallucinations. Instead, we focus on internal **feature-level and neuron-level dynamics**, using sparse autoencoders (SAEs) to study the activation of interpretable sparse features. CNS operates by **enhancing or suppressing specific internal components** in this feature space to influence model outputs, which is conceptually distinct from attention-based approaches such as OPERA. This internal neuron-level intervention perspective provides a **complementary and mechanistic understanding of hallucinations**, offering a new direction that is not captured by prior attention-focused analyses.
>
> In comparison with VCD, which require **two full LVLM forward passes** during decoding because logits from the original and perturbed images must be compared **at each decoding step**, CNS is far more efficient. CNS performs contrastive analysis only once during **the prefill stage of the image encoder**, **with no additional decoding pass needed**. Decoding afterward proceeds identically to standard inference. **CNS can be combined with existing mitigation strategies**, further enhancing output reliability and visual grounding while maintaining efficiency and broad applicability.
>
> In summary, our work leverages SAE to analyze and explain hallucination mechanisms in LVLMs, and introduces the lightweight, elegant CNS method to mitigate hallucinations.
>
>
>
> Several reviewers recognized the novelty and potential of our paper:
>
> + `Q189`:Identifies ‘always-on’ neurons that encode global statistics, which is **a potential insight for interpretability.**
> + `yV54`:“This paper applies SAE-based interpretability techniques to vision-language models. **This is a promising research direction for mechanistic interpretability in multimodal models**.”
> + `MCny`: **Novel interpretability perspective**. The paper offers an insightful neuron-level analysis of LVLM hallucinations, revealing how perturbations reshape visual semantics. This represents a clear advancement over purely decoding-based methods. & **Simple yet elegant approach**. The method is conceptually intuitive, leveraging noisy–clean contrasts to steer neuron activations and does not require retraining.
> + `Ttv4`:  The analysis section effectively reveals a clear relationship between noise strength, model performance, and neuron change ratio, **providing valuable insights into the model’s internal dynamics**.   & **The approach is relatively lightweight,** suggesting that it can be applied efficiently without excessive computational overhead.

---

### Official Review · Reviewer_yV54 · 2025-11-02

**Soundness:** 2
**Presentation:** 2
**Contribution:** 2
**Rating:** 2
**Confidence:** 4

**Summary:**

This paper introduce SAEs to interpret and steer the internal visual representations of LVLMs. They find that neuron-level interventions can modulate LVLM outputs for targeted concepts. Then they propose CNS to amplifies meaningful neurons while suppressing perturbation induced activations for hallucination mitigation.

**Strengths:**

1. This paper tackles visual hallucinations in LVLMs, a important problem that directly impacts the trustworthiness and safety of multimodal AI systems.

2. This paper applys SAE-based interpretability techniques to vision-language models. This is a promising research directions for mechanistic interpretability in multimodal models.

**Weaknesses:**

1.  Flawed Logic in "Neuron-Level Statistical Analysis".
- It is a well-established phenomenon in representation learning that some neurons naturally encode global features (e.g., color, texture, brightness) that are present across diverse images. The observation seems trivial.
- The authors claim that always-active neurons (appearing in top-20 activations across all images) encode "over-generalized pseudo-global concepts" and are therefore detrimental. However, this claim lacks both theoretical justification and empirical evidence.

2. Flawed Logic in “Diagnosing Hallucinations with SAEs.”
- The part fails to establish a causal link with the former part "Neuron-Level Statistical Analysis".
- The current observation that "perturbing neurons increases hallucinations" is trivial without comparing.
- The diagnosis methodology lacks critical baselines: Always-active neurons vs. image-specific neurons vs. randomly selected neurons
- No experimental setup is provided for this part. How is hallucination rate measured? No metrics or protocols is described.

3. Limited Experimental Validation and Insufficient Baselines.
- The proposed method shows modest gains across benchmarks. From Table1, Table 2, and Table 4, there is only ~1% improvement on average. Given the added computational cost (generating perturbed images at inference time, SAE encoding/decoding), the improvements do not convincingly demonstrate the method's effectiveness.
- The paper omits several state-of-the-art hallucination mitigation methods. Missing reference and comparisons:
OPERA (Huang et al., 2023), HALC (Chen et al., 2024),  ADHH (Yang et al., 2025)

OPERA: Alleviating Hallucination in Multi-Modal Large Language Models via Over-Trust Penalty and Retrospection-Allocation. CVPR 2023

HALC: Object Hallucination Reduction via Adaptive Focal-Contrast Decoding. ICML 2024.

Understanding and Mitigating Hallucinations in Large Vision-Language Models via Modular Attribution and Intervention. ICLR 2025.

**Questions:**

1. Why always-active neurons (appearing in top-20 activations across all images) encode "over-generalized pseudo-global concepts"? Is there theoretical justification and empirical evidence?

2. What is the causal link between "Neuron-Level Statistical Analysis" and "Diagnosing Hallucinations with SAEs"? What is the experimental setup for this part? Which dataset and model is used? How is hallucination rate measured?

---

> ### Author Response · Authors · 2025-11-24
> **Response to yV54 (1/n)**
>
> _Thank you for your constructive and thoughtful comments. They were indeed helpful in improving the paper. We take this opportunity to address your concerns:_
>
> _**List of changes in the manuscript:**_
>
> > 1. `Section F.1 in Appendix` **Evaluation on more challenging benchmarks.**
> > 2. `Section F.2 in Appendix` **Evaluation on more methods and models.**
> > 3. `Section F.3 in Appendix` **More cases illustrating the link between hallucinations and neuron activations.**
> > 4. `Figure 5 in Main Paper` **Add the analyses of always-active neurons and image-specific neurons.**
> > 5. `Table 1 in Main Paper` **Effects of suppressing always-active, image-specific, and random neurons.**
> > 6. `Lines 248-255 in Main Paper` **We have added the detailed experimental setup.**
>
> ## W1: More discussion about "Neuron-Level Statistical Analysis"
> A:
>
> These neurons are not inherently harmful, but their activation magnitudes—ranging from 10 to 80—are substantially higher than those of most top-40 neurons, which fall in the 5–15 range. Such disproportionately high activations may reduce the model’s sensitivity to image-specific features. To investigate this, we conducted additional comparison experiments analyzing three neuron groups: Always-active neurons, image-specific neurons, and randomly selected neurons ranked beyond 20. In these experiments, we set the activations of the selected neurons to zero to directly measure their influence on model predictions and hallucination behavior.
>
> As shown in Table 1, the results support our hypothesis. Suppressing Always-active neurons has minimal impact on outputs despite their high baseline activations, and randomly selected neurons also exert negligible influence due to low relevance. In contrast, perturbing image-specific neurons induces substantial activation shifts and significantly alters model predictions. This highlights that image-specific neurons are the primary drivers of input-dependent behavior and hallucination sensitivity, whereas Always-active neurons contribute little to image-specific reasoning. Furthermore, appropriately modulating these neurons can improve overall accuracy and F1 score. Our ablation studies (Table 4) further confirm that suppressing globally active neurons consistently enhances model performance.
>
> We also added evaluations on benchmarks beyond POPE, including AMBER (Table 16), MMHal-Bench (Table 15), CHAIR (Table 14), and HallusionBench (Table 17) in `Appendix F.1 and F.2` all of which explicitly require image-specific reasoning and fine-grained semantic understanding. Across these benchmarks, our method shows clearer and more consistent improvements, supporting our claim that reducing overly general global activations strengthens the model’s ability to capture image-specific semantics. As the reviewer noted, POPE is heavily dominated by global signals and therefore may not fully expose the drawbacks of ignoring such fine-grained or image-dependent cues.
>
> We agree that certain neurons naturally encode global features that appear across many images. Our intention was not to claim the existence of such neurons as a new finding, but to show through SAE-based decomposition that there exists a specific subset whose activations are unusually strong, unusually stable, and present across nearly all inputs. SAE enables us to isolate these neurons in an interpretable way, providing a clearer and more intuitive characterization of this phenomenon and offering an additional piece of evidence beyond standard observations in representation learning.  Moreover, SAE can be further applied to interpretability analysis within LVLMs and to mitigating hallucinations, as suppressing or adjusting internal neurons directly improves visual semantic understanding and reduces spurious outputs.
>
> We sincerely appreciate this feedback. The revised version now provides a clearer explanation and stronger empirical evidence supporting our observations.
>
> | Neuron Type | Accuracy (%) ↑ | F1-score (%) ↑ |
> | --- | --- | --- |
> | baseline | 84.63 | 84.99 |
> | always-on | 84.68 | 85.08 |
> | image-specific | 63.08 | 57.36 |
> | random | 84.31 | 84.65 |

---

> ### Author Response · Authors · 2025-11-24
> **Response to yV54 (2/n)**
>
> ## W2 and Q2: More discussion about "Diagnosing Hallucinations with SAEs”
> We thank the reviewer for this very insightful suggestion.
>
> Following the reviewer’s suggestion, we implemented three improvements to deepen the analysis.
>
> **First**, we refined the original patch-level neuron-change analysis by decomposing activation changes across different neuron categories. Specifically, we separately visualized the activation changes of Always-active neurons and image-specific neurons. Consistent with our earlier findings, the top 10 neurons are predominantly Always-active neurons, while ranks 10–20 contain a mixture of image-specific neurons. Figure 5 shows that increasing noise steps causes LLaVA performance to degrade and simultaneously increases the neuron-change ratio. Patch-level neurons change the fastest, indicating that noise perturbs nearly all token-level neurons, which is consistent with our previous observation that patch-level neurons do not have stable semantic roles and therefore vary significantly under perturbation. In contrast, Always-active neurons exhibit minimal change, reflecting their stability across images. Neurons ranked 10–20, which contain many image-specific neurons, show much larger variations. These results indicate that image perturbations primarily affect image-specific neurons, which in turn drive hallucinations. This provides empirical support for using perturbations in VCD as a mechanism to induce hallucinations for controlled analysis.
>
> **Second**, as further suggested by the reviewer, we conducted additional comparison experiments analyzing three neuron groups: Always-active neurons, image-specific neurons, and randomly selected neurons from ranks beyond 20. In these experiments, we set the activations of the selected neurons to zero to directly measure their influence on model predictions and hallucination behavior. As shown in Table 1, the results align with our hypothesis: suppressing Always-active neurons has minimal effect on output despite their high baseline activations, and randomly selected neurons also have negligible influence due to their low relevance. In contrast, perturbing image-specific neurons causes substantial activation shifts and significantly alters model outputs. This highlights that image-specific neurons are the key drivers of input-dependent behavior and hallucination sensitivity, while Always-active neurons contribute little to image-specific reasoning.
>
> **Third**, we have added additional figures and analyses that specifically require fine-grained, localized visual understanding. As shown in the newly included Figures 15 and 16in `Appendix F.3`, the model fails on a simple attribute recognition task by predicting that a black apple is red. Our SAE-based analysis reveals that this error is caused by abnormally strong activation of irrelevant semantic units, which overwhelms the evidence from neurons genuinely associated with apples. By selectively suppressing the overly active neurons or enhancing the apple-specific ones, we can restore the correct prediction. This intervention provides direct causal evidence that inappropriate global activations can dominate over local visual cues and lead to hallucinations, and that neuron-level modulation offers a principled way to recover accurate, visually grounded behavior.
>
> These new analyses more directly lay the foundation for our CNS method, and we sincerely thank the reviewer for prompting this deeper investigation. CNS is built on the principle of identifying image-specific neurons and selectively enhancing them. By comparing neuron activations between original and perturbed images, we can reliably distinguish neurons whose activations decrease under noise (image-specific neurons) from those that remain stable or increase (typically Always-active, content-irrelevant neurons). CNS then enhances the former and suppresses the latter. Moreover, the small but sometimes positive effect of down-weighting Always-active neurons further motivated our ANS strategy, which explicitly suppresses these neurons for additional gains.
>
> In summary, these analyses establish a clearer connection between neuron-level behavior and hallucination patterns, validate the key design choices of CNS, and provide an interpretable empirical basis for selectively modulating neuron activations to mitigate hallucinations. We sincerely thank the reviewer for the insightful suggestion that led to these improvements.
>
> | Neuron Type | Accuracy (%) ↑ | F1-score (%) ↑ |
> | --- | --- | --- |
> | baseline | 84.63 | 84.99 |
> | always-on | 84.68 | 85.08 |
> | image-specific | 63.08 | 57.36 |
> | random | 84.31 | 84.65 |

---

> ### Author Response · Authors · 2025-11-24
> **Response to yV54 (3/n)**
>
> **Experimental setup**
>
> We have added the experimental setup to the main text. To clarify, all experiments are conducted using **LLaVA-1.5** on the **POPE benchmark (COCO, random setup)**. Hallucination effects are quantified using **F1 and accuracy metrics** defined by the benchmark. We apply controlled image perturbations (e.g., additive noise) and measure their impact on model performance in image-specific reasoning tasks. The **SAE latent space decomposition** is used to analyze neuron-level responses to these perturbations, allowing us to identify which neurons are most sensitive and how changes in their activation correlate with hallucination occurrence.
>
> We further examine how model performance is affected by selectively zeroing out different types of SAE neurons. Based on our earlier analysis, we roughly categorize SAE neurons into three groups: top-ranked always-active neurons (top-10), image-specific neurons (primarily within ranks 10–20), and ten randomly selected neurons from ranks beyond 20. This allows us to investigate the distinct characteristics of each neuron group and their respective impacts on model behavior. This setup provides a consistent and quantitative framework linking neuron-level behavior to hallucination patterns.

---

> ### Author Response · Authors · 2025-11-24
> **Response to yV54 (4/n)**
>
> ## W3. More Experiments on additional benchmarks, methods and models
> We thank the reviewer for raising this concern. We address it in two aspects: **computational overhead** and **evaluation across more benchmarks and methods**.
>
> ---
>
> **1. Computational Overhead**
>
> We clarify that, despite involving an additional perturbed image, CNS introduces **very low computational overhead** compared to existing contrastive-decoding-based hallucination mitigation approaches.
>
> + **Comparison with VCD:** Contrastive Decoding Methods like VCD require **two full LVLM forward passes** during decoding, as logits from the original and perturbed images must be compared at each decoding step. This increases latency and limits the use of some acceleration techniques.
> + **CNS efficiency:** In contrast, CNS performs contrastive analysis **only once during the prefill stage**, before generation begins. The perturbed image is processed solely through the **image encoder**, with no second LVLM decoding path required. Decoding afterward proceeds identically to standard inference and remains fully compatible with acceleration strategies.
> + **Lightweight SAE module:** Our **SAE** consists of only **two linear layers**, whose computational cost is negligible relative to the LVLM backbone. Therefore, CNS introduces minimal overhead and does not significantly affect inference time.
> + **Compatibility with other methods:** When integrated with contrastive decoding methods such as VCD, CNS only adds a single SAE pass on the original image while keeping a single-branch decoding process, resulting in almost no additional cost but providing almost **“free gains”** in accuracy.
>
> ---
>
> **2. Evaluation Across More Benchmarks and Methods**
>
> To further verify the **generality and robustness** of CNS, we extend our evaluation to a broader set of **hallucination mitigation methods**, including DoLA, OPERA, VCD, Woodpecker, LURE, HALC, CODE, EAH, VHR, AD-HH, SID, SECOND, and MFCD, as well as diverse **LVLM architectures**, such as MiniGPT-4, mPLUG-Owl2, LLaVA-Next, LLaVA-OneVision, and Qwen2.5-VL. All models are evaluated on both **POPE** and **CHAIR** benchmarks.
>
> As shown in Tables 10, 11, 12, 13, and 14, CNS **consistently improves performance** across different mitigation methods and decoding strategies. In particular, it **reduces hallucination rates** (lower CHAIRS_SS and CHAIRI_II scores) while maintaining or improving captioning quality (higher F1 and BLEU scores). These results demonstrate that CNS is a **generally applicable and robust module** for enhancing the reliability of open-ended visual question answering and image captioning tasks.
>
> We further evaluate CNS on **more challenging benchmarks** to assess its effectiveness in scenarios requiring **precise image-specific feature understanding**:
>
> + **AMBER (Table 16):** Models generate captions in response to “Describe the image.” Evaluation uses four metrics: CHAIR (hallucination detection), Cover (completeness of object coverage), Hal (hallucination rate), and Cog (human-like hallucination patterns).
> + **MMHal-Bench (Table 15):** Evaluates attributes, relations, counting, and overall informativeness. An automatic evaluator compares model outputs with human-written references and ground-truth object labels.
> + **HallusionBench (Table 17):** Designed to test nuanced **image-context reasoning**, containing 346 images and 1129 human-crafted questions.
>
> Across these benchmarks, CNS **consistently improves over baseline and other mitigation methods**, showing that targeted neuron-level modulation can **reduce hallucinations**, enhance **semantic and image-specific grounding**, and strengthen reasoning in diverse and complex evaluation scenarios.
>
> More detailed results and analysis can be found in `Appendix F.1 and F.2`.

---

> ### Author Response · Authors · 2025-11-24
> **Response to yV54 (5/n)**
>
> **Table R.1: POPE Benchmark – Newer LLaVA-Next and LLaVA-OneVision**
>
> | Model | LLM | Method | MSCOCO F1 ↑ | OKVQA F1 ↑ | GQA F1 ↑ |
> | --- | --- | --- | --- | --- | --- |
> | LLaVA-Next (CLIP-336) | Vicuna-7B | baseline | 86.5 | 88.8 | 86.3 |
> | | | baseline + CNS (ours) | 87.0 | 89.3 | 86.8 |
> | | | VCD | 87.3 | 88.6 | 84.9 |
> | | | SECOND | 87.5 | 89.1 | 86.3 |
> | | | SECOND + CNS (ours) | 88.2 | 90.2 | 87.1 |
> | LLaVA-OneVision (SigLIP-384) | Qwen2-0.5B | baseline | 87.4 | 88.7 | 86.3 |
> | | | baseline + CNS (ours) | 87.9 | 89.2 | 86.8 |
> | | | VCD | 86.4 | 88.9 | 86.4 |
> | | | SECOND | 86.3 | 88.1 | 86.7 |
> | | | SECOND + CNS (ours) | 87.1 | 89.6 | 87.6 |
>
>
> ---
>
> **Table R.2: POPE Benchmark – Newer Qwen2.5-VL**
>
> | Model | Decoding | Random F1 ↑ | Popular F1 ↑ | Adversarial F1 ↑ |
> | --- | --- | --- | --- | --- |
> | Qwen2.5-VL | Sample (base) | 80.03 | 78.93 | 80.03 |
> | | Sample + CNS (ours) | 80.61 | 79.52 | 80.58 |
> | | Dola | 77.46 | 77.43 | 77.47 |
> | | VCD | 81.39 | 79.91 | 79.98 |
> | | SID | 79.95 | 79.38 | 78.82 |
> | | MFCD (ours) | 82.91 | 82.01 | 81.75 |
> | | MFCD + CNS (ours) | 83.45 | 82.54 | 82.29 |
>
>
> ---
>
> **Table R.3: CHAIR Benchmark – Newer LLaVA-Next**
>
> | Method | CHAIR(_S) ↓ | CHAIR(_I) ↓ |
> | --- | --- | --- |
> | Greedy | 29.08 ± 2.09 | 8.08 ± 0.74 |
> | Greedy + CNS (ours) | 28.22 ± 1.82 | 7.26 ± 0.56 |
> | DoLa | 28.76 ± 2.58 | 8.12 ± 0.78 |
> | VCD | 30.80 ± 2.48 | 8.72 ± 0.94 |
> | CODE | 27.84 ± 2.73 | 7.98 ± 0.92 |
> | EAH | 28.13 ± 1.13 | 6.62 ± 0.49 |
> | VHR | 24.96 ± 2.09 | 6.80 ± 0.59 |
> | VHR + CNS (ours) | 24.42 ± 1.68 | 6.28 ± 0.36 |
>
>
> ---
>
> **Table R.4: CHAIR – COCO and Nocaps (AD-HH Protocol)**
>
> | Methods | COCO LLaVA-7B CHAIR(_S) ↓ | CHAIR(_I) ↓ | COCO MiniGPT-4 CHAIR(_S) ↓ | CHAIR(_I) ↓ | Nocaps LLaVA-7B CHAIR(_S) ↓ | CHAIR(_I) ↓ | Nocaps MiniGPT-4 CHAIR(_S) ↓ | CHAIR(_I) ↓ |
> | --- | --- | --- | --- | --- | --- | --- | --- | --- |
> | Greedy | 51.8 | 13.3 | 40.6 | 13.7 | 43.2 | 14.3 | 57.4 | 20.0 |
> | Greedy + CNS (ours) | 49.6 | 12.2 | 40.4 | 13.2 | 39.6 | 13.4 | 53.8 | 18.2 |
> | DoLA | 53.8 | 13.9 | 41.0 | 13.8 | 42.0 | 13.7 | 57.2 | 20.4 |
> | OPERA | 50.2 | 14.5 | 35.2 | 12.8 | 44.2 | 14.4 | 46.2 | 16.2 |
> | VCD | 55.4 | 15.7 | 38.8 | 14.8 | 43.6 | 14.4 | 48.2 | 17.5 |
> | LURE | 51.2 | 13.4 | 46.4 | 14.2 | 41.8 | 14.4 | 55.8 | 19.6 |
> | HALC | 50.2 | 12.4 | 36.4 | 11.8 | 40.2 | 12.2 | 53.0 | 18.0 |
> | AD-HH | 29.6 | 8.0 | 35.2 | 11.7 | 35.6 | 9.4 | 46.8 | 16.2 |
> | AD-HH + CNS (ours) | 28.9 | 7.8 | 34.6 | 11.3 | 35.1 | 9.1 | 46.2 | 15.6 |
>
>
> ---
> **Table R.5: CHAIR Benchmark – HALC Setup**
>
> | Method | MiniGPT-4 CHAIR(_S) ↓ | CHAIR(_I) ↓ | BLEU ↑ | LLaVA-1.5 CHAIR(_S) ↓ | CHAIR(_I) ↓ | BLEU ↑ | mPLUG-Owl2 CHAIR(_S) ↓ | CHAIR(_I) ↓ | BLEU ↑ |
> | --- | --- | --- | --- | --- | --- | --- | --- | --- | --- |
> | Greedy | 30.87 ± 5.45 | 12.33 ± 2.07 | 14.33 ± 0.00 | 20.80 ± 0.08 | 6.77 ± 0.07 | 15.93 ± 0.00 | 23.20 ± 0.35 | 8.33 ± 0.28 | 15.37 ± 0.00 |
> | Greedy + CNS (ours) | 30.52 ± 3.82 | 12.05 ± 1.65 | 14.58 ± 0.00 | 20.42 ± 0.07 | 6.55 ± 0.06 | 16.28 ± 0.00 | 22.85 ± 0.28 | 8.08 ± 0.21 | 15.69 ± 0.00 |
> | DoLA | 30.87 ± 2.52 | 11.70 ± 0.13 | 14.93 ± 0.00 | 21.00 ± 0.67 | 6.70 ± 0.38 | 15.93 ± 0.00 | 24.60 ± 0.24 | 8.73 ± 0.30 | 15.40 ± 0.00 |
> | OPERA | 30.00 ± 0.43 | 11.67 ± 0.22 | 14.87 ± 0.00 | 21.13 ± 0.12 | 6.73 ± 0.18 | 16.27 ± 0.01 | 22.13 ± 0.86 | 7.57 ± 0.16 | 15.53 ± 0.00 |
> | Woodpecker | 28.87 ± 2.20 | 10.20 ± 0.85 | 15.30 ± 0.01 | 23.85 ± 4.62 | 7.50 ± 0.01 | 17.05 ± 0.00 | 26.33 ± 1.98 | 8.43 ± 0.80 | 16.43 ± 0.00 |
> | LURE | 27.88 ± 2.25 | 10.20 ± 0.85 | 15.03 ± 0.11 | 19.48 ± 2.35 | 6.50 ± 0.38 | 15.97 ± 0.01 | 21.27 ± 0.06 | 7.67 ± 0.16 | 15.65 ± 0.05 |
> | VCD | 30.27 ± 0.44 | 12.60 ± 0.45 | 14.33 ± 0.00 | 23.33 ± 5.66 | 7.90 ± 0.53 | 14.67 ± 0.01 | 27.27 ± 7.32 | 9.73 ± 1.22 | 14.40 ± 0.00 |
> | HALC | 17.80 ± 0.03 | 8.10 ± 0.14 | 14.91 ± 0.00 | 13.80 ± 0.08 | 5.50 ± 0.14 | 16.10 ± 0.01 | 17.33 ± 4.30 | 7.43 ± 0.11 | 16.27 ± 0.00 |
> | HALC + CNS (ours) | 17.35 ± 0.02 | 7.70 ± 0.12 | 15.25 ± 0.00 | 13.30 ± 0.07 | 5.15 ± 0.12 | 16.45 ± 0.01 | 16.70 ± 3.80 | 7.05 ± 0.09 | 16.65 ± 0.00 |

---

> ### Author Response · Authors · 2025-11-24
> **Response to yV54 (6/n)**
>
> ---
>
> **Table R.6: MMHal-Bench**
>
> | Method | Average Score ↑ | Hallucination Rate ↓ |
> | --- | --- | --- |
> | baseline | 1.86 | 63.5 |
> | baseline + CNS (ours) | 2.09 | 54.8 |
> | VCD | 2.12 | 54.2 |
> | VCD + CNS (ours) | 2.28 | 53.6 |
> | OPERA | 2.33 | 50.0 |
> | Less-is-more | 2.15 | 54.2 |
> | VACoDe | 2.13 | 54.4 |
>
>
> ---
>
> **Table R.7: AMBER Benchmark**
>
> | Method | CHAIR ↓ | Cover ↑ | Hal ↓ | Cog ↓ |
> | --- | --- | --- | --- | --- |
> | Regular | 7.8 | 51.0 | 36.4 | 4.2 |
> | Regular + CNS (ours) | 7.2 | 51.8 | 34.1 | 3.8 |
> | VCD | 7.5 | 50.8 | 36.2 | 4.1 |
> | VCD + CNS (ours) | 7.1 | 51.6 | 33.2 | 3.4 |
> | OPERA | 7.3 | 49.6 | 32.0 | 3.5 |
> | DoLA | 7.6 | 51.6 | 36.0 | 4.0 |
> | Woodpecker | 6.9 | 48.9 | 30.4 | 3.6 |
> | M3ID | 7.4 | 49.9 | 33.2 | 3.7 |
>
>
> ---
>
> **Table R.8: HallusionBench**
>
> | Model | Q. Pair Acc ↑ | Figure Acc ↑ | Easy Q. Acc ↑ | Hard Q. Acc ↑ | Question Acc ↑ |
> | --- | --- | --- | --- | --- | --- |
> | LLaVA-1.5 (GPT Eval) | 10.55 | 24.86 | 49.67 | 29.77 | 46.94 |
> | + VCD | 10.92 | 25.13 | 49.88 | 30.05 | 47.12 |
> | + CNS (ours) | 11.10 | 25.30 | 50.02 | 30.21 | 47.25 |
> | + VCD + CNS (ours) | 11.35 | 25.58 | 50.19 | 30.44 | 47.38 |

---

### Author Response · Authors · 2025-11-25
**Overall Response**

# Overall Response
We would like to thank all of the reviewers for their constructive and valuable feedback on our work!
We will incorporate the relevant discussions into the paper to further improve clarity and completeness.

In this post:

+ (1) We furthermore summarize the strengths of our paper from the reviewers.
+ (2) We summarize the changes to the updated PDF document.

**In the individual replies**, we address other comments.

**- (1) **_**Strengths of Our Paper**_** -**

+ **Novel and insightful interpretability perspective**
    - `MCny`: "The paper offers an insightful neuron-level analysis of LVLM hallucinations, revealing how perturbations reshape visual semantics. This represents a clear advancement over purely decoding-based methods."
    - `Q189`: "Identifies ‘always-on’ neurons that encode global statistics, which is a potential insight for interpretability."
    - `Ttv4`: "The analysis effectively reveals the relationship between noise strength, model performance, and neuron change ratio, providing valuable insights into the model’s internal dynamics."
    - `yV54`: "Applying SAE-based interpretability techniques to vision-language models is a promising research direction for mechanistic interpretability in multimodal models."
+ **Clear motivation and relevance**
    - `yV54`: "This paper tackles visual hallucinations in LVLMs, an important problem that directly impacts the trustworthiness and safety of multimodal AI systems."
    - `Ttv4`: "The authors use SAE to interpret and manipulate the internal visual features, showing how image noise alters neuron activations, disrupts semantic structure, and leads to hallucinated outputs. " & "They demonstrate that targeted neuron-level interventions—enhancing or suppressing specific neurons—can effectively control LVLM outputs, and that coordinated multi-neuron modulation is more powerful than single-neuron manipulation."
+ **Simple, elegant, and efficient method**
    - `MCny`: "The method is conceptually intuitive, leveraging noisy–clean contrasts to steer neuron activations and does not require retraining."
    - `Q189`: "CNS is simple, efficient, and can be applied plug-and-play at inference time."
    - `Ttv4`: "The approach is lightweight and can be applied efficiently without excessive computational overhead."
+ **Strong compatibility with existing methods**
    - `MCny`: "CNS can be combined with decoding-based techniques such as VCD or M3ID, showing consistent improvements across multiple LVLMs."
    - `Ttv4`: "CNS integrates seamlessly with existing decoding-based approaches and achieves consistent hallucination reduction."
+ **Solid experimental validation**
    - `MCny`: "The authors conduct extensive experiments and provide thorough analyses, achieving improvements across multiple benchmarks."
    - `Ttv4`: "The proposed applications demonstrate convincing performance improvements."
+ **Good presentation and clarity**
    - `Ttv4`: "The paper is well written and clearly structured."
    - `MCny`: "The paper demonstrates clear organization and strong explanations of insights."

**- (2) **_**Changes to PDF**_** -**

We have proofread the paper and added extra experimental results in the revised version (highlighted in blue).

**Main text**

Following the reviewers’ recommendations, we made the required corrections and included additional analyses and experiments to further support our claims:
+ `Lines 56-57` **We have moderated the description.**
+ `Lines 248-255` **We have added the detailed experimental setup.**
+ `Figure 5` **Add the analyses of always-active neurons and image-specific neurons.**
+ `Table 1` **Effects of suppressing always-active, image-specific, and random neurons.**

**Appendix**

Additional experiments and analyses have been incorporated in response to the reviewers' suggestions:

+ `Section F.1` **Evaluation on more challenging benchmarks.**
+ `Section F.2` **Evaluation on more methods and models.**
+ `Section F.3` **More cases illustrating the link between hallucinations and neuron activations.**
+ `Section F.4` **Scaling SAEs.**
+ `Section F.5` **Discussion and comparison with previous approaches that edit internal representations of VLMs.**
+ `Section F.6` **Discussion and comparison with standard fine-tuning-based robustness techniques.**
+ `Section F.7` **Discussion and comparison with register neurons.**
+ `Section F.8` **Comparison against both training-free and fine-tuning–based hallucination mitigation methods.**
+ `Tables 10-20` **We have added the additional experimental results.**
+ `Figures 15-16` **We provide additional cases illustrating how hallucinations relate to neuron activation, and show that modulating specific SAE-identified neurons can effectively reduce hallucinations.**

---

### Comment · Area_Chair_Sbqh · 2025-11-26
**Author-Reviewer-AC Discussion (DDL: 12/3 9PM UTC)**

Dear Reviewers,

Thank you once again for your service to ICLR 2026. Now that the authors have submitted their rebuttal, I kindly ask you to take the following steps (if you have not done so already):

- Read the authors’ response and other reviews.
- Consider whether the rebuttal and additional comments affect your assessment of the paper.
- Engage in **interactive discussion** with the authors. You may post the feedback to the authors so that they can further follow up. If you have more concerns/questions (e.g., requesting clarifications, new results), it is recommended to post your request *asap*, so that the authors have enough time to address them. **Note the Author-Reviewer-AC discussion period ends on 12/3 9PM UTC**.

The current reviews for this paper are **mixed (scores: 2/4/6/4)**. Your further contributions are essential for forming a well-informed final decision.

I am happy to join and support the discussions between you and the authors. Please feel free to share your thoughts and participate actively in the discussion. Thanks!

Best regards,

AC

---

### Author Response · Authors · 2025-12-02
**Rebuttal Summary (1/2)**

# Rebuttal Summary
We sincerely thank all reviewers for their insightful comments and constructive suggestions, **with a reviewer confirming that "concerns have been addressed" (MCny).**



To assist the Area Chair in evaluating our submission, we summarize the key feedback and our major updates below.

---

## **1. Summary of Reviewer Feedback**
We thank the reviewers for their constructive feedback and their recognition of our key contributions, including:

● **Novel and insightful interpretability perspective**
Reviewers consistently praise the paper for bringing new interpretability insights to LVLM hallucinations.
○ MCny: _“Insightful neuron-level analysis… revealing how perturbations reshape visual semantics, representing a clear advancement over decoding-based methods.”_
○ Q189: _“Identifies  'always-on' neurons that encode global statistics, which is a potential insight for interpretability.”_
○ Ttv4: _“Reveals the relationship between noise strength, model performance, and neuron change ratio, providing valuable insights into internal dynamics.”_
○ yV54: _“Applying SAE-based interpretability techniques to vision-language models is a promising direction for mechanistic interpretability.”_

● **Simple, elegant, and efficient method**
○ MCny: _“Conceptually intuitive… leverages noisy–clean contrasts without retraining.”_
○ Q189: _“Simple, efficient, and plug-and-play at inference time.”_
○ Ttv4: _“Lightweight and efficient without computational overhead.”_

We have actively engaged with all reviewers during the discussion and provided detailed clarifications supported by new experiments and analyses.

---

## **2. Key Clarifications**
### (1) Further analyses on neuron behavior and its connection to hallucinations
We thank the reviewers for the insightful suggestions. In response, we conducted additional analyses linking neuron behavior to hallucinations:

1. We first analyzed how different neuron types respond to image noise. As noise increases, model accuracy drops and the neuron-change ratio rises. Always-active neurons remain largely stable, while image-specific neurons vary significantly, indicating that hallucinations are primarily driven by disruptions to image-specific neurons.
2. We then performed targeted neuron suppression experiments. Suppressing Always-active or random neurons has minimal effect, whereas suppressing image-specific neurons severely degrades accuracy. This confirms that image-specific neurons are the main carriers of input-dependent semantics and are most sensitive to perturbation.
3. Finally, we added fine-grained visual reasoning cases. When irrelevant neurons dominate, the model hallucinates nonexistent attributes or objects. By selectively enhancing relevant neurons and suppressing irrelevant ones, CNS restores correct predictions, providing causal evidence that neuron-level modulation mitigates hallucinations.

These analyses directly motivate CNS, which enhances image-specific neurons while suppressing stable, content-irrelevant neurons.

---

### (2) Broader comparisons across mitigation methods, model variants, and challenging benchmarks

(a) **More hallucination-mitigation baselines:** We evaluated CNS against a broad set of methods—including training-free, fine-tuning, DPO-based, and representation-editing approaches (DoLA, OPERA, VCD, Woodpecker, LURE, HALC, CODE, EAH, VHR, AD-HH, SID, SECOND, MFCD, EFUF, HA-DPO, POVID, CLIP-DPO, RLAIF-V, TPO, SENTINEL, jiang et.al).  CNS consistently reduces hallucinations while maintaining or improving caption quality, demonstrating strong compatibility and reliability.

(b) **More challenging benchmarks:** We further tested CNS on AMBER, MMHal-Bench, and HallusionBench to assess complex reasoning and hard hallucination scenarios. CNS consistently improves grounding, reduces hallucination rates, and strengthens feature-aware reasoning.

Overall, these evaluations show that CNS is broadly effective across **diverse LVLM architectures**(MiniGPT-4, mPLUG-Owl2, LLaVA-Next, LLaVA-OneVision, Qwen2.5-VL), mitigation methods, and benchmark settings.

## **3. Revisions to the Manuscript**
In response to reviewer suggestions, we have revised and enhanced the manuscript, with all changes **highlighted in blue**. We proofread the paper and added additional experimental results.

### Main text
+ `Lines 248-255` **We have added the detailed experimental setup.**
+ `Figure 5` **Add the analyses of always-active neurons and image-specific neurons.**
+ `Table 1` **Effects of suppressing always-active, image-specific, and random neurons.**

### Appendix
+ `Section F.1-F.8` **Additional comparisons across more methods, models, and benchmarks**
+ `Tables 10-20`**Additional experimental results.**
+ `Figures 15-16` **Additional cases illustrating how hallucinations relate to neuron activation, and show that modulating specific SAE-identified neurons can effectively reduce hallucinations.**

---

---

### Author Response · Authors · 2025-12-02
**Rebuttal Summary (2/2)**

## **4. Reiterating Core Contributions**
(1) **We introduce SAEs to interpret and steer LVLM internal visual representations**, providing extensive analyses and visualizations that reveal how image noise perturbs neurons, disrupts semantic structure, and ultimately induces hallucinations.

(2) **We show that neuron-level interventions are effective for semantic control**, demonstrating that enhancing or suppressing specific neurons can modulate targeted concepts in LVLM outputs, and that coordinated multi-neuron modulation outperforms single-neuron manipulation.

(3) We propose CNS, a simple, efficient, and plug-and-play method that amplifies image-relevant neurons and suppresses perturbation-induced activations for robust hallucination mitigation.

(4) **We demonstrate that CNS is highly compatible with existing decoding approaches**, yielding consistent hallucination reduction across diverse LVLM architectures and benchmarks.

---

We hope these clarifications, experiments, and revisions adequately address the reviewers' concerns. We sincerely thank the Area Chair and reviewers for their time and thoughtful feedback.

---

### Meta-Review · Area_Chair_4zAj · 2025-12-15

**Summary:**

This paper introduces sparse autoencoders to interpret and steer the internal visual representations of large vision-language models for hallucination mitigation. The authors propose Contrastive Neuron Steering, a method that selectively amplifies neurons associated with clean images while suppressing those activated by perturbed (noisy) images. The initial reviews were mixed with scores of 2, 4, 6, and 4. Reviewers acknowledged the novel interpretability perspective and the simplicity of the method, but raised several concerns. Key criticisms included a lack of rigorous causal evidence linking neuron behavior to hallucinations, limited experimental validation with insufficient baselines and benchmarks, modest performance improvements, and questions about the novelty compared to existing contrastive methods like VCD. In the rebuttal, the authors provided additional analyses, including new experiments on neuron suppression for different neuron types (always-active, image-specific, random) and expanded evaluations across more models, benchmarks (AMBER, MMHal-Bench, HallusionBench), and mitigation methods. They also clarified the experimental setup and moderated some claims. Unfortunately, before the rollback of the OpenReview system, no reviewers gave a clearly positive response.

**Reviewer Concerns:**

The authors have partially addressed concerns about experimental breadth by adding more benchmarks and baselines. However, core issues regarding the causal link between neuron instability and hallucinations, the marginal nature of performance gains, and the method's novelty remain largely outstanding. One reviewer found their concerns addressed, but the others' fundamental reservations about the paper's contribution and soundness persist.

**Reviewer Scores:**

Reviewer yV54 (score: 2) and Q189 (score: 4) raised significant concerns about causal evidence, baselines, and novelty. Their detailed rebuttal responses suggest they might maintain or slightly adjust their scores downward, as the new experiments do not fundamentally resolve the core methodological and conceptual weaknesses they identified. Reviewer MCny (score: 6) explicitly stated their concerns were addressed and maintained their score. Reviewer Ttv4 (score: 4) questioned the monosemantic claim and novelty; while some clarifications were provided, their concern about the method being compared to fine-tuning approaches was reiterated, likely leading them to maintain a score below the acceptance threshold.

---

### Decision · Program_Chairs · 2026-01-26

Reject